

# GPS-PWV jumps before intense rain events

Luiz F. Sapucci[1], Luiz A. T. Machado[1], Eniuce Menezes de Souza[2], Thamiris B. Campos[3]

[1]Centro de Previsão de Tempo e Estudos Climáticos, Instituto Nacional de Pesquisas Espaciais, Cachoeira Paulista, Postal Code: 12630-000, Brazil.

[2]Departamento de Estatística - Universidade Estadual de Maringá, Maringá, Postal Code: 87020-900, Brazil

[3]Programa de Pós Graduação em Meteorologia, Instituto Nacional de Pesquisas Espaciais, São José dos Campos Postal Code: 12227-010, Brazil

*Correspondence to*: Luiz Sapucci (luiz.sapucci@cptec.inpe.br)

**Abstract.** A rapid increase in atmospheric water vapor is a fundamental ingredient for many intense rainfall events. High-frequency precipitable water vapor (PWV) estimates from a Global Positioning System meteorological site (GPS) are evaluated here for intense rainfall events during the CHUVA Vale field campaign in Brazil (November-December 2011), in which precipitation events of differing intensities and spatial dimensions, as observed by an X-band radar, have been explored. A wavelet cross-correlation analysis shows that there are important spikes in the PWV that precede the more

intense rainfall events on a timescale from 30 to 60 minutes. The correlation and lags between the GPS-PWV and rainfall events are evaluated, and a sharp increase in the GPS-PWV prior to the more intense events has been found and termed GPS-PWV "jumps". These jumps are associated with water vapor convergence and the continued formation of cloud condensate and precipitation particles. The GPS-PWV time-derivative histogram for the period of 60 minutes before the rainfall event reveals different distributions for higher intensity events. This feature could indicate the occurrence of severe precipitation

and consequently has the potential for application in nowcasting activities.

## 1 Introduction

The application of the Global Positioning System (GPS) tropospheric-induced signal delay to estimate the precipitable water vapor (hereafter, GPS-PWV) is a good example of an unconventional solution for quantifying atmospheric humidity. The magnitude of this delay is associated with the atmospheric density (i.e., temperature, pressure and water vapor) (Bevis et al.

1992). The wet component of this delay provides the precipitable water vapor (PWV) (Bevis et al. 1994), with an error of approximately 5% under all weather conditions (Wolfe and Gutman 2000) relative to other measurement techniques (Sapucci et al. 2007) and in near real time (Rocken et al. 1994). An important application of the GPS-PWV estimate is its assimilation into the Numerical Weather Prediction process, which has a positive impact on short-range forecasts of humidity fields and, consequently, better precipitation forecasts for heavy rainfall events (Cucurull et al. 2004; Bennitt and

Jupp 2012). Although the vertical humidity structure is not captured in GPS-PWV estimates, the great advantage of GPS-PWV (in addition to its all-weather capacity) is its high temporal resolution (~minutes). Other promising applications





become viable in dense networks and transects. GPS-PWV has been useful for tracking water vapor advection (Adams et al. 2011), propagating convective events (Adams et al. 2015), determining topographic effects on the evolution of convection and the diurnal cycle (Sato and Kimura 2005; Kursinski et al. 2008; Serra et al. 2016) and nowcasting thunderstorm activities, as foreseen by Jerrett and Nash (2001).

The relationship between the occurrence of intense rainfall and high concentrations of atmospheric water vapor is well known and has been explored in nowcasting applications. PWV data from a microwave radiometer (MWR) with high temporal resolution have been used to describe the observed relationship between the PWV and precipitation in the tropics (Muller et al. 2009). Chan (2009) evaluated the performance of a ground-based microwave radiometer in intense convective weather and reported useful indications of the accumulation of water vapor and an increasing degree of instability of the
troposphere before the occurrence of heavy rain. Madhulatha et al. (2013) reported a sharp increase in the PWV values approximately 2–4 hours prior to the occurrence of thunderstorms and developed a nowcasting technique using PWV values and 7 other thermodynamic indices from microwave radiometer observations.

The relationship between deep convective activity and the temporal evolution of GPS-PWV has been studied for well over a decade. Mazany et al. (2002) developed a lightning prediction index for Florida based on the GPS-PWV magnitude and its
temporal evolution. Nowcasting employing GPS-PWV was reported by de Haan et al. (2004), who demonstrated its viability in improving thunderstorm and heavy precipitation forecasts during a cold front passage. De Haan (2006) developed a method for inferring the atmospheric stability from a nonisotropic GPS path-delay signal (slant delay). More recently, in the tropics, Adams et al. (2013) utilized 3.5 years of Amazon GPS-PWV to derive a water vapor convergence time scale associated with the shallow-to-deep convective transition. This study showed that prior to deep convective events in the
central Amazon, a 4-hour "ramp up" in the time derivative of GPS-PWV is observed, reaching a maximum approximately one hour before heavy precipitation. Given this one-hour lead time, near real-time GPS-PWV could, in principle, be employed to improve the nowcasting of these events.

The motivation of the present study is to evaluate the large and rapid increases in the PWV characteristics prior to the onset of deep convective rainfall events observed during the CHUVA (Cloud processes of tHe main precipitation systems in
Brazil: a contribUtion to cloud resolVing modeling and to the GPM (GlobAl Precipitation Measurement)) Vale experiment in Brazil in 2011 (Machado et al. 2014) using data from a GPS meteorological site. The sharp increase in the GPS-PWV values approximately one hour before the occurrence of more intense rainfall events, as found in this study and that of Adams et al. (2013), is hereafter termed the GPS-PWV jump. The GPS-PWV time series was evaluated using wavelet analysis in a study of correlation and lags with rainfall events to form a conceptual model with predictive capacity that can
hence be used in developing a nowcasting tool for strong precipitation events.

In section 2, we present the data collected during the CHUVA Vale experiment in Brazil and describe the GPS data processing technique for obtaining PWV values. An X-band radar was utilized to quantify the spatial-temporal distribution of the precipitation events. Section 3 presents the wavelet analysis for PWV and the precipitation time series. In section 4, a



case study of GPS-PWV jumps is characterized, the time lag correlations between the precipitation and GPS-PWV time series are studied, and the GPS-PWV derivatives before precipitation events of different intensities are evaluated. Section 5 presents the conclusions.

## 2 Data collection design and processing method

The data employed in this study were collected during the CHUVA Vale campaign, one of the field campaigns of the CHUVA project (Machado et al. 2014). The CHUVA project was designed to investigate cloud microphysical and precipitation processes through six intensive field campaigns covering various precipitation regimes in Brazil. The CHUVA Vale campaign was carried out in São José dos Campos in São Paulo State in an elevated valley between the Mantiqueira and Serra do Mar mountain ranges. This region is dominated by deep convection with typical rainfall systems that are forced

by sea breeze-mountain convergence zones as well as squall lines associated with cold front penetration (Machado et al. 2014). During the CHUVA campaigns, GPS meteorology was used to monitor the horizontal and temporal variations in the PWV associated with the wide variety of convection-producing mechanisms for the 6 geographic regions. For example, Adams et al. (2015) described the temporal and spatial evolution of tropical sea-breeze convection with GPS meteorological transects during the CHUVA-Belem field campaign. The next subsections describe the data collected and the processing

techniques applied to obtain the time series of the GPS-PWV and precipitation.

### 2.1 Data from the CHUVA Vale experiment

The CHUVA Vale experiment performed in São José dos Campos City (23° 12' 30" S and 45° 57' 08" W) consisted of an intensive observation period from November 3rd to December 28th of 2011. The instruments used in this study were a dual-frequency GPS receiver for scientific applications, a disdrometer and a mobile X-band dual polarization radar (XPol). The

TRIMBLE brand GPS receiver, model NETR8, utilized in this study was installed 11 km from the XPol radar at the site denominated the Institute of Advanced Studies from the Department of Aerospace Science and Technology (IEAV) site. Any sky obstructions around the GPS receiver were avoided to minimize the multi-path effect in GPS signal propagation. Fig. 1 shows the geographic location of the CHUVA Vale campaign, emphasizing the sites at which the instruments were placed.

The GPS satellite signals were sampled at one-second frequencies, whereas the collocated meteorological sensor captured the pressure and temperature at one-minute frequencies. A Joss-Waldvogel brand acoustic impact disdrometer (Joss and Waldvogel 1967), model RD 80, was installed a few meters from the GPS receiver. The XPol radar scan strategy collected one volume scan every 6 minutes at 13 elevations, from 1° to 25°, with 1° and 150 m angular and radial resolutions, respectively. The radar data were pre-processed using the attenuation correction of the reflectivity, employing the algorithm

for ground-based polarimetric radars (ZPHI algorithm) proposed by Testud et al. (2000). For a detailed description of the radar and disdrometer pre-processing, see Calheiros and Machado (2014).



## 2.2 High temporal resolution GPS-PWV time series

The zenith total delay (ZTD) was obtained by processing the GPS data using GOA-II [(Gipsy, GPS Inferred Positioning System) (OASIS, Orbit Analysis and Simulation Software II) Gregorius 1996] software by applying the precise point positioning method in post-processing mode with the precise ephemerides of the GPS constellation provided by the NASA

Jet Propulsion Laboratory. A Global Mapping Function (Boehm et al. 2006) with a cut-off elevation angle of 10° was used. To ensure the quality of the PWV time series with high temporal resolution required in this study, a rigorous data-processing strategy was adopted with possible noise sources taken into consideration. The latest version of the GOA-II software (version 6.3) was used. The ocean tide model FES 2004, recommended by the International Global Navigation Satellite System Service, was applied in this processing (Lyard et al. 2006), and absolute calibration was performed to ensure the

correct phase center variation, as reported by Görres (2006). The zenith wet delay was obtained from the ZTD after removing the zenith hydrostatic delay obtained through the application of a representative tropospheric temperature model and a surface pressure measurement (Davis et al. 1985). The zenith wet delay was converted into the PWV using the relationship suggested by Bevis et al. (1992). The mean tropospheric temperatures ($T_m$) were obtained from the temperature and pressure measured at the GPS antenna by applying the regional model suggested by Sapucci (2014), which is the most

suitable for this region. The sampling rate of the GPS-PWV values was 1 minute. The GPS-PWV time series suffered some short failures due to interruptions in data collection, problems with the satellite ephemerides, unavailable pressure measurements and other unknown causes. These time series interruptions occurred in 3,183 epochs (3.1% of the total period), and the missing values were filled by a cubic spline interpolation method. These failures were concentrated in two specific days [Day of Year (hereafter called DoY) 331 and 348] and were taken into consideration during the data analysis.

## 2.3 Precipitation time series from disdrometer and XPol radar data

The reason for choosing one disdrometer to quantify the precipitation is that this instrument is able to provide an instantaneous measurement of the rainfall intensity (mm h$^{-1}$) at the same GPS-PWV sampling rate; however, this information is only representative of a very small spatial scale. Notably, the XPol radar data are able to generate information with a different and more representative spatial resolution around the GPS receiver.

The precipitation rates were obtained from the Joss disdrometer data by applying the methodology suggested by Kinnell (1976), in which the precipitation intensity of each minute is inferred from the size and concentration of the rain drops observed during this time period. The precipitation data from the XPol radar observation were obtained by applying the Dual Polarization Surface Rainfall Intensity algorithm, which calculates the rainfall rate ($R$) from the reflectivity ($Z$) and specific differential phase data obtained in multiple-elevation polar volumes. In this method, polarimetric measurements are

used to calculate $R$ by applying a combined $Z$ $R$ relationship (Gematronik 2007). The final product of this process is a gridded map (100-km radius around the XPol) with a horizontal resolution of 200 m, in which each of the grid point values of the rainfall intensity (mm h$^{-1}$) at a sampling rate of 6 minutes is available.



The time series of the precipitation fractional area around the GPS receiver observed by radar were calculated by determining the position of the GPS antenna in the gridded precipitation points and taking into consideration the area formed by points in the longitudinal direction for the same number of points in the latitudinal direction where the nearest point of the GPS antenna is located in the center of these areas. Fig. 1 shows the details of the composition of this area using points from

5 the XPol gridded map. The dimensions of the precipitation area that influences the GPS-PWV is a key factor in the development of this study because its choice is associated with the lead time of the GPS-PWV information indicating strong precipitation. A reduced precipitation area around the GPS antenna (e.g., values measured via a rain gauge) is not sufficient to represent the rain associated with a significant variation in the GPS-PWV time series. However, larger areas permit distant precipitation (which is not directly associated with the GPS-PWV measurement) to be taken into account. Different areas

were tested, and an area of 22x22 (longitudinal per latitudinal direction grid point values of rainfall intensity (mm h$^{-1}$)) was found to be more representative of the observed area by GPS and better for exploring the correlation between the precipitation occurrence and GPS-PWV. The resultant area is 4.4 km per 4.4 km (~20 km$^2$) around the GPS antenna. Although this area does not exactly match the cone of the GPS observation, it was chosen because it most highly correlates with the precipitation measurements taken by radar around the GPS antenna. The rainfall area employed in this study is only

a reference for the description of the GPS-PWV jump. Notably, the area employed has the highest correlation with GPS-PWV, but it is expected to vary as a function of the region and satellite configuration. Fig. 1 shows the configuration of the XPol radar, GPS antenna and disdrometer in the CHUVA Vale experiment and highlights the mentioned area of 4.4 km by 4.4 km around the GPS antenna. The precipitation field observed by XPol on December 14$^{th}$ (DoY 348) is presented in Fig. 1 to illustrate the points from the XPol gridded map used to represent the precipitation area around the GPS antenna taken into

account in this study. This event was chosen because it was the most intense, with 70% of the area around the GPS antenna recording precipitation above 50 mm h$^{-1}$.

As the focus here is to study the GPS-PWV behavior during intense rainfall, the statistical measurements calculated from the radar data were in the 95$^{th}$ percentile of the intensity of the precipitation observed in the area of 4.4 km per 4.4 km around the GPS antenna. Additionally, to evaluate rainfall events of different intensities, the rain fraction was computed as the fraction

of the studied area with precipitation rates above some chosen threshold. The first approach emphasizes the more intense localized precipitation events, and the second simultaneously quantifies the intensity and extension of each event. The chosen thresholds were computed to create time series of the following rainfall event intensities: moderate to heavy (20 mm h$^{-1}$), heavy to intense (35 mm h$^{-1}$) and intense to torrential (50 mm h$^{-1}$). The disdrometer was used in this study only for a reference and for comparison with the radar rainfall estimations. Fig. 2 presents time series of the GPS-PWV (Fig. 2a), the

precipitation from the disdrometer data (Fig. 2b) and, from the XPol radar, the 95$^{th}$ percentile of the precipitation intensity (Fig. 2c) and the rain fractions for different thresholds (Fig. 2d). The period studied here is from November 9$^{th}$ (DoY 313) to December 28$^{th}$ (DoY 362) of 2011, during which the GPS receiver, XPol radar and disdrometer were simultaneously





collecting data. The disdrometer time series has a good correlation with the 95$^{th}$ percentile time series, although differences are expected due to the different areas covered by each instrument.

## 3 Wavelet analysis

Wavelet analysis was used to perform a detailed analysis of the GPS-PWV time series and to evaluate the variability within
different time scales (denoted here as intra-relation), as well as to assess the relationship between the GPS-PWV time series and the precipitation time series (denoted here as interrelation wavelet analysis). The wavelet analysis was used to decompose the GPS-PWV data in time-variability space to permit an evaluation of the main frequencies composing the GPS-PWV time series during intense rainfall events. As argued by Adams et al. (2013), the PWV increase prior to heavy deep convective precipitation principally results from water vapor convergence, given that the surface evaporation is small in
the cloudy, showery conditions preceding the event. Therefore, a specific time scale should be associated with this convergence, providing a lead time for nowcasting, a specific time scale for calculations and additional information to understand the physical mechanisms associated with intense rainfall events in this region.

This methodology enables simultaneous decomposition of the PWV time series as a function of time and frequency (Daubechies 1992). Consequently, access to the information regarding the signal amplitude/frequency and its variation as a
function of time becomes possible. In this study, both continuous and discrete wavelets are investigated to achieve intra- and interrelation analysis, respectively. The mother wavelets used in the continuous and discrete cases were Morlet (Torrence and Compo 1998) and Symlets, respectively. Because the bivariate relationship between the two time series is essential for this study, a wavelet cross-correlation (WCC) constructed from the non-decimated discrete wavelet transform (NDWT) (Whitcher et al. 2000; Percival and Walden 2000) (which is time invariant and ideal for analyzing different scale structures
and the interrelations of the dynamic behavior of two time series) was used to represent the lead-lag relationships. Some lead-lag relationships that could not be distinguished in the usual cross-correlation can be investigated in the WCC, which decomposes the cross-correlation on a scale-by-scale basis.

In practice, the NDWT is easily calculated using the pyramidal algorithm, but it requires a time series of dyadic length $n=2^J$, where $J$ represents the largest scale. Thus, the two time series were restricted to a power of two, with lengths of 65,536 ($2^{16}$)
observations, excluding one day at the beginning and another at the end of the GPS-PWV series. The time span corresponds to 45 days, 12 hours and 16 minutes, beginning on DoY 314 (00 UTC 10 November) and finishing on DoY 359 (1216 UTC 27 December). The same time series length was used for both the intra- and interrelation analyses. Because each scale $j$ corresponds to a frequency band from $2^j$ to $2^{j+1}$, the inversion of which produces the period of time, an interrelation analysis can be performed with the WCC of GPS-PWV and precipitation time series to evaluate the correlation at different lags in
time, scale by scale. This enables the identification of the most important time scale of the PWV oscillation during precipitation events.



## 3.1 Wavelet power spectrum analysis

For an intra-relation analysis, the wavelet power spectrum of the GPS-PWV and the 95[th] percentile of the precipitation intensity time series are presented in Fig. 3. To emphasize the highest-frequency oscillations, the power spectrum in Fig. 3 shows the scales that represent the period below 512 minutes (~8.5 h). The PWV diurnal cycle presents strong power along all time series, and consequently, it was not taken into consideration in this analysis.

The methodology employed to process the GPS data in one-minute intervals did not provide any additional information. Fig. 3 shows that the GPS-PWV energy variability begins to be significant only for time scales longer than 16 minutes. Therefore, the one-minute time series representativeness is not a limitation, and if there is noise, it is white noise. The PWV series with a one-minute temporal resolution presents oscillations of high frequency; however, the frequency of occurrence of rainfall events is very low. For this reason, it is necessary to take into consideration a long time period (with several precipitation events) and a short time step, e.g., the one-minute interval used in this study. Consequently, in a general analysis of the PWV wavelet power spectrum, it is difficult to clearly discern which power is associated with each time step. However, a more specific analysis of the wavelet power spectrum during precipitation events indicates that there are expressive changes in the power between different time scales in those cases in which an increase in the power of the oscillation from low to high frequency is observed. This result indicates that PWV oscillations on time scales smaller than 128 minutes occur more frequently during precipitation events than in periods without rain. Fig. 4 presents the same wavelet power spectrum presented in Fig. 3 but with an amplification applied for precipitation events observed during the period from DoY 340 to DoY 343. The analysis of the power spectrum in these cases makes the result discussed for Fig. 3 more evident. A vertical line was put at the peak of the maximum precipitation in each event to simplify the analysis. Fig. 4 shows that the power of the PWV oscillations between 128 and 32 minutes is more expressive during more intense precipitation events (for example, the events that occurred on DoY 341 at 1836 UTC and DoY 342 at 1636 UTC) than during light rainfall events (events observed during DoY 343) and periods without rain (DoY 340). The wavelet power spectrum (Fig. 3) also shows the impact of the GPS failures that occurred during DoY 331 and 348, which are unfortunately very close to the other intense precipitation events that occurred on DoY 332 and in the beginning of DoY 348. For this reason, these cases are not spotlighted in the wavelet analysis.

## 3.2 Wavelet cross-correlation analysis

Although the precipitation and GPS-PWV time series present very distinct behaviors, the WCC permitted the identification of the correlation in each time scale, indicating which GPS-PWV oscillations are important for predicting precipitation events. Fig. 5 shows the WCC between the GPS-PWV and the 95[th] percentile of the precipitation intensity as a function of the wavelet scale. The mother wavelet used in the NDWT to compute the WCC was a Symlet wavelet (SYM10), which has good properties, such as quasisymmetry (Daubechies 1992). The results show that the wavelet correlation between the PWV and precipitation intensity is more evident and significant for the time scale between 32 and 64 minutes, indicating the time





scale on which the most important GPS-PWV oscillations associated with precipitation events occur. After this time scale, the correlation decreases, followed by another increase due to the influence of the diurnal cycle. To evaluate the results presented in Fig. 5 for different intensities of precipitation, Fig. 6 shows the WCC between the GPS-PWV and precipitation fractions as a function of the wavelet scale for different rain fraction intensities (>20 mm h$^{-1}$, Fig. 6a; >35 mm h$^{-1}$, Fig. 6b; and >50 mm h$^{-1}$, Fig. 6c). The peak of the WCC observed in the time scale from 32 to 64 minutes is only significant for rain fractions larger than 35 mm h$^{-1}$. The plots of 6b and 6c show that this peak is more expressive when only heavy to torrential precipitation events are taken into consideration (hereafter called intense rain events). This result indicates that the GPS-PWV carries some information on the time scale from 32 to 64 minutes that signals the occurrence of more intense precipitation events, emphasizing its potential in a nowcasting application. It is also important to highlight that the correlation is larger on this time scale than on adjacent scales. This information corroborates the results reported by Adams et al. (2013), who showed that the strongest water vapor convergence is typically ~1 hour before heavy precipitation. However, this feature should be considered as specific to each region because the time scale between the humidity convergence and rainfall may depend on many physical processes and environmental conditions (e.g., gravity wave-induced convergence, wind shear or thermodynamic instability).

## 4 Behavior of PWV time series before precipitation events: the GPS-PWV jumps

As described above, the high temporal resolution obtained with the GPS-PWV enables the evaluation of high frequency variations and their relationship with intense precipitation events. The GPS-PWV series shows a well-defined pattern before the occurrence of precipitation. There are strong oscillations, predominantly positive, generating a significant increase in the total water vapor content until a maximum peak is reached. Subsequently, a strong GPS-PWV reduction is observed, and after a short interval, the precipitation also reaches a maximum peak. An analysis of the time lag correlation is necessary to define this interval between the maximal GPS-PWV and precipitation, which is carried out in the next subsection. Here, this pattern is called GPS-PWV jumps. The physical explanation for this behavior could be explained by different physical processes. First, the water vapor may increase through low-level moisture convergence, as suggested by Adams et al. (2013), on the time scale of 32-64 minutes, as suggested by the wavelet cross-correlation analysis. The variation of the moisture convergence generates a sequence of pulses of positive increases in the PWV value. The presence of low-level water vapor convergence can be attributed to mechanisms such as gravity wave forcing (Raymond 1987) or other larger-scale forcing mechanisms. The increased water vapor convergence (i.e., positive PWV derivatives) may also simply be a reflection of the unstable surface parcels accelerating upwards, thereby vertically advecting a larger surface specific humidity to higher levels in the atmosphere without necessarily any larger-scale dynamical forcing. Given the limitations of the observations, our interpretation of the physical mechanisms responsible for the jumps remains speculative. After the increase in the PWV at the crest of the jump, rainfall begins and the PWV begins to decrease. In our case, we show the 95% percentiles of the precipitation field; some positive lag between the maximum PWV and the 95% rainfall percentiles is therefore expected.



Adams et al. (2013) considered that on the scale of the GPS cone of observation, the conversion of water vapor to cloud liquid water is secondary in the progression toward deep convection. That is, the cloud water sink is compensated for by the increased vertical acceleration of convective updrafts and resulting water vapor convergence in the column until strong precipitation begins.

Fig. 7 shows a typical case exemplifying the PWV behavior before precipitation occurs on DoY 341; this was one of the strongest events registered during the CHUVA Vale experiment. Before the precipitation begins, the GPS-PWV follows several pulses, increasing the value and forming the PWV jump, until it reaches a peak of maximum value. After the GPS-PWV crest, a decreasing period is observed some minutes before precipitation. Fig. 7 clearly shows this configuration of a crest in the GPS-PWV time series around precipitation (composed of several pulses) and its subsequent decrease immediately before the beginning of precipitation. The wavelet analysis indicates that oscillations on the timescale from 32 to 64 minutes are associated with a more intense rainfall occurrence, which suggests that the GPS-PWV jumps are, for these storms evaluated, concentrated in this time scale. This GPS-PWV behavior before precipitation occurs not only for more intense events but also for lower rainfall rates, as shown in Fig. 4 before the event that occurred on DoY 343 at 2330 UTC. However, the intensity of these oscillations (jumps) is greater before more intense precipitation, which corroborates the results presented by the WCC analysis discussed in the previous section.

## 4.1 Time lag correlation analysis

The relationship between the rainfall intensity (or rain fraction) and the GPS-PWV is different for each event; however, the GPS-PWV peak is a well-delineated pattern. The time interval between the time of the PWV crest and the maximal precipitation can vary between cases. The WCC shows on which temporal scale (lags) the correlation between the GPS-PWV and precipitation time series is more evident. However, evaluating the lag correlation for positive and negative correlations is also important for further developing nowcasting tools. Fig. 8 shows histograms of the lag correlations (the time lags for the maximum and minimum correlations) between the GPS-PWV and rain fraction for all events in which the rainfall rate observed around the GPS antenna was above 20 mm h$^{-1}$. A total of 18 events were evaluated, in which different rain fractions were observed using XPol radar. Table 1 presents these precipitation events and the respective precipitation values observed using the radar. The histogram was constructed based on the correlation time lags found for the positive correlation (maximum) and negative correlation (minimum). The search time lag interval for the positive (negative) correlation was restricted for a period before (after) the precipitation because there are some subsequent rain events very nearby that could distort the results.

The type of correlation used in this study was Spearman's $\rho$ (Best and Roberts 1975) because the GPS-PWV does not have a normal distribution. Additionally, the statistical significance of each correlation was evaluated by rejecting the hypothesis of a null correlation when the p-value was smaller than a 5% significance level.

The histograms presented in Fig. 8 indicate that the GPS-PWV crests are more frequent in the time interval between 15 and 30 minutes before maximum precipitation occurs (39% of the cases evaluated), and 85% of the positive correlations occurred



between 15 and 60 minutes. After the rainfall events, when the GPS-PWV decreased, the time lag observed was between 15 and 60 minutes. In 50% of the cases, the minimum GPS-PWV occurred between 45 and 60 minutes. As already mentioned, several mechanisms can be responsible for the PWV decrease, and the specific physical process for each event can vary. This result corroborates the pattern observed in Fig. 7, showing the GPS-PWV maximum before the precipitation event and its

minimum after the maximum precipitation, which indicates a conceptual model that can be explored for nowcasting applications.

## 4.2 GPS-PWV derivative analysis

To evaluate the GPS-PWV derivative before precipitations of different intensities, taking into consideration the extension of the events, the area fraction with precipitation observed was used. Fig. 9 shows the distribution of the GPS-PWV time

derivative for the period of 60 minutes before the maximum peak of precipitation for the 18 evaluated events for different terciles of rain fraction (>35 mm h$^{-1}$). This threshold was employed in this analysis because it has been shown to be the lowest threshold of rainfall intensity evaluated that presents a significant WCC between the GPS-PWV and precipitation in this time window. Table 1 shows the list of rainfall events in the terciles and the precipitation fraction observed using XPol radar during the maximum precipitation peak for each event. In Fig. 2d, each event can be visualized by observing the

respective DoY. The same calculation was conducted for all other rainfall events with intensities smaller than 20 mm h$^{-1}$ and periods without observed precipitation, which is denominated in Fig. 9 as "other cases". The derivatives were calculated every minute using a $\Delta t$ of 6 minutes. This interval time was selected because it is the sampling rate of the precipitation observed using XPol radar. The statistical metrics of the derivative for different terciles of the precipitation intensity are shown in Table 2.

Fig. 9 clearly shows an expressive change in the pattern of the derivative distribution as a function of the different precipitation intensity terciles. In the period without significant rain, the derivative frequency distribution is similar to a Gaussian distribution with an average of zero and a standard deviation of 2.5 mm h$^{-1}$. However, the average of the derivative increases when the precipitation fraction increases, and the maximum peak changes to positive values for higher terciles. Notably, this effect is evident for the middle and upper terciles and nearly undetectable for the lower terciles. The lower

terciles represent the events with observed precipitation in a reduced area, and the derivative histogram shows that in these cases, the differences for rain events smaller than 20 mm h$^{-1}$ and periods without observed precipitation are not expressive. This pattern impacts the average value of the derivative, which is +0.13 kg m$^{-2}$ h$^{-1}$ ($\pm$5.57 mm h$^{-1}$) for the upper tercile, -0.38 mm h$^{-1}$ ($\pm$4.76 mm h$^{-1}$) for the middle tercile and -0.18 mm h$^{-1}$ ($\pm$3.18 mm h$^{-1}$) for the lower tercile.

The negative derivatives (the majority of which occur after the peak maximum GPS-PWV) less than -9.5 mm h$^{-1}$ are more

frequent for events with a larger precipitation fraction, such as the upper and middle terciles (5.47% and 4.68%, respectively), than for the events contained in the lower tercile, in which these derivative values are not observed. This increase in the stronger negative derivative frequency before more intense events (upper and middle terciles) is associated with the conversion process from water vapor to liquid water, which is also more intense in these cases.





### 4.3 Potential of the GPS-PWV jumps for nowcasting application

Although the GPS-PWV pattern before precipitation described in the previous section is well defined, its use as an index for the occurrence of severe storms is not simple. Several studies have taken into account the intensity of rainfall events. The use of only a maximum threshold from a GPS-PWV derivative, as suggested by Iwabuchi et al. (2006), is not sufficient to

predict intense rainfall because, in nature, the processes responsible for the maintenance of precipitable water suspended in the atmosphere are very complex and highly nonlinear. The intensity of the precipitation is highly correlated with the intensity of the PWV value, which is formed by a succession of pulses of positive increases in the PWV value. The increase in moisture convergence appears to be due to an increase in the frequency of convergence pulses. Therefore, the occurrence of more intense rain is signaled not only by the maximum derivative but also by the increase in the frequency distribution of

the positive and negative derivatives before the occurrence of precipitation.

The GPS-PWV derivative analysis showed that the majority of the positive derivative frequencies occur before the peak maximum GPS-PWV. The positive derivative above $+9.5$ mm h$^{-1}$ increases more substantially before rainfall events in the upper tercile (7.81%) than for events in the middle (0.78%) and lower terciles, which are not observed. In other words, these derivatives are important in the frequency distribution when the rainfall tercile increases and are significant before the most

precipitation-intense events. This result suggests that an algorithm for intense precipitation forecasting using the GPS-PWV should consider the following points: (a) increases in GPS-PWV positive variations compared with negative ones in which the median values of the variation in the last 60 minutes reach positive values and (b) a simultaneous increase in the population of the GPS-PWV derivatives above $+9.5$ mm h$^{-1}$.

The GPS-PWV values evaluated in this study are post-processed, and an additional study is required to determine whether

these estimates in real time are able to capture the jumps before the precipitation reported. True real-time processing for PWV estimates in dense and regional GPS networks has been explored in other studies related to nowcasting applications (Iwabuchi et al. 2006). De Haan and Holleman (2009) reported the construction and validation of a real-time PWV map from a GPS network combined with data from weather radar, a lightning detection network, and surface wind observations. They tested a nowcasting algorithm for three thunderstorm case studies and concluded that the GPS-PWV in real time can be

helpful for the nowcasting of severe thunderstorms.

### 5 Conclusions

This work evaluates the correlation between large and rapid increases in the GPS-PWV and the occurrence of intense rainfall events observed by radar during the CHUVA Vale experiment in Brazil. The wavelet analysis for the GPS-PWV time series was explored, and it clearly shows that during precipitation events, there are expressive changes in the power spectrum

between different time scales, in which an increase of the power of the oscillation from low to high frequency is observed. Additionally, the application of wavelet cross-correlation between the PWV and precipitation showed that important oscillations exist between these variables on the time scale from 32 to 64 minutes, which is more evident and significant for





events of large intensity, corroborating the results reported by Adams et al. (2013), who showed that the strongest water vapor convergence is typically ~1 hour before heavy precipitation.

A detailed analysis of the GPS-PWV time series was carried out, and strong and sudden oscillations, predominantly positive, before the precipitation events were identified and called GPS-PWV jumps. In this process, a crest in the PWV series is
remarkable before the precipitation events. Although this pattern can be observed for any precipitation event, it is preponderant before more intense precipitation events, similar to results found by Adams et al. (2013). A time lag-correlation histogram shows that in 85% of the studied events, a crest in the PWV time series occurs between 15 and 60 minutes before the maximum precipitation. The GPS-PWV derivative histogram shows the distribution change for different precipitation intensity terciles. The average values of the GPS-PWV derivatives present an increase in positive values as a function of the
increase in the rainfall intensity terciles. The results suggest that the derivative average values in the interval of 60 minutes before precipitation changes to positive values, and an increase in the frequency of the derivative above +9.5 mm h$^{-1}$ can indicate the occurrence of severe precipitation. Consequently, a methodology based on the monitoring of the GPS-PWV derivative histogram presents potential for exploration in nowcasting applications, but additional studies will be necessary to define an appropriate algorithm and characterize the skill of this tool.

**Competing interests:** The authors declare that they have no conflict of interest.

**Acknowledgments:** The authors are grateful to David Adams for his constructive comments, which helped us to improve the final technical and scientific quality of this paper. Thanks are given to the CHUVA team who were involved directly or
indirectly in the data collection by XPol radar, disdrometer and GPS receiver during the CHUVA-VALE experiment in São José dos Campos-S.P. Special thanks is given to Thiago Souza Biscaro. This study was supported by the Fundação de Amparo a Pesquisa do Estado de São Paulo (FAPESP), which directly supported this experiment [Grant Process 2009/15235-8 (CHUVA project)], and the Contractual Instrument of the Thematic Network of Geotectonic Studies CT-PETRO (PETROBRAS) and INPE (Grant: 600289299), which provided the GPS receiver used in this experiment. The raw
data (Level 0) from the XPol radar, disdrometer and GPS receiver used in this study and the values obtained by applying the methodology reported in section 2 (Level 2) are freely available through the Brazilian Institute for Space Research (ftp://pararaca.cptec.inpe.br/) after registration in the CHUVA-PROJECT portal (http://chuvaproject.cptec.inpe.br).

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



**Tables**

**Table 1. Precipitation events observed by radar during CHUVA Vale experiment in different intensity terciles as a function of the precipitation fractions above 35 mm h⁻¹.**

| Event | DoY | Maximal radar precipitation UTC Time (hh:mm) | Precipitation fraction observed by XPol radar (%) | | | Terciles |
|---|---|---|---|---|---|---|
| | | | Above 50 mm h$^{-1}$ | Above 35 mm h$^{-1}$ | Above 20 mm h$^{-1}$ | |
| 1 | 348 | 02:42 | 73 | 85 | 95 | |
| 2 | 354 | 21:12 | 28 | 45 | 63 | |
| 3 | 341 | 18:36 | 36 | 41 | 45 | Upper |
| 4 | 315 | 17:12 | 26 | 41 | 49 | tercile |
| 5 | 335 | 19:24 | 30 | 38 | 42 | |
| 6 | 352 | 20:00 | 2 | 33 | 84 | |
| 7 | 342 | 16:36 | 24 | 27 | 28 | |
| 8 | 326 | 21:18 | 8 | 26 | 45 | |
| 9 | 343 | 01:06 | 4 | 9 | 15 | Middle |
| 10 | 338 | 20:18 | 0 | 8 | 19 | tercile |
| 11 | 332 | 19:18 | 5 | 5 | 5 | |
| 12 | 333 | 19:42 | 0 | 2 | 8 | |
| 13 | 314 | 21:06 | 2 | 2 | 7 | |
| 14 | 327 | 00:36 | 0 | 1 | 25 | |
| 15 | 358 | 23:00 | 0 | 1 | 10 | Lower |
| 16 | 317 | 21:48 | 0 | 1 | 2 | tercile |
| 17 | 318 | 08:48 | 0 | 0 | 19 | |
| 18 | 331 | 17:12 | 0 | 0 | 2 | |



**Table 2. Statistical measurements of the GPS-PWV derivative for different intensity terciles of precipitation events.**

| Statistical Measurements | Other cases | Terciles | | |
|---|---|---|---|---|
| | | Lower | Middle | Upper |
| Average value (mm h$^{-1}$) | +0.04 | -0.18 | -0.38 | 0.13 |
| Standard deviation (mm h$^{-1}$) | ±2.52 | ±3.18 | ±4.76 | ±5.57 |
| Median (mm h$^{-1}$) | 0.00 | 0.00 | +0.29 | +0.65 |
| Mode (mm h$^{-1}$) | 0.00 | 0.00 | +1.00 | +1.00 |
| Maximal value (mm h$^{-1}$) | +21.13 | +8.42 | +11.00 | +13.25 |
| Minimal value (mm h$^{-1}$) | -19.07 | -6.99 | -17.30 | -14.15 |
| % > +9.5 mm h$^{-1}$ | 0.21% | 0.00% | 0.78% | 7.81% |
| % < -9.5 mm h$^{-1}$ | 0.38% | 0.00% | 4.68% | 5.47% |



**Figures**

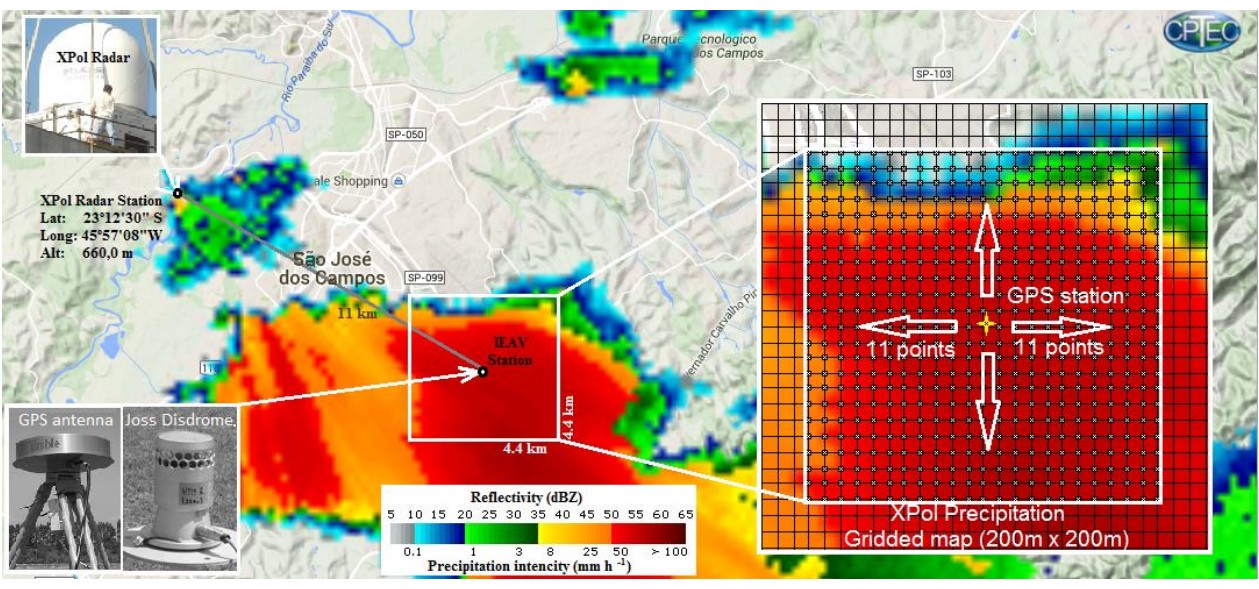

**Figure 1: Illustration of the CHUVA Vale experiment, in which the sites where the XPol radar, GPS receiver and disdrometer were installed are indicated. The area of 4.4 km per 4.4 km around the GPS station is highlighted in this figure over the precipitation field observed by XPol radar on December 14th (DoY 348) of 2011. Some details about the composition of this area using the points of the XPol gridded map are additionally presented.**





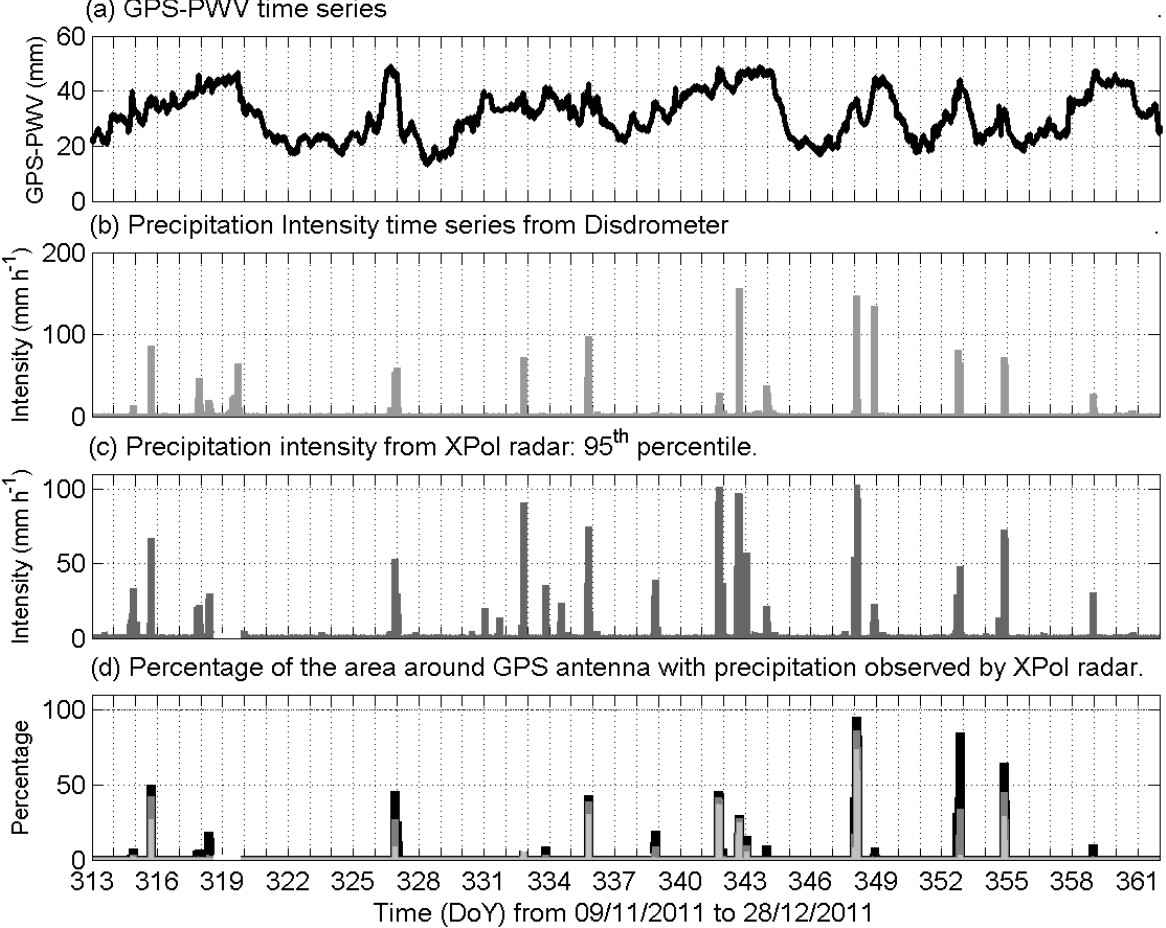

**Figure 2: Time series of the precipitation and GPS-PWV obtained during CHUVA Vale campaign: (a) GPS-PWV time series; (b) precipitation intensity observed by Joss disdrometer; (c) 95th percentile of the precipitation intensity observed by XPol radar in the area of 4.4 km per 4.4 km centered on the GPS antenna; and (d) precipitation fraction in the area of 4.4 km per 4.4 km centered on the GPS antenna, where the black bar is the fraction above 20 mm h$^{-1}$, the dark gray bar is the fraction above 35 mm h$^{-1}$ and the light gray bar is the fraction above 50 mm h$^{-1}$.**





**Figure 3: GPS-PWV time series at IEAV station (a) and wavelet power spectrum analysis (b). The precipitation intensity values observed by XPol radar (95$^{th}$ percentile) in the area of 4.4 km per 4.4 km around the GPS antenna are included in the bottom of this plot.**



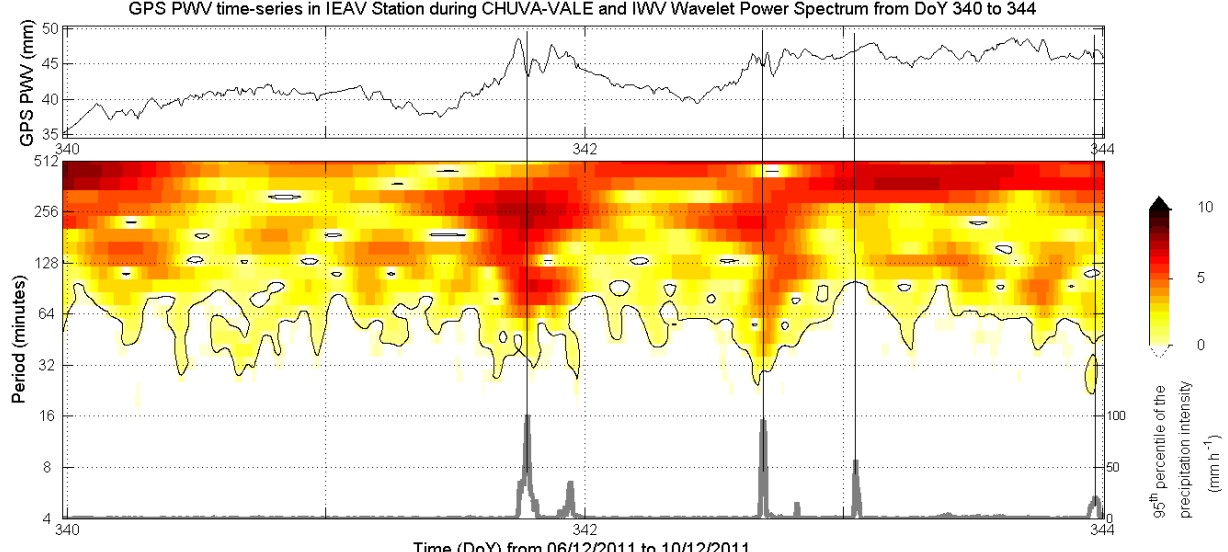

**Figure 4: Similar to Fig. 3 for a shorter period (from 0000 UTC on DoY 340 to 0000 UTC on DoY 344) to emphasize the details in the wavelet power spectrum during the occurrence of precipitation events.**




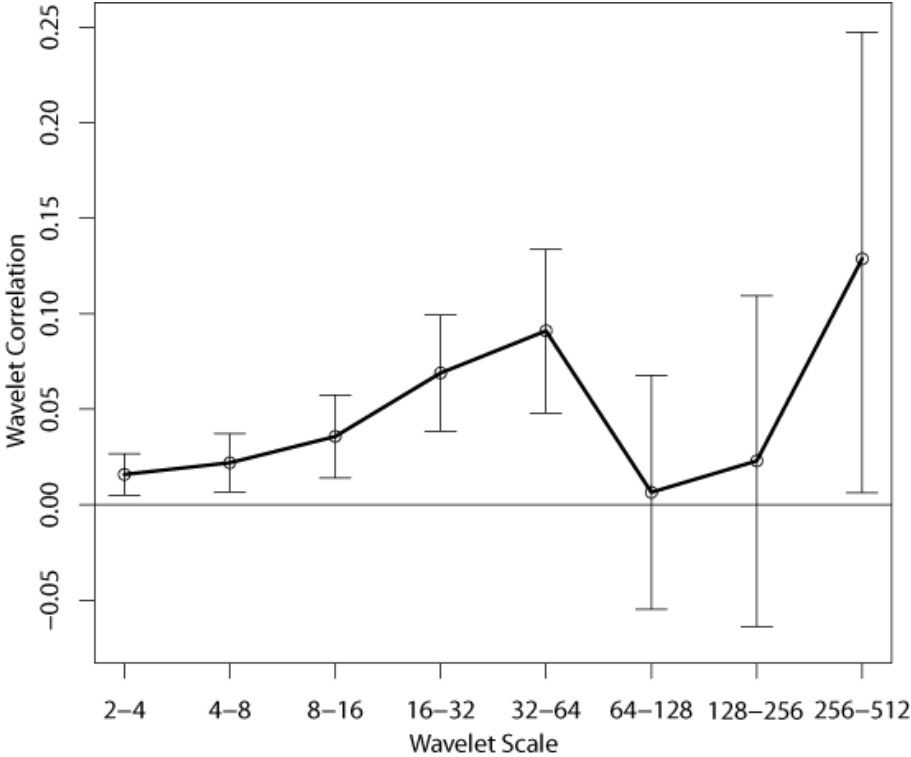

**Figure 5: Wavelet cross-correlation values between GPS-PWV and 95$^{th}$ percentile of the precipitation intensity from XPol radar time series as a function of the different wavelet scales.**





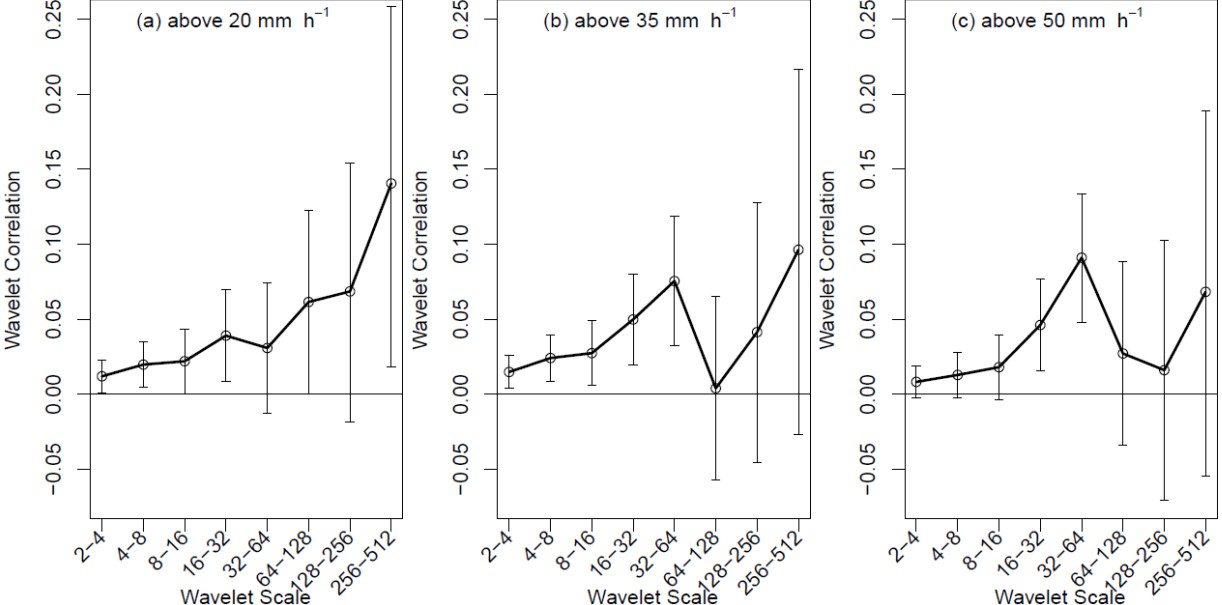

**Figure 6: Wavelet cross-correlation values between GPS-PWV and precipitation time series in different wavelet scales for (a) percentage of points above 20 mm h$^{-1}$ observed by radar around the GPS antenna; (b) the same above 35 mm h$^{-1}$; (c) and the same above 50 mm h$^{-1}$.**







**Figure 7: GPS-PWV jumps observed in the 2-hour period before a heavy storm occurred during DoY 341 (1836 UTC 7 December 2011).**





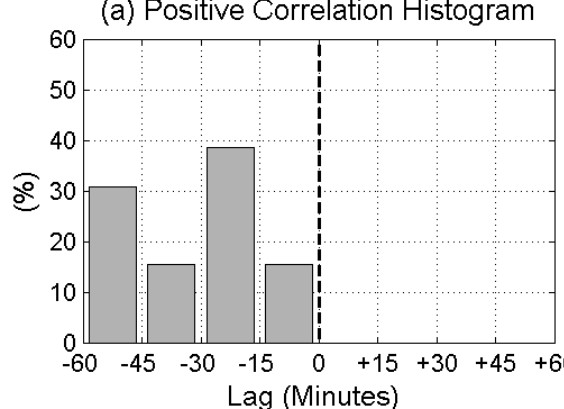
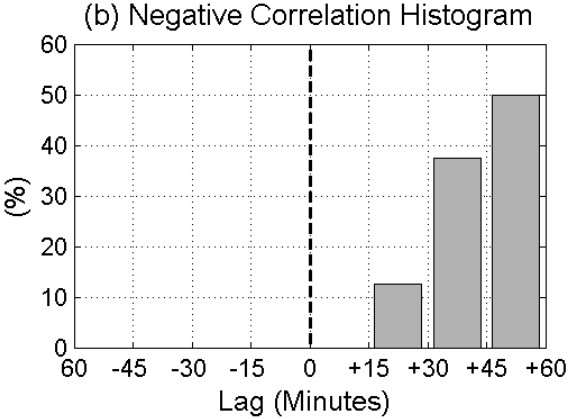

**Figure 8: Spearman correlation histograms of positive (a) and negative (b) correlations as functions of the lag of occurrence for GPS-PWV values and precipitation events. All 18 precipitation events listed in Table 1 were taken into account in this analysis.**





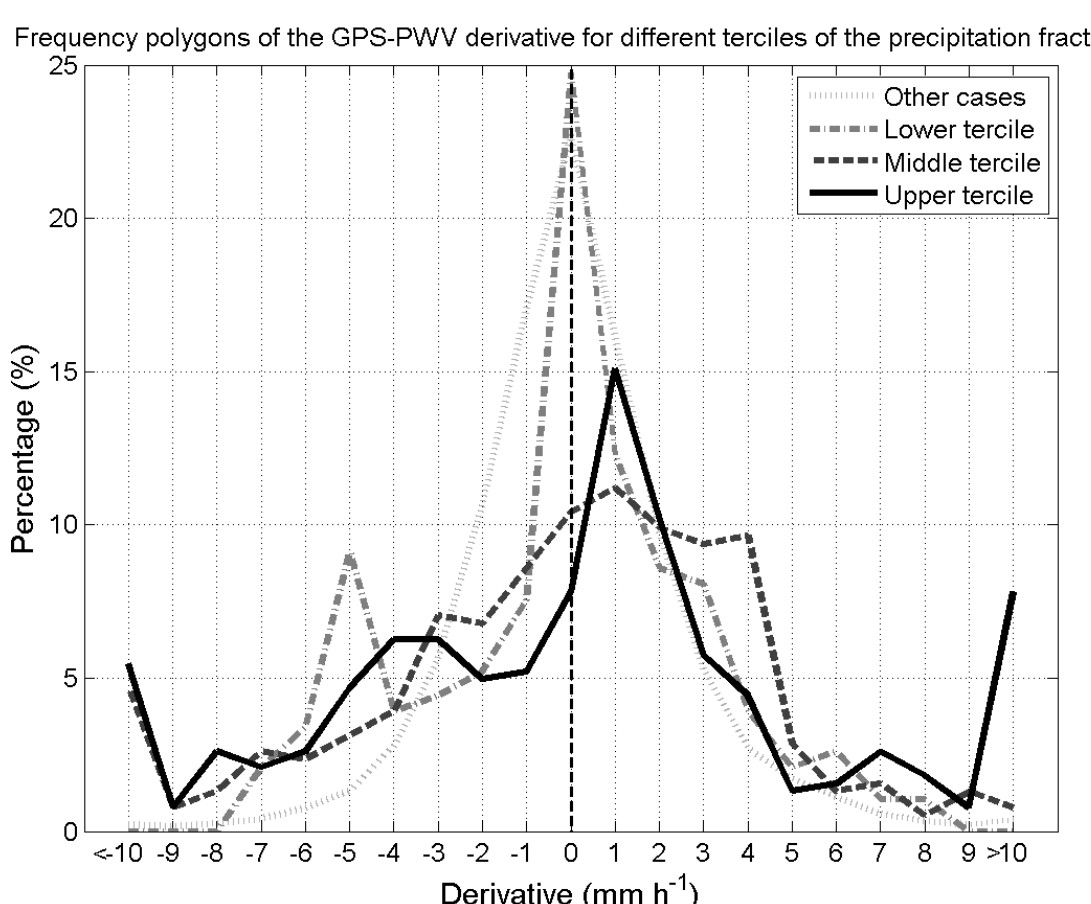

**Figure 9: Frequency polygons of the GPS-PWV derivatives calculated over the period of 60 minutes before precipitation events for different terciles of the precipitation fraction above 35 mm h[-1] observed using XPol radar.**

