# Peer review of "GPS-PWV jumps before intense rain events"

_Atmospheric Measurement Techniques, 2016_

## Referee Comment (RC1) · Anonymous Referee #1 · 28 Jan 2017

**January 2017**

The manuscript presents a combined analysis of time series of precipitable water (PW) derived from GPS (Global Positioning System) observations and surface precipitation estimated from an X-band Radar. Both datasets were obtained in 2011 during the CHUVA Vale measurement campaign. The study mainly focusses on a statistical analysis of the relationships between precipitable water fluctuations and precipitation intensity (with wavelets in particular). The results suggest a potential of high-frequency PW for nowcasting.

A major originality of the manuscript lies in the particularly high frequency of the PW dataset: one minute. I had never seen GPS time series of PW provided with such a small temporal sampling, and I think that this is very interesting. I also think that this specificity of the study is not emphasized enough, for instance the abstract does not mention the 1-min sampling of precipitable water.

My main comments for now concern the presentation of the results.

A) In my opinion, the authors jump too quickly to statistical results. The results will be much more convincing with the addition of the time series centred on rain events, because I think that they must first present, and in much more details, these results (see specific comments).

B) This goes with some reorganization of the manuscript, with section 4 placed before section 3.

C) The datasets, methods and results are not all clearly separated. It would have been easier for the reader if the datasets, data processing and methods (wavelets...) had been presented before the results. In addition, such a structure would have prevented repetitions. In case you choose to stick to the present organization (apart from the one indicated in B), please make sure to remove unnecessary repetitions (I noted some in the specific comments below).

**Specific comments**

1) Page 2, introduction, "PWV data from a microwave radiometer (MWR) with high temporal resolution have been used to describe the observed relationship between the PWV and precipitation in the tropics (Muller et al. 2009). Muller et al. (2009) do not use any microwave radiometer (MWR), they use a a purely model-based approach. On the other hand, Holloway and Neelin (2010) is a proper reference.

2) Page 2, Introduction, references to studies using GPS PW: several references are given, they mainly concern American areas. Similar studies were carried out by Bock and colleagues in Africa (e.g. Bock et al. J. Geophys. Res. 2008) and in Europe, and very likely by others elsewhere.

3) Page 2, Introduction, paragraph "The motivation ...": the authors should clearly state at this point that they will use very high frequency datasets.

4) Page 3, section 2, first paragraph: the three last sentences ("During the CHUVA...
and precipitation." are unnecessary.

5) Page 3, section 2: the same information is given twice,

lines 7,8: "The CHUVA Vale campaign was carried out in São José dos Campos"

line 16: "The CHUVA Vale experiment performed in São José dos Campos City"

Please correct, be more concise

6) Page 5, "The time series of the precipitation fractional area around the GPS receiver observed by radar were calculated by determining the position of the GPS antenna in the gridded precipitation points and taking into consideration the area formed by points in the longitudinal direction for the same number of points in the latitudinal direction where the nearest point of the GPS antenna is located in the center of these areas": this sentence is unclear.

7) Page 5, "representative" in "Different areas were tested, and an area of 22x22 (longitudinal per latitudinal direction grid point values of rainfall intensity (mm h -1 )) was found to be more representative of the observed area by GPS'. Can you precise what you mean with "representative" here? This is too vague.

8) Page 6, "The disdrometer time series has a good correlation with the 95 th percentile time series": how did you do precisely? The time samplings of these variables are 1 min and 6 min. Did you regrid the disdrometer time series, please precise. The same apply to the legend of Fig. 2.

9) Page 6, 1st paragraph of section 3, "As argued by Adams... in this region". These sentences are not so much about the methodology. They would be better placed in the introduction or discussion.

10) Page 7, section 3.1: I found that the paragraph was not always very clear.

For instance, the authors write "The methodology employed to process the GPS data in one-minute intervals did not provide any additional information." (line 6) Interactive comment

and a few lines below

"For this reason, it is necessary to take into consideration ... a short time step, e.g., the one-minute interval used in this study."

11) "expressive": this word is frequently used in the manuscript but I am not sure that is is appropriate (note that I am not a native English speaker).

12) Page 7, section 3.1, replace "A vertical line was put at the peak of the maximum precipitation in each event to simplify the analysis" by "For clarity, a vertical line was drawn for each precipitation maximum". This information is missing in the legend, where it would possibly be more appropriatly introduced.

13) Page 7, section 3.2: did you need to re-sample the GPS PW dataset on the 6 min time step of radar precipitation here? And if so, how? e.g. via sub-sampling, time averaging?

14) Section 3.5, Figures 5 and 6: correlations are not very high. You need to comment their magnitude. Also, the caption does not indicates the error bars not how they were drawn.

15) Page 8. You recall the same information several times. For instance, below, an information that was already given before.

lines 10-11: "...the results reported by Adams et al. (2013), who showed that the strongest water vapor convergence is typically  $\sim$ 1 hour before heavy precipitation."

Page 7, lines 23-24: "... the water vapor may increase through low-level moisture convergence, as suggested by Adams et al. (2013)on the time scale of 32-64 minutes"

Please reorganize and avoid repetitions.

- 16) Raymond (1987) is missing in the list of references.
- 17) Section 4, inset in Figure 7: this is the first and only time that the fluctuations of
high-frequency GPS PW are presented. I think that this is late in the manuscript, and that this is not enough. I think that you have to present several of the cases presented in table 1, in the form of graphs jointly showing time series of radar precipitation (either fraction above 50 mm/h or another rainfall diagnostic) and GPS PW. I would prefer to see the 18 cases, organized by terciles, or at leat 3 of them per tercile.

I further strongly suggest to move section 4 before section 3 (see main comment B).

References

Holloway, C. and J. Neelin, 2010: Temporal Relations of Column Water Vapor and Tropical Precipitation. J. Atmos. Sci., 67, 1091–1105, doi: 10.1175/2009JAS3284.1.

Bock, O., et al., 2008: The West African Monsoon observed with groundbased GPS receivers during AMMA, J. Geophys. Res., 113 (D21105) doi : 10.1029/2008JD010327.

---

## Referee Comment (RC2) · Anonymous Referee #2 · 14 Feb 2017

General comments: The manuscript presents the behavior of GPS-PWV time series during severe precipitation events recorded by an X-band Radar during the CHUVA Vale measurement campaign in 2011. GPS-PWV jumps have been detected between 32 and 64 minutes before the more intense rainfall events. The statistical characterization of this phenomenon has potential for nowcasting. The reasoning of the paper is based on a wavelet analysis of the GPS-PWV times series and on GPS-PWV derivative analysis and distinguishes meteorological events into 3 classes of precipitation. The main value of the article is to focus on the behavior of PWV during intense weather events, expecting to improve their forecasting.

However, the article contains a number of imprecisions that are important to dissipate : too many repetitions, a presentation that deserves to be more rigorous, more structured, more synthetic on the basic methodology and more explicit on the work done: 1. The presentation of GPS data processing is only too partial and often confusing.

2. The use of GPS-PPP times series sampled at 1-min intervals is interesting but the manuscript did not present well which specific PPP products were used to get it. 3. The use of wavelet analysis is interesting only for part "3.2 Wavelet cross-correlation analysis" even if shown correlations seem to be weak (figures 5 & 6). Part 3.1 "Wavelet power spectrum analysis" is the effect of the GPS-PWV jumps, a well know wavelet power spectrum of a Dirac function. Part 3 and 4 should be merged : part 3.1 for presenting the GPS-PWV jumps and part 3.2 for presenting time lag correlation. 4. Part 4 is really interesting and could be more developed. However, the criterion on GPS-PWV derivative >  $+9.5 \text{ mm.h}^{-}(-1) \text{ and }

to the integral of the refractivity index of the air as a function of temperature, pressure and water vapor (Bevis et al. 1992) on the optical path followed by the GNSS signal.

3) Explanation line 25-27 is too expeditious. - lines 25-26: "with an error of approximately 5% under all weather conditions (Wolfe and Gutman 2000)" The given reference is outdated using old version of GPS data processing software, relative calibrations of antennas etc. The evaluation of the accuracy of GPS-PWV at around 5% should also be given in millimeter. The accuracy of GPS-PWV estimates remains a active topic of research, especially during severe weather conditions when all other meteorological instruments are down. - lines 26-27 "and in near real time (Rocken et al. 1994)." Outdated reference that could be used to put into perspective the improvements made since. It could be very interesting to emphasize on the methodological improvements made from 2000 and the first utilization of the GPS-PWV estimate until now. (Guerova et al. , 2016)  $\rightarrow$  Guerova G, Jones J, Dousa J, Dick G, De Haan S, Pottiaux E, Bock O, Pacione R, Elgered G, Vedel H, Bender M (2016) Review of the state-of-the-art and future prospects of the ground-based GNSS meteorology in Europe. Atmos Meas Tech Discuss. doi:10.5194/amt-2016-125

4) line 31: "high temporal resolution (~minutes)." I know it is not easy to present it but this formulation hides many methodological points about the methodology of GPS data processing: zero, single, double difference analysis or PPP, the different ways to model ZWD in analysis... The easiest way to get high temporal resolution (~minutes) on GPS-PWV estimates is the PPP strategy (Zumberge et al, 1997)  $\rightarrow$  Zumberge, J. F., M. B. Heflin, D. C. Jefferson, M. M. Watkins, and F. H. Webb (1997), Precise point positioning for the efficient and robust analysis of GPS data from large networks, J. Geophys. Res., 102(B3), 5005–5017, doi:10.1029/96JB03860.

5) Page 1 Line 31  $\rightarrow$  page 2 line 1: "Other promising applications become viable in dense networks and transects": using dense networks to do tomography is not a "promising application" even if it remains methodological issues (e. g. Champollion et al. (2005); Bastin et al. (2005); Brenot et al. (2014) )  $\rightarrow$  Bastin, S., C.
Champollion, O. Bock, P. Drobinski, and F. Masson (2005), On the use of GPS tomography to investigate water vapor variability during a Mistral/sea breeze event in southeastern France, Geophys. Res. Lett., 32, L05808, doi:10.1029/2004GL021907.  $\rightarrow$  Brenot, H., Walpersdorf, A., Reverdy, M., van Baelen, J., Ducrocq, V., Champollion, C., Masson, F., Doerflinger, E., Collard, P., and Giroux, P.: A GPS network for tropospheric tomography in the framework of the Mediterranean hydrometeorological observatory Cévennes-Vivarais (southeastern France), Atmos. Meas. Tech., 7, 553-578, doi:10.5194/amt-7-553-2014, 2014.  $\rightarrow$  C. Champollion, F. Masson, M.-N. Bouin, A. Walpersdorf, E. Doerflinger, O. Bock, J. Van Baelen, GPS water vapour tomography: preliminary results from the ESCOMPTE field experiment, Atmospheric Research, Volume 74, Issues 1–4, March 2005, Pages 253-274, ISSN 0169-8095, http://dx.doi.org/10.1016/j.atmosres.2004.04.003.

Page 2:

6) line 3: "the diurnal cycle": more discussion about meteorological processes who have been detected according to areas? (For West African Monsoon, Bock et al. (2007))  $\rightarrow$  Bock, O., F. Guichard, S. Janicot, J. P. Lafore, M.-N. Bouin, and B. Sultan (2007), Multiscale analysis of precipitable water vapor over Africa from GPS data and ECMWF analyses, Geophys. Res. Lett., 34, L09705, doi:10.1029/2006GL028039.

7) About §2 (lines 5-12) and §3 (lines 13-22): Is it possible to merge §2 and §3 to emphasize on links between PWV, deep convective activity and the occurrence of intense rainfall?

8) line 7: "PWV data from a microwave radiometer (MWR) with high temporal resolution" Provide an order of magnitude.

9) Line 8: Wrong reference with Muller et al. (2009) who do not used MWR-PWV data. However, it remains interesting to provide a reference in the article about relationship between the PWV and tropical precipitation.

AMTD
10) Line 8: Using "Chan 2009" can be useful to discuss about differences between MWR-PWV and GPS-PWV during severe weather events (see in particular the interesting §3.3 Comparison with GPS receivers of the article).

11) Line 9: "useful indications of the accumulation of water vapor" : be more specific, in which these indications are useful ?

12) Line 14: "Mazany et al. (2002) developed a lightning prediction index for Florida based on the GPS-PWV magnitude and its temporal evolution": if this index is interesting, can you write more about it?

13) Line 19: "This study showed that prior to deep convective events in the central Amazon, a 4-hour "ramp up" in the time derivative of GPS-PWV is observed, reaching a maximum approximately one hour before heavy precipitation."ÂăThis sentence should be in the next paragraph to better distinguish results from precedent studies and results of the article. It is a repetition of sentence line 26-27 "The sharp increase in the GPS-PWV values approximately one hour before the occurrence of more intense rainfall events, as found in this study and that of Adams et al. (2013)" and should be merged with it.

**Page 3**

14) line 8: "The CHUVA Vale campaign was carried out in São José dos Campos in São Paulo State in an elevated valley between the Mantiqueira and Serra do Mar mountain ranges." Add the reference of fig.1 given line 23: "Fig. 1 shows the geographic location of the CHUVA Vale campaign, emphasizing the sites at which the instruments were placed."

15) line 11: "During the CHUVA campaigns," Too many repetitions.

16) Lines 11-14: "GPS meteorology was used to monitor the horizontal and temporal variations in the PWV associated with the wide variety of convection-producing mechanisms for the 6 geographic regions. For example, Adams et al. (2015) described the

AMTD
temporal and spatial evolution of tropical sea-breeze convection with GPS meteorological transects during the CHUVA-Belem field campaign." Of topics in this part : be more concise. You can eventually add this point in your introduction part. You speak about " 6 geographic regions" without using it after.

17) line 23: "Fig. 1 shows the geographic location of the CHUVA Vale campaign, emphasizing the sites at which the instruments were placed." See comments about line 11.

**Page 4**

18) subsection "2.2 High temporal resolution GPS-PWV time series" is too confusing and must be structured and clarified.: - GPS data of receiver sampled at one-second frequencies  $\rightarrow$  under-sampling ? - Orbit and clock data products : what kind of products did you use ? What is the sampling of these products ? Orbit and clock data products from JPL for PPP applications are sampled at 5 minutes (https://gipsy-oasis.jpl.nasa.gov/index.php?page=data). This point determines the rest : I would like to be sure that the final sampling rate of the GPS-PWV values is really 1 minute and not 5 minutes, as usual. If you made a specific processing, you have to present it. In addition, you have to ensure that this kind of estimation is appropriated for your purposes and your estimation doesn't suffer the effect of an artifact. - elevation weight function used for GPS observations (constant ?, elevation dependent (a/sin(elev)^2) ?) - Cut-off (OK line 5) - Tropospheric models used : mapping function (OK line 5), ZHD a priori (Is it GPT ?), ZWD time evolution constraint as a random walk ? Tropospheric gradients have been estimated ?

19) Line 6: "To ensure the quality of the PWV time series with high temporal resolution required in this study, a rigorous data-processing strategy was adopted with possible noise sources taken into consideration." What does it mean? Give a reference if you used a validated data-processing strategy else explicit it please.

20) Line 8: " recommended by the International Global Navigation Satel-
lite System Service" : use IERS conventions that are authoritative in the field.  $\rightarrow$  IERS Conventions (2010). Gérard Petit and Brian Luzum (eds.). (IERS Technical Note ; 36) Frankfurt am Main: Verlag des Bundesamts für Kartographie und Geodäsie, 2010. 179 pp., ISBN 3-89888-989-6 https://www.iers.org/IERS/EN/DataProducts/Conventions/conventions.html

21) lines 9-10: "absolute calibration was performed to ensure the correct phase center variation, as reported by Görres (2006)" : It would be clearer to distinguish ÂńÂăabsolute calibration (Schmidt et al., 2009)ÂăÂż and the specific absolute calibration of your antenna (Görres, 2006) if you have done it. → Schmid, R., P. Steigenberger, G. Gendt, M. Ge and M. Rothacher (2007), Generation of a consistent absolute phase-center correction model for GPS receiver and satellite antennas, Journal of Geodesy, Volume 81, Number 12, 781-798, doi :10.1007/s00190-007-0148-y.

22) Line 10: New paragraph to explain the conversion  $ZTD \rightarrow PWV$ ?

Line 13-14: Which TM and Pressure data have been used, and with which time resolution? How are they computed at 1 minute sampling?

23) Line 15: "The sampling rate of the GPS-PWV values was 1 minute." Again, if the sampling rate is 1 minute, that implies PPP products cannot be sampled at 5 minutes. Have you done a specific GPS data processing to compute your own PPP products to obtain a sampling rate of the GPS-PWV values at 1 minute?

24) Line 16: "problems with the satellite ephemerides" Can you explicit these problems? I don't understand why you got a problem with it.

25) Lines 16-17: "unavailable pressure measurements" Is it possible to complete it with a meteorological modelÂăor the problem was not so important to solve it?

26) Lines 17: "other unknown causes": If we consider GPS data from receiver, orbit/clock products for PPP processing and pressure measurements, I don't see which other unknown causes can be possible.
Subsection "2.3 Precipitation time series from disdrometer and XPol radar data" should be clarified:

27) Line 23: "very small spatial scale": Provide an order of magnitude.

**Page 5Âă**

28) line 5-6 "The dimensions of the precipitation area that influences the GPS-PWV is a key factor in the development of this study": If it is the case, you should explicit what you have really done and not summarize quickly your tests to directly provide the area of 22x22 1. "Different areas were tested": explicit 2. "found to be more representative of the observed area by GPS": what criteria have been used? 3. "better for exploring the correlation between the precipitation occurrence and GPS-PWV": Could you please provide quantified results?

29) Line 14: "around" Can you specify?

30) Line 22-24: "the statistical measurements calculated from the radar data were in the 95th percentile of the intensity of the precipitation observed in the area of 4.4 km per 4.4 km around the GPS antenna" OK it is a statistical way in order to examine the intensity of the precipitation.

Figure 2:Âă

1 Lack of data DOY 319Âăfrom Xpol radar?

2 b and c should be at the same scale.

3 On 3 rainfall events, precipitation intensities are above 125 mm/h according to Disdrometer (Fig. 2 b) whereas the 95th percentile of the intensity of the precipitation observed by Xpol radar radar are below 100 mm/h (Fig. 2c): It seems strange that the disdrometer measures a statistical anomaly 3 times on 20 (>95th).

page 6
"3 Wavelet analysis"

31) Lines 15-16: "In this study, both continuous and discrete wavelets are investigated to achieve intra- and interrelation analysis, respectively." Distinguish what wavelet decomposition should be used to answer to what scientific questions.

**Page 7**

32) Lines 6-7: "The methodology employed to process the GPS data in one-minute intervals did not provide any additional information. Fig. 3 shows that the GPS-PWV energy variability begins to be significant only for time scales longer than 16 minutes." The influence of stochastic constrains applied on temporal evolution of ZWD during the GPS data processing must be taken into account. If you have used a too small random walk parameter, that could explain what you have observed.

33) line 8: "Therefore, the one-minute time series representativeness is not a limitation, and if there is noise, it is white noise." Can you explicit and prove it?

34) It is obvious that a jump in a time series will produce what is described lines 13-15: "there are expressive changes in the power between different time scales in those cases in which an increase in the power of the oscillation from low to high frequency is observed." It is the well known example of the Wavelet power spectrum of a Dirac signal.

35) Figure 3: GPS-PWV presents a jump DOY 358 with a strong signal in the Wavelet Power Spectrum but Xpol radar did not detect any precipitation : have you any comment on it?

36) lines 31-32: "The results show that the wavelet correlation between the PWV and precipitation intensity is more evident and significant for the time scale between 32 and 64 minutes". Again it would be clearer to speak about PWV jump before speaking about the lag between GPS-PWV jump and rainfall and introduce wavelet to determine the lag precisely.
37) lines 31-32: Correlations shown figure 5 do not exceed 0.15 and do not look significant: it seems clear there is a lag between GPS-PWV and rainfall and the evaluation of this lag seems good but the correlations of figures 5 and 6 diminish the strength of the demonstration.

**page 8**

38) "4 Behavior of PWV time series before precipitation events: the GPS-PWV jumps": I appreciate the meteorological interpretation of GPS-PWV jump during severe weather events but this interpretation seems to be founded only on a single reference (Adams et al., 2013): Is there any other references on these meteorological processes during severe weather events?

**Page 10**

39) line 3-5 "This result corroborates the pattern observed in Fig. 7, showing the GPS-PWV maximum before the precipitation event and its minimum after the maximum precipitation" It would be clearer if the zoom of figure 7 has shown precipitation.

40) "4.2 GPS-PWV derivative analysis" Line 20-21: "Fig. 9 clearly shows an expressive change in the pattern of the derivative distribution as a function of the different precipitation intensity terciles." : I suppose you did it but did you check that for each severe weather event of upper tercile you got around 7.8% of GPS-PWV derivative >  $+9.5 \text{ mm.h}^{-}(-1)$  and around 5.47% of GPS-PWV derivative

there is a jump in GPS-PWV time series during severe weather events (see comment 34).

---

## Referee Comment (RC3) · Anonymous Referee #3 · 18 Feb 2017

Review of manuscript amt-2016-378: "GPS-PWV jumps before intense rain events" by Luiz F. Sapucci and co-authors.

This manuscript presents an analysis of time series of PWV from a GPS station and of precipitation from a nearby X-band radar for a period of 56 days in Brazil. The characteristics of PWV variations before and after the peak in precipitation rate are described with wavelet analysis and more classical tools (spearman correlation coefficient and histograms). The discussion is focused on the rapid variation of PWV preceding the peak in precipitation (increase followed by decrease). Though this feature is quite well known, both for the Amazon and for other regions of the world, the main novelty I see from this work is the fact that the increasing phase in PWV can be built upon successive pulses as suggested by Fig. 7. This subtle modulation of PWV was detected in this study thanks to the high temporal resolution of the PWV time series (1-minute). The potential for predicting the occurrence of intense precipitation with the help of real-time GPS PWV data is also suggested but not demonstrated. The topic of this study is interesting and might contribute to a better understanding of the moist processes involved in deep convection over land in the tropics (though such physical interpretation is beyond the scope of this paper). However, in its present form, this manuscript cannot be accepted for publication. A major rework of the data analysis and presentation is required. Major and specific comments are given below.

A) Major comments

1) Wavelet analysis:

- The wavelet correlation method and results are not properly used and interpreted. Several times the authors interpret the period bands for which the wavelet correlation values are displayed as lag times (P2L29, P620, P6L29, P9L19). For example (P6L29): "an interrelation analysis can be performed with the WCC of GPS-PWV and precipitation time series to evaluate the correlation at different lags in time, scale by scale". The results provided in figure 5 and 6 are correlation values for a range of time periods indicated in the x-axis and wrongly labelled "wavelet scale". This kind of plot is typically given for a zero-lag (see e.g., Whitcher et al., 2000). Where are the wavelet correlations for different lags? The analysis could be complemented with wavelet cross-correlations as a function of lag for different scales such as in Whitcher et al., 2000. Proper use of wavelet cross-correlations for lead/lag analysis would probably not require using the classical Spearman correlation (Fig. 8).

- The significance of wavelet spectra (Fig. 3 and 4) and correlations (Fig. 5 and 6) is not addressed, though mention to "significant" frequencies is made in several places in the text. Most tools used for wavelet analysis provide also results of significance tests or confidence intervals. These should be used to highlight objectively the significant frequencies.

- It is not said what the error bars in Fig. 5 and 6 represent.

- The description of wavelet analysis methodology in section 3 is not sound. Only fragmented information is given on the used methods, without justification of the choices (e.g. of mother wavelets, continuous vs. discrete wavelet transforms). On the other hand very highly-technical information is given (e.g. use of non-decimated discrete wavelet transform, use of pyramidal algorithm…) which won't be explicit to the general readers of AMT journal. So I recommend that this section explains a bit more about the general principles and choices that were made, and the

implication of the choices that are made. For example, one might question about the robustness of the results regarding the choice of mother wavelets. For the more technical aspects, refer to the proper literature where appropriate.

2) Analysis of data and interpretation of results

The GPS data processing options should be more detailed (see specific comments below). Some fundamental aspects are missing, such as the constraints of the stochastic model of the ZTD parameters. They impact directly the magnitude of variability in the retrieved PWV time series.

The dimensions of the area around the GPS station used to study the PWV-precipitation relationship is said to be a "key factor" (P5L5). In consequence, some preliminary tests should be presented which give insight into the sensitivity of results to this parameter. It is not clear how the representativeness (P5L11) is estimated.

Fig. 7 shows one case of heavy precipitation where PWV shows a strong peak achieved in several steps (so-called "pulses" in the text and inconsistently labelled "jumps" in the Figure). This kind of analysis should be made for other cases to establish if the pulses are a robust feature of the heavy precipitation events. It is not clear if the authors associate the 32-64 min periodicities detected by the WCC with these pulses? It would also be interesting to add a composite of time series to highlight the most prominent features of the PWV jumps.

The authors refer extensively to Adams et al. (2013) regarding a maximum of PWV 1-h before the maximum of precipitation. Inspecting this reference carefully reveals that on average the peak in PWV is rather coincident with the peak in precipitation (Fig. 2 in Adams et al., 2013) or slightly ahead (Fig. 3 and 4), in accordance with other results (e.g. Holloway and Neelin, 2010). The case illustrated in Fig. 7 shows indeed a lead time of 1 hour but about 50% of the cases show lead times between 0 and 30 min (Fig. 8). So the 1-h lead time should not be considered as a general rule.

In section 4.1, show an example of correlation function to help explaining what is meant by the "positive and negative correlations" (in fact maximum positive/minimum negative correlations) and specify the time window of analysis (+/- 1 hour). Again, the histograms in Fig. 8 suggest that the case in Fig. 7 might be a specific case because the minimum correlation is probably reached for a lag time around 0. Illustrating other cases and providing composite plots would help to better catch the general situation. Moreover, Fig. 7 is a case where the pulses preceding the peak in PWV are outside of the correlation window. The link between the pulses and the peak in PWV and in precipitation should be investigated in more detail as this is to my opinion the real innovation of this work.

In section 4.2, the PWV derivatives are computed in a 1-hour window preceding the peak in precipitation. This window is too small, namely for the case of Fig. 7, as it does not include the part of time series with the pulses. Extend the time window to analyse this feature in more detail and also to not give too much weight to the decreasing phase after the peak in the statistics computed in Table 2. Some sensitivity tests should be made to choose the best window size.

I suggest to add also a figure with the time series of PWV derivatives for all 6 cases of the upper tercile (e.g. superposed on one graph showing also the average PWV derivative and precipitation).

The sentence P2L29 announces a "study of correlation and lags with rainfall events to form a conceptual model with predictive capacity that can hence be used in developing a nowcasting tool for strong precipitation events". However, section 4.3 doesn't provide any proof of concept of this model. I suggest a short hindcast study is performed, based on the data from this campaign, to evaluate the skill of the proposed detection method.

Some interpretation is made about the underlying physical processes leading to the observed PWV jumps:

P8L25-29: "The presence of low-level water vapor convergence can be attributed to mechanisms such as gravity wave forcing (Raymond 1987) or other larger-scale forcing mechanisms. The increased water vapor convergence (i.e., positive PWV derivatives) may also simply be a reflection of the unstable surface parcels accelerating upwards, thereby vertically advecting a larger surface specific humidity to higher levels in the atmosphere without necessarily any larger-scale dynamical forcing"

P10L31: "This increase in the stronger negative derivative frequency before more intense events (upper and middle terciles) is associated with the conversion process from water vapor to liquid water…"

P11L5: 'the processes responsible for the maintenance of precipitable water suspended in the atmosphere are very complex and highly nonlinear"

P11L8: "The increase in moisture convergence appears to be due to an increase in the frequency of convergence pulses."

These assertions relate either to very general atmospheric processes or are highly speculative as recognized by the authors, P8L29, "Given the limitations of the observations, our interpretation of the physical mechanisms responsible for the jumps remains speculative."

I suggest to remove sentences which cannot be supported by other studies of similar phenomena or provide the necessary proofs, e.g. by analysing additional data.

3) Inappropriate and vague terminology

The PWV jumps which are the main topic of this work (as advertised in the title) are defined inconsistently in several places in the manuscript: P2L28 ("sharp increase in PWV"), P8L22 (a "pattern" of maximal PWV), P9L14 ("oscillations"), and Fig. 7 (pulses defined P8L25 are labelled "jumps"). Please be consistent.

Precipitation intensity is measured as the $95^{th}$ percentile of all precipitation data in the area (P5L22). However, later, precipitation fractions are referred to as precipitation intensity (Table 1 and 2). Intensity terciles are referred to in section 4.2 and 5. Please be consistent.

It is not clear if the precipitation fractions are computed from the $95^{th}$ percentile of precipitation or from all data and grid points?

Expressions such as "expressive", "expressive changes", "evident", "more evident" are not adequate. Please use conventional scientific terminology.

What is a positive oscillation? (P8L18, P12L3…) or a positive increase?

B) Specific comments

P4L7: specify which "possible noise sources" are taken into account and how.

I understand that you used the regional model for Tm of Sapucci 2014. So it is not the Bevis 1992 relationship that is used (P4L13). Please correct.

This Tm model requires RH. It is not said what RH data are used to compute Tm.

P4L17: "3.1% of data are missing in 2 days". This represents about 1.7 days for the total period of 56 days. I suggest that you completely remove these 2 days from the analysis because the spline interpolation won't give correct results.

It is important to specify the tropospheric model used in the GPS data processing: is it a random walk? Are both ZWD and gradients estimated? What are the constraints of temporal evolution of these two parameters? Did you make some tests with different constraints? (the analysis of 6-min time differences in section 4.2 might be strongly impacted if the constraints parameters are too small or too big).

P6L8-10 and in many other places, results from Adams et al., 2013, should be rather described in the introduction.

P7L17: what do you mean by "amplification"?

P7L31: why is quasi-symmetry ("a good property for the mother wavelet") important to this study?

P8L19:20: "maximum peak" replace with "maximum"

P10L12: "the lowest threshold" complete with "among the 3 that were tested".

P10L16: what are rainfall events with periods without precipitation?

P11L11-19: description of results from Fig. 9 should go to section 4.2

P11L21-25: results from past studies about nowcasting should go to the introduction. Note that these references can probably be updated.

---

## Author Comment (AC1) · 25 Mar 2017

**Response latter for Referee1**

**General comments from Referee#1**

The manuscript presents a combined analysis of time series of precipitable water (PW) derived from GPS (Global Positioning System) observations and surface precipitation estimated from an X-band Radar. Both datasets were obtained in 2011 during the CHUVA Vale measurement campaign. The study mainly focusses on a statistical analysis of the relationships between precipitable water fluctuations and precipitation inten-

sity (with wavelets in particular). The results suggest a potential of high-frequency PW for nowcasting.

A major originality of the manuscript lies in the particularly high frequency of the PW dataset: one minute. I had never seen GPS time series of PW provided with such a small temporal sampling, and I think that this is very interesting. I also think that this specificity of the study is not emphasized enough, for instance the abstract does not mention the 1-min sampling of precipitable water.

My main comments for now concern the presentation of the results

A) In my opinion, the authors jump too quickly to statistical results. The results will be much more convincing with the addition of the time series centred on rain events, because I think that they must first present, and in much more details, these results (see specific comments).

B) This goes with some reorganization of the manuscript, with section 4 placed before section 3.

C) The datasets, methods and results are not all clearly separated. It would have been easier for the reader if the datasets, data processing and methods (wavelets...) had been presented before the results. In addition, such a structure would have prevented repetitions. In case you choose to stick to the present organization (apart from the one indicated in B), please make sure to remove unnecessary repetitions (I noted some in the specific comments below).

**Response:** All the comments and suggestion presented by referee were taken into consideration in this revised version of the manuscript, which generated strong and important improvement in the presentation of the ideas associated with this conceptual study and obtained results. It was necessary a substantial restructure of the paper and more figures with additional results had to be produced and they are included. About the general comments:

- It is true that PWV-GPS with high temporal resolution has not been explored for now-casting activities, we agree that this aspect was not emphasized enough in previous version of the manuscript, which was corrected in the new one. Information about the high temporal resolution in GPS-PWV time series was included in abstract and the objective of this work, it was suitably emphasized. Motivated by this suggestion and comments from other referee, in the revised version of the manuscript we have better explored the results to study rapid oscillations of PWV time series (time scale of 8-16 and 16-32) before precipitation, which we associated with PWV jumps. This was done using Wavelet Cross-correlation analysis with lead-lag (new figure included) in the section 4, now titled "High temporal resolution GPS-PWV time series analysis".

- The discussion about the GPS-PWV Jump was moved to before of the statistical analysis of the GPS-PWV time series, as suggested. We agree that the GPS-PWV jumps it the most important subject of this work and deserve special treatment. In this new strategy used to present the PWV-Jump, many parts of manuscript had to be changed, such as: abstract, introduction, the new section 3 dedicated to discuss the GPS-PWV jumps. The other analyzes, such as wavelet and time-lag, are presented in new section 4. The methodology applied wavelet analysis was better detailed, as suggested by other referee. We moved it to introduction of the sub-section 4.1.

- Two new figures was included in the manuscript. The new Fig. 4 shows other GPS-PWV jumps observed before different extension rain events, one of each tercile. GPS-PWV time series before of the 18 events were efficiently organized in the composite presented in the new Fig.5.

Following the recommendations presented in the specific comments, several parts of the manuscript were rewritten, correcting repetitions, some not appropriated terms were excluded and several phrases were rewritten. The two recommended paper were included in the bibliographic revision. Some information about the comparison of time series were clarified and information about the statistical significance of the results presented are discussed. Specific comments point-by-point are presented below. The

numbers (in the new version of the manuscript) of the changed lines and respective page in each point are listed and highlighted in green. The numbers of figure in the response refer to 13 figures of the new version of the manuscript.

**Specific comments**

1) **Referee#1:** Page 2, introduction, "PWV data from a microwave radiometer (MWR) with high temporal resolution have been used to describe the observed relationship between the PWV and precipitation in the tropics (Muller et al. 2009). Muller et al. (2009) do not use any microwave radiometer (MWR), they use a a purely model-based approach. On the other hand, Holloway and Neelin (2010) is a proper reference.
P2L17: Response : This phrase was rewritten, the mistake about the MRW-PWV data was removed, and the reference was maintained to provide a reference about relationship between the PWV and tropical precipitation, as suggested by reviser 2. The information about the MWR data applied to analyse and better understand the temporal relations of column water vapor and tropical precipitation was referenced by Holloway and Neelin (2010), as suggested.

2) **Referee#1:** Page 2, Introduction, references to studies using GPS PW: several references are given, they mainly concern American areas. Similar studies were carried out by Bock and colleagues in Africa (e.g. Bock et al. J. Geophys. Res. 2008) and in Europe, and very likely by others elsewhere.
P2L33 Response: We agree that the mentioned reference (Bock et al. 2008) has an important contribution for the evolution of this research and was included in the manuscript, as suggested.

3) **Referee#1:** Page 2, Introduction, paragraph "The motivation ...": the authors should clearly state at this point that they will use very high frequency datasets.
**P3L10 Response:** This phrase was changed. The information about the high frequency of PWV-GPS data used in this work was evidenced, as suggested.

4) **Referee#1:** Page 3, section 2, first paragraph: the three last sentences ("During the CHUVA... and precipitation." are unnecessary.
**P3L27 Response:** We agree that these phrases can be excluded from manuscript, because the information presented are not relevant by the work's focus.

5) **Referee#1:** Page 3, section 2: the same information is given twice, lines 7,8: "The CHUVA Vale campaign was carried out in São José dos Campos" line 16: "The CHUVA Vale experiment performed in São José dos Campos City" Please correct, be more concise
**P4L02 Response:** Both phrases were rewritten to make the manuscript more concise. The localization (with geographic coordinates) of the CHUVA Vale experiment was organized in the first phrase and excluded from the second one.

6) **Referee#1:** Page 5, "The time series of the precipitation fractional area around the GPS receiver observed by radar were calculated by determining the position of the GPS antenna in the gridded precipitation points and taking into consideration the area formed by points in the longitudinal direction for the same number of points in the latitudinal direction where the nearest point of the GPS antenna is located in the center of these areas": this sentence is unclear.
**P5L28-P5L31 Response:** This sentence was rewritten in more direct way to be clearer.

7) **Referee#1:** Page 5, "representative" in "Different areas were tested, and an area of 22x22 (longitudinal per latitudinal direction grid point values of rainfall intensity (mm h -1 )) was found to be more representative of the observed area by GPS'. Can you precise what you mean with "representative" here? This is too vague.

**P6L6-P6L8 Response:** The term "representative" in this phrase we mean that the area of precipitation observed by Radar more adequate to associate with observed area by GPS. This phrase was rewritten to make clear the idea and more information about the tested areas were included.

8) **Referee#1:** Page 6, "The disdrometer time series has a good correlation with the 95 th percentile time series": how did you do precisely? The time samplings of these variables are 1 min and 6 min. Did you regrid the disdrometer time series, please precise. The same apply to the legend of Fig. 2.

**P6L29-P6L32 Response:** In fact, the term "correlation" is not appropriated metric in this case. This sentence was totally rewritten in function of your comment and one other from reviser 2 (specific comment number 30). The new version of this phrase is: "The figure show that the disdrometer time series is consistent with the 95th percentile time series, although differences are expected due to different areas covered by each instrument and besides total precipitation from disdrometer is always larger than the one measured by radar and raingauge, due to problems with the large droplet concentration (Giangrande et al, 2016)."

9) **Referee#1:** Page 6, 1st paragraph of section 3, "As argued by Adams... in this region". These sentences are not so much about the methodology. They would be better placed in the introduction or discussion.

**P9L28 to P3L12 Response:** The reviser is right. The first phrase was changed to introduction section and the second was rewritten.

10) **Referee#1:** Page 7, section 3.1: I found that the paragraph was not always very clear. For instance, the authors write "The methodology employed to process the GPS data in one-minute intervals did not provide any additional information." (line 6) and a few lines below "For this reason, it is necessary to take into consideration ... a short time step, e.g., the one-minute interval used in this study."

**P10L15 P10L17 Response:** This first mentioned phrase was excluded from manuscript because was misunderstanding of the results showed by color scale used in the wavelet power spectrum. This color scale was selected to make clear the information in this plot. Other scales were tested, but the presentation of the results were not so good. Information about the color scale used were included. The second mentioned phrases were separated in different paragraph, which now treat of the difference between temporal frequency of rainfall events and PWV-GPS jumps and the consequent difficulties in the results analyses.

11) **Referee#1:** "expressive": this word is frequently used in the manuscript but I am not sure that is is appropriate (note that I am not a native English speaker).

**Response:** We agree that this term can generate misunderstanding and it was replaced by other more appropriated. All six occurrences in the text was rewritten and the term "expressive" was avoided. P7L14 "more expressive" was changed to "stronger"

P10L22 changed to "strong"

P10L28 "more expressive" was changed to "stronger"

P11L27 "an expressive" the phrase was excluded

P13L22 "the differences... are not expressive" was changed to " the differences are small between..."

P15L08 "expressive" was changed to "strong"

12) **Referee#1:** Page 7, section 3.1, replace "A vertical line was put at the peak of the maximum precipitation in each event to simplify the analysis" by "For clarity, a vertical line was drawn for each precipitation maximum". This information is missing in the legend, where it would possibly be more appropriatly introduced.

**P10L27 Fig.7 Response:** The phrase was replaced as suggested. In the caption of the Fig. 7 this information was included also.

13) **Referee#1:** Page 7, section 3.2: did you need to re-sample the GPS PW dataset on the 6 min time step of radar precipitation here? And if so, how? e.g. via sub-sampling, time averaging?

**P11L10 Response:** Actually, the radar data were re-sampled to match the higher temporal resolution of PWV-GPS estimates. A linear interpolation function was used in the precipitation time series from radar. This was done for applying the method in the higher resolution without lose of information from GPS. This information has been forgotten in the previous version of the manuscript. The following phrase was included "In this analysis, the precipitation data from Radar, originally with sampling rate of 6 minutes, were linearly interpolated to one-minute rate."

14) **Referee#1:** Section 3.5, Figures 5 and 6: correlations are not very high. You need to comment their magnitude. Also, the caption does not indicates the error bars not how they were drawn.

**P11L15 Fig.8 Fig.9 Response:** Although the correlations are not very high, it was possible to identify statistical significance in this results. This comment was included in the manuscript. Information about the bars were included in the Caption: "The 95% Confidence Interval for each WCC is estimated considering a Gaussian Distribution after applying the Fisher's Z Transformation (Whitcher et al 2000)".

15) **Referee#1:** Page 8. You recall the same information several times. For instance, below, an information that was already given before. lines 10-11: "...the results reported by Adams et al. (2013), who showed that the strongest water vapor convergence is typically 1 hour before heavy precipitation." Page 7, lines 23-24: "... the water vapor may increase through low-level moisture convergence, as suggested by Adams et al. (2013)on the time scale of 32-64 minutes" Please reorganize and avoid repetitions.

**P15L14 Response:** We agree that this information was repeated in the manuscript. This information was removed in this analysis and maintained in the conclusion section, in which is more appropriated for it. The phrase "as suggested by Adams et al. (2013)" was removed by simplification, because this idea and reference was well discussed in the introduction section, as suggested by reviser in the specific comment (9).

16) **Referee#1:** Raymond (1987) is missing in the list of references.

**P19L12 Response:** This reference is in the reference list of the new version of manuscript.

17) **Referee#1:** Section 4, inset in Figure 7: this is the first and only time that the fluctuations of high-frequency GPS PW are presented. I think that this is late in the manuscript, and that this is not enough. I think that you have to present several of the cases presented in table 1, in the form of graphs jointly showing time series of radar precipitation (either fraction above 50 mm/h or another rainfall diagnostic) and GPS PW. I would prefer to see the 18 cases, organized by terciles, or at leat 3 of them per tercile. I further strongly suggest to move section 4 before section 3 (see main comment B).

**P7L1-P8L30 Fig.4 Fig.5 Response:** We agree that the discussion about the GPS-PWV Jump before a high temporal resolution GPS-PWV time series is more appropriated. It was done in this new version of the manuscript. In this new strategy

used to present the PWV-Jump, many parts of the manuscript had to be changed, such as: abstract, introduction in the content of the sections, the section 3 (denominated how "3. Behavior of PWV time series before precipitation events: the GPS-PWV jumps"). The sections 3 and 4, from previous version of the manuscript, were merged (as suggested by reviser 2) denominated now as "4. High temporal resolution GPS-PWV time series analysis". In the conclusions section was changed the sequence of presentation of the results obtained in this work, as consequence of this restructure. Based on this comment was included two new figures in the manuscript. The new Fig. 4 shows other GPS-PWV jumps observed before different extension rain events, one of each tercile. It is important highlight that the there are many precipitation events of lower intensity (lower tercile) which the GPS-PWV jump are not observed. Instead of present the GPS-PWV before of the 18 events, this information were efficiently organized and presented in the new Fig. 5.

Please also note the supplement to this comment:
http://www.atmos-meas-tech-discuss.net/amt-2016-378/amt-2016-378-AC1-
supplement.pdf
* * *
[Figure]

**Supplement:**

**GPS-PWV jumps before intense rain events**

Luiz F. Sapucci[1], Luiz A. T. Machado[1], Eniuce Menezes de Souza[2], Thamiris B. Campos[3]

[1]Centro de Previsão de Tempo e Estudos Climáticos, Instituto Nacional de Pesquisas Espaciais, Cachoeira Paulista, Postal Code: 12630-000, Brazil.

[2]Departamento de Estatística - Universidade Estadual de Maringá, Maringá, Postal Code: 87020-900, Brazil

[3]Programa de Pós-Graduação em Meteorologia, Instituto Nacional de Pesquisas Espaciais, São José dos Campos, Postal Code: 12227-010, Brazil

*Correspondence to*: Luiz Sapucci (luiz.sapucci@cptec.inpe.br)

**Abstract.** A rapid increase in atmospheric water vapor is a fundamental ingredient for many intense rainfall events. High-frequency precipitable water vapor (PWV) estimates (one minute) from a Global Positioning System meteorological site (GPS) are evaluated here for intense rainfall events during the CHUVA Vale field campaign in Brazil (November-December 2011), in which precipitation events of differing intensities and spatial dimensions, as observed by an X-band radar, have been explored. A sharp increase in the GPS-PWV prior to the more intense events has been found and termed GPS-PWV "jumps". These jumps are associated with water vapor convergence and the continued formation of cloud condensate and precipitation particles. The correlation and lags between the high temporal resolution GPS-PWV time series and rainfall events are evaluated. A wavelet cross-correlation analysis shows that there are important spikes in the PWV that precede the more intensity/extension rainfall events on scales related to time periods from about 30 to 60 minutes. The GPS-PWV time-derivative histogram for the period of 60 minutes before the rainfall event reveals different distributions for higher intensity and extension events. This feature could indicate the occurrence of severe precipitation and consequently has the potential for application in nowcasting activities.

**Comentário [s1]:** Reviser1E3

**Comentário [s2]:** Reviser1E17

**1 Introduction**

The application of the Global Positioning System (GPS) tropospheric-induced signal delay to estimate the precipitable water vapor (hereafter, GPS-PWV) is a good example of an indirect solution for quantifying atmospheric humidity. The magnitude of this delay is related to the integral of the refractivity index of the air as a function of temperature, pressure and water vapor (Bevis et al. 1992) on the optical path followed by the GNSS signal. The wet component of this delay provides the precipitable water vapor (PWV) (Bevis et al. 1994), with an error of approximately 5% under all weather conditions (Wolfe and Gutman 2000) relative to other measurement techniques (Sapucci et al. 2007) and in near real time (Rocken et al. 1994). The methodology employed in GPS data processing has been improvement continually to minimize the uncertainty and the PWV estimate has been determined with an accuracy better than 2 mm (Moore et al. 2015; Shangguan et al. 2015) Although the vertical humidity structure is not captured in GPS-PWV estimates, the great advantage of GPS-PWV (in addition to its

all-weather capacity) is its high temporal resolution (minutes) (Zumberge et al. 1997). An important application of the GPS-PWV estimate is its assimilation into the Numerical Weather Prediction process, which has a positive impact on short-range forecasts of humidity fields and, consequently, better precipitation forecasts for heavy rainfall events (Cucurull et al. 2004; Bennitt and Jupp 2012). Other applications become viable in dense networks and transects. GPS-PWV has been useful for:
5   studying the diurnal cycle of convective instability in Japan (Sato and Kimura 2005); investigating water vapor variability during a mistral/sea breeze event in southeastern France exploring GPS-PWV tomography (Bastin et al. 2005); studying water vapor diurnal cycle over African continent (Bock et al. 2008); evaluation of the GPS-PWV values before precipitation (Kursinski et al. 2008); tracking water vapor advection over Amazonian (Adams et al. 2011); GPS-PWV tomography have been used to investigate the water vapor distribution related to convective rainfall events and better understanding and
10  quantification of the hydrological cycle in southeastern France (Brenot et al. 2014); it has been used to propagate convective events and determining topographic effects on the evolution of convection (Adams et al. 2015); studies of convective mesoscale events during the North American Monsoon (Serra et al. 2016) and nowcasting thunderstorm activities, as foreseen by Jerrett and Nash (2001). Guerova et al. (2016) showed development and test of new multi GNSS (Global Navigation Satellite System) products to forecasting of severe weather and emphasized the fact that for short-term, high-
15  resolution forecasting or nowcasting models require more detailed humidity observation, e. g. GPS-PWV estimates.

The relationship between the occurrence of intense rainfall and high concentrations of atmospheric water vapor is well known and has been explored in nowcasting applications. Muller et al. (2009) described a model for the  relationship between the PWV and convective precipitation event in the tropics. PWV data from a microwave radiometer (MWR) with high temporal resolution have been used to analyse and better understand the temporal relations of column water vapor and
20  tropical precipitation (Holloway and Neelin, 2010). Chan (2009) evaluated the performance of a ground-based microwave radiometer in intense convective weather and reported an increasing degree of instability of the troposphere before the occurrence of heavy rain. In addition, he compared PWV from GPS and radiometer, the result showed that radiometer exhibited rather rapid fluctuations during intense precipitation, which are not observed in the GPS-PWV data. Madhulatha et al. (2013) reported a sharp increase in the PWV values approximately 2–4 hours prior to the occurrence of thunderstorms
25  and developed a nowcasting technique using PWV values and 7 other thermodynamic indices from microwave radiometer observations. The relationship between deep convective activity and occurrence of intense rainfall and the temporal evolution of GPS-PWV has been studied for well over a decade. Mazany et al. (2002) developed a lightning prediction index for Florida based on the GPS-PWV magnitude and its temporal evolution. This index is a binary logistic regression model based PWV-GPS and other two variable predictors. The plot of the GPS lightning index time series showed a pattern several
30  hours prior to a lightning, which was tested as forecasting tool of this events. Nowcasting employing GPS-PWV was reported by de Haan et al. (2004), who demonstrated its viability in improving thunderstorm and heavy precipitation forecasts during a cold front passage. De Haan (2006) developed a method for inferring the atmospheric stability from a nonisotropic GPS path-delay signal (slant delay). Book et al. (2008) carried out studies of the West African Monsoon using

Comentário [s3]: Reviser1E1

Comentário [s4]: Reviser1E2

[revised manuscript text omitted]

20    positioning method in post-processing mode with the precise ephemerides of the GPS constellation provided by the NASA Jet Propulsion Laboratory. Sampling rate of the used GPS satellites ephemeris is 15 minutes for orbits and 5 minutes for GPS satellite crock.

To ensure the quality of the PWV time series with high temporal resolution required in this study, in the data-processing strategy  adopted  the  known   uncertainty sources were taken into consideration applying the recommended models and

25    adjustment of parameter exploring available stochastic models. The latest version of the GOA-II software (version 6.3) was used, which estimates parameter with high temporal resolution exploring the sophisticated orbit integrator package to estimate GPS satellite position in each epoch. The ocean tide model FES 2004 (Lyard et al. 2006), recommended by the International Earth Rotation and Reference Systems Service (IERS Conventions 2010) was applied in this processing. Method of antenna absolute calibration (Schmid et al. 2007) was applied by GOA-II to ensure the correct phase center

30    variation of the satellites and receiver antennas using parameters provide by IGS web site (Montenbruck et al. 2015). The data processing with GOA-II software to obtain ZTD estimates was done selecting the Global Mapping Function (Boehm et al. 2006) and the sampling rate of the ZTD estimates of 60 seconds. The others possible parameters that can be selected were

used the configuration basic suggested by JPL. As the configuration items associated with ZTD estimates and respective values used can impact in the variability of the ZTD in high temporal resolution (basic information used in this study), they are listed in Table 1 in order to highlight them.

[revised manuscript text omitted]

The high temporal resolution obtained with the GPS-PWV enables the evaluation of high frequency variations and their relationship with intense precipitation events. The GPS-PWV time series shows a well-defined sharp increase before the occurrence of precipitation, as reported by Kursinski et al. (2008) as a rapid rise in PWV preceding the rain events. Shi et al. (2015) using GPS -PWV to monitoring the water vapor variation shown that ascending and descending patterns of GPS-PWV can be identified before and after each rainfall event. There are strong oscillations, generating a significant increase in the total water vapor content until a maximum is reached. Subsequently, a strong GPS-PWV reduction is observed, and after a short interval, the precipitation also reaches a maximum peak. Here, this sharp increase is called GPS-PWV jump. Fig. 3 shows a typical case exemplifying the PWV behavior before precipitation occurs on DoY 341; this was one of the strongest events registered during the CHUVA Vale experiment. Before the severe precipitation begins, the GPS-PWV follows several pulses, increasing the value and forming the PWV jump, until it reaches a peak of maximum value. After the GPS-PWV crest, a decreasing period is observed some minutes before severe precipitation. Fig. 3 clearly shows this configuration of a crest in the GPS-PWV time series around precipitation (composed of several pulses) and its subsequent decrease immediately before the beginning of stronger precipitation.

This GPS-PWV behavior before precipitation occurs not only for more intense events but also for lower rainfall rates. Fig. 4 shows other GPS-PWV jumps observed before rain events with different intensity/extension occurred on: DoY 315 (41% of precipitation fraction above 35 mm h$^{-1}$), DoY 332 (5%) and DoY 358 (only 1% of precipitation fraction above 35 mm h$^{-1}$). This figure shows the intensity of these sharp increases in GPS-PWV are larger before more extensive precipitation. Table 2 presents all precipitation events in which the rainfall above 20 mm h$^{-1}$ was observed by XPol radar in the area of 4.4 km per 4.4km around GPS antenna. This table presents the DoY and time of maximal radar precipitation of the each event and the respective fraction rain above 20 mm h$^{-1}$, 35 mm h$^{-1}$ and 50 mm h$^{-1}$. In order to obtain an overview of the hardiness of the GPS-PWV jumps feature before precipitation, Fig. 5 show the composite mean of GPS-PWV time series from 60 minutes before to 60 minutes after maximum observed precipitation (fraction rain above 35 mm h$^{-1}$) for 18 events listed by Table 2. The composite presented in Fig. 5 is normalized by maximum GPS-PWV values before precipitation. The composite mean shows that the GPS-PWV jump is strongly remarkable before the maximum precipitation, which the maximum of the composite mean is observed in 30 minutes and lower dispersion in 25 minutes before the maximum precipitation. The time lag between the maximum GPS-PWV and the time of the maximum rainfall is presented in a sub plot in Fig.5. It is important highlight that the there are many precipitation events of lower intensity and extension in which the GPS-PWV jumps are not observe. In these cases, the maximum GPS-PWV is observed in the maximum precipitation, consequently the time lag is zero (37% of the cases evaluated). In this histogram, it is clear there is a range of lag time between the GPS-PWV and precipitation maximums, the precipitation area and the temporal resolution of GPS-PWV here employed have an impact on this lag time. For instance, Adams et al. (2013) using rain gauge and PWV with sampling rate of 30 minutes over Amazonian region found out lag zero, however, in this study we shows that the GPS-PWV signal can represent a precipitation area and

**Comentário [s13]:** Reviser1E17

**Comentário [s14]:** Reviser1E17

not only a punctual measurement over the GPS antenna. An analysis of the time lag correlation is necessary to define this interval between the maximal GPS-PWV and precipitation, which is carried out in the next section.

The physical explanation for this behavior could be explained by different physical processes. First, the water vapor may increase through low-level moisture convergence. The variation of the moisture convergence generates a sequence of pulses of positive increases in the PWV value. The sequence of pulses of positive increases in the PWV could be a result of several physical processes. Some of the physical process that can explain these pulses could be low-level water vapor convergence forced by gravity wave (Raymond 1987) or simply unstable surface parcels accelerating upwards. After the increase of GPS-PWV at the crest of the jump, rainfall starts and PWV starts to decrease. The lag time between the crest and the maximum precipitation can vary from one region to another, from one rainfall cell to another because it depends on the cloud condensation nuclei and the precipitation efficiency, normally a function of the wind shear. In this study using precipitation measures based on area about the GPS antenna, we found a time lag between these maximum. Addams et al. (2013) considered that the conversion of water vapor to liquid water and precipitation are of second order during the process of PWV increasing. It is probably true during the phase of cloud formation, however, when the precipitation starts, water vapor decreases due to the formation of liquid water. The conversion of the water vapor to liquid water changes the dielectric medium, where the refractivity is induced by the displacement of charge (Solheim et al. 1999). While the refractivity from water vapor is due to the polar nature of the water molecule, the GPS phase delay induced by liquid water (hydrometeor) is proportional to the electric permittivity of the formed dielectric medium and, consequently, much lower than the delay generated by water vapor. Another important physical mechanism is the storm downdraft that is dryer and colder that the ascending moisture air and this downdraft can also contributes to the decrease in PWV. In addition, PWV decreases after precipitation starts can be simply associated to the final process of surface convergence as function of the rainfall and downdrafts on the surface, or by the advection process forced by the shear and storm movement. Given the limitations of the observations, our interpretation of the physical mechanisms responsible for the jumps remains speculative. To understand what the physical mechanisms responsible for the pulses and GPS-PWV jumps, a specific field campaigns design are needed.

**4 High temporal resolution GPS-PWV time series analysis**

The high temporal resolution GPS-PWV time series and precipitation in different intensity and extension are evaluated and the PWV-GPS jumps are characterized. The wavelet analysis are explored to evaluating which the timescale of the GPS PWV oscillations are associated with a more intense rainfall occurrence and time lag correlation analysis are used to determine the lag between the rainfall intensity and the GPS-PWV time series. Additionally, PWV-GPS derivative analysis is explored and its potential for nowcasting application is discussed.

**4.1 Wavelet analysis**

Wavelet analysis was used to perform a detailed analysis of the GPS-PWV time series and to evaluate the variability within different time scales (denoted here as intra-relation), as well as to assess the relationship between the GPS-PWV time series and the precipitation time series (denoted here as interrelation wavelet analysis). This methodology enables simultaneous decomposition of the PWV time series as a function of time and frequency (Daubechies 1992). Consequently, accessing to the information regarding the signal amplitude/frequency and its variation as a function of time becomes possible.

To perform the intra-relation analysis evaluation of how spectral characteristics change over scales ($s$) and time ($t$), but with highly redundant information, the continuous wavelet analysis (Torrence and Compo 1998) was used to estimate the wavelet power spectrum. Thus, a decomposition of the GPS-PWV data into time-variability space allows an evaluation of the main frequencies composing the GPS-PWV time series during intense rainfall events. With continuous analysis, some hidden features of the time series can be identified, e.g., in which scale are the most representative behaviors of the time series. Continuous analysis is often easier to interpret because its redundancy tends to reinforce the traits and makes all information more visible. However, for some specific choices of values for time and frequency, it is possible to apply a discrete wavelet transform, which does not lose important information and has advantages of implementation and computational effort. This is the case of the non-decimated discrete wavelet transform (NDWT), also called Maximal Overlap Discrete Wavelet Transform, which can be seen as a compromise between the discrete wavelet transform (DWT) and CWT because of its redundancy, but not as redundant as CWT. The NDWT can be computed similarly to the ordinary DWT but without subsampling (decimation), ensuring the translational invariance, which is ideal for analysing time series, especially interrelations between different time series. A time-variant transform disrupts the lag-resolution in a cross-correlation analysis. Furthermore, estimators calculated using the NDWT are considered more preferable because they are asymptotically more efficient than the estimator based on the DWT (Percival and Walden 2000). As the bivariate relationship between two time series is essential for this research, a wavelet cross-correlation (WCC) constructed from NDWT (Whitcher et al. 2000) is ideal for analysing different scale structures and the interrelations of the dynamic behaviour of two time series, as well as the lead-lag relationships. Some lead-lag relations that could not be distinguished in the usual cross-correlation can be investigated in the WCC, which decomposes the cross-correlation on a scale-by-scale basis. Thus, a specific scale may be associated with water vapor convergence before precipitation occurrence, also providing a lead time for nowcasting, a specific time scale for calculations and additional information to understand the physical mechanisms associated with intense rainfall events in this region.

Considering the large quantity of available discrete mother wavelets, some of the most used in the literature were evaluated: Daubechies with 4, 6, 8, and 16 coefficients, denoted by D4, D6, D8, D16, and Daubechies Least Asymetric with 8, 16, and 20 coefficients, denoted by LA8, LA16, and LA20, respectively. To verify the statistically significance of the estimated wavelet correlations, the 95% confidence interval was estimated considering a Gaussian Distribution after applying the Fisher`s Z Transformation (Whitcher et al. 2000).

**Comentário [s15]:** Reviser1E9

**Comentário [s16]:** Reviser1E14

Because of implementation aspects considered in this study, the two time series were restricted to a power of two length, with 65,536 ($2^{16}$) observations, excluding one day at the beginning and another at the end of the GPS-PWV series. The time span corresponds to 45 days, 12 hours and 16 minutes, beginning on DoY 314 (00 UTC 10 November) and finishing on DoY 359 (1216 UTC 27 December). The same time series length was used for both intra- and interrelation analyses. Because each wavelet scale $j$ in the interrelation analysis corresponds to a frequency band from $2^j$ to $2^{j+1}$, its inversion allows the interpretation in terms of a period of time also in a dyadic interval. Thus, WCC enables the identification of the most important scale of the PWV oscillation during precipitation events. Furthermore, the WCC of GPS-PWV and precipitation time series also allows evaluating the lead-lag correlation that may exist between these time series for the different  time periods.

**4.1.1 Wavelet power spectrum analysis**

For an intra-relation analysis, the wavelet power spectrum of GPS-PWV and 95[th] percentile of the precipitation intensity time series are presented in Fig. 6. To emphasize the highest-frequency oscillations, the power spectrum in Fig. 6 shows the scales that represent the period below 512 minutes (~8.5 h). The PWV diurnal cycle presents strong power along all time series, and consequently, it was not taken into consideration in this analysis. The range of power spectrum associated with color scale used in Fig. 6 is for scales related to time periods larger than 16 minutes.

The PWV series with a one-minute temporal resolution presents oscillations of high frequency; however, the frequency of occurrence of rainfall events is very low. For this reason, it is necessary to take into consideration a long time period (with several precipitation events) and a short time step, e.g., the one-minute interval used in this study. Consequently, in a general analysis of the PWV wavelet power spectrum, it is difficult to clearly discern which power is associated with each time step. However, a more specific analysis of the wavelet power spectrum during precipitation events indicates that there are, as expected in function of described GPS-PWV jump, strong changes in the power between different scales; cases with an increase in the power of the oscillation from low to high frequency are observed. This result indicates that PWV oscillations on scales related to time periods smaller than 128 minutes occur more frequently during precipitation events than in periods without rain. Fig. 7 presents the same wavelet power spectrum presented in Fig. 6 but with an enlargement applied for precipitation events observed during the period from DoY 340 to DoY 343. The analysis of the power spectrum in these cases makes the result discussed for Fig. 6 more understandable. For clarity, a vertical line was drawn for each precipitation maximum. Fig. 7 shows that the power of the PWV oscillations between 128 and 32 minutes is stronger during more intense precipitation events (for example, the events that occurred on DoY 341 at 1836 UTC and DoY 342 at 1636 UTC) than during light rainfall events (events observed during DoY 343) and periods without rain (DoY 340). It is interesting to observe that GPS-PWV presents a jump at the end of the DoY 357 with a strong signal in the wavelet power spectrum, but XPol radar did not detect any precipitation. The wavelet power spectrum (Fig. 6) also shows the impact of the GPS failures

**Comentário [s17]:** Reviser 1E10

**Comentário [s18]:** Reviser1E11

**Comentário [s19]:** Reviser1E12

**Comentário [s20]:** Reviser1E11

that occurred during DoY 331 and 348, which are unfortunately very close to the other intense precipitation events that occurred on DoY 332 and 348. For this reason, these cases are not spotlighted in the wavelet analysis.

**4.1.2 Wavelet cross-correlation analysis**

Before presenting the inter-relation results, one might question about the robustness of the method regarding the choice of mother wavelets. We estimated the wavelet correlations using Daubechies and Least Asymmetric wavelets with different filter lengths (coefficient number) as presented in Section 4.1. The results were quite similar independently of the mother wavelet, but the correlations were maximized when the mother wavelet has a larger filter or more coefficients (D16, LA16, and LA20). Thus, all estimated wavelet correlations were presented using the LA20 mother wavelet. Fig. 8 shows the wavelet correlation and its 95% confidence interval between the GPS-PWV and the 95[th] percentile of the precipitation intensity as a function of the wavelet scale, represented by the respective time periods, considering the lag zero. In this analysis, the precipitation data from XPol radar, originally with sampling rate of 6 minutes, were linearly interpolated to one-minute rate. The results show that the wavelet correlation between the PWV and precipitation intensity is stronger for the scale related to the period between 32 and 64 minutes, indicating the scale on which the most important GPS-PWV oscillations associated with precipitation events occur. After this scale, the correlation decreases, followed by another increase due to the influence of the diurnal cycle. Although the correlations are not very high, 95% confidence interval showed that these results are statistically significant.

To evaluate the results presented in Fig. 8 for different intensity and extension of precipitation events, Fig. 9 shows the wavelet correlation for lag zero between GPS-PWV and precipitation fractions as a function of the period bands for different rain fraction intensities (>20 mm h$^{-1}$, Fig. 9a; >35 mm h$^{-1}$, Fig. 9b; and >50 mm h$^{-1}$, Fig. 9c). The 95% percentiles give an information about the maximum rain rate (which can be only one point) on the area of 4.4 km per 4.4 km around the GPS antenna. The rain fraction gives an information about the fraction of these studied area covered by rain rate above of these thresholds. There are some events where the rain intensity can be high and the fraction small, for instance, the cases of isolate clouds. Therefore, the area fraction presents a more close representation of rainfall events related to low level convergence because it gives information about the amount of liquid water in the area. The peak of the wavelet correlation observed in the time period from 32 to 64 minutes is significant for the three rain thresholds. The plots of Fig. 9 also show when only heavy to torrential precipitation events are taken into consideration (hereafter called intense rain events), stronger wavelet correlation is observed.

To perform the lead-lag analysis, the WCC is showed in Fig 10 between GPS-PWV and rain fraction for different intensities (>20 mm h$^{-1}$, Fig. 10a; >35 mm h$^{-1}$, Fig. 10b; and >50 mm h$^{-1}$, Fig. 10c). Although the precipitation and GPS-PWV time series present very distinct behaviors, the WCC permitted the identification of the correlation in lead-lag of about 30 minutes mainly for scales related to time periods from 32 to 64 minutes, indicating which GPS-PWV oscillations are important for predicting precipitation events. This results show also that considering the rain fraction for stronger intensities (>50 mm h$^{-1}$), some statistically significant correlations in lead-lag of about 30 minutes also appear stronger from period bands of 4- 16

**Comentário [s21]:** Reviser1E13

**Comentário [s22]:** Reviser1E14

**Comentário [s23]:** Reviser1E11

**Comentário [s24]:** Reviser1E1

and 16-32 minutes. This result indicates that the GPS-PWV carries some information, mainly, on the scale related to the period from 32 to 64 minutes. That signals the occurrence of precipitation events of large intensity and extension, which suggests that the GPS-PWV jumps are, for these storms evaluated, concentrated in this scale emphasizing its potential in a nowcasting application. It is also important to highlight that the correlation is larger on this scale than on adjacent scales.

5    Furthermore, these results indicate that, although GPS-PWV jump before precipitation occurs not only for more intense events, the intensity of these oscillations associated to jumps is greater before the most intense and extensive ones. However, this feature should be considered as specific to each region because the time scale between the humidity convergence and rainfall may depend on many physical processes and environmental conditions (e.g., gravity wave-induced convergence, wind shear or thermodynamic instability).

10   **4.2 Time lag correlation analysis**

The relationship between the rainfall intensity (or rain fraction) and the GPS-PWV is different for each event; however, the GPS-PWV peak is a well-delineated pattern. The time interval between the moment of the PWV crest and the maximal precipitation can vary among cases. The WCC shows on which scale the correlation between GPS-PWV and precipitation time series is  higher, as well as the lead-lag interrelation between them. However, evaluating the lag correlation for positive

[revised manuscript text omitted]

The result found by GPS-PWV derivative analysis suggests that an algorithm for intense precipitation forecasting using the

15     GPS-PWV should consider the following points: (a) increases in GPS-PWV positive variations compared with negative ones in which the median values of the variation in the last 60 minutes reach positive values and (b) a simultaneous increase in the population of the GPS-PWV derivatives above +9.5 mm h$^{-1}$. The value of the stochastic constrains applied on temporal evolution of ZWD during the GPS data processing can make influence the PWV variability in high temporal resolution. The value used in this study was the default, but some tests with different constraint can be carried out to identify the impact of

20     this parameter in the derivative analysis before intense precipitation.

The GPS-PWV values evaluated in this study are post-processed, and an additional study is required to determine whether these estimates in real time are able to capture the jumps before the precipitation reported. True real-time processing for PWV estimates in dense and regional GPS networks has been explored in other studies related to nowcasting applications (Iwabuchi et al. 2006). De Haan and Holleman (2009) reported the construction and validation of a real-time PWV map from

25     a GPS network combined with data from weather radar, a lightning detection network, and surface wind observations. They tested a nowcasting algorithm for three thunderstorm case studies and concluded that the GPS-PWV in real time can be helpful for the nowcasting of severe thunderstorms. Shi et al. (2015) studied the PWV estimates in real time for rainfall monitoring and forecasting and shown that this estimate has quality comparable with post-processed product. A significantly reduction in the latency was obtained with GPS data processing proposed by Shi et al. (2015), which demonstrated

30     promising perspective of the PWV-GPS data for rainfall forecasting.

**5 Conclusions**

This work evaluates the correlation between large and rapid increases in the GPS-PWV and the occurrence of rainfall events observed by radar during the CHUVA Vale experiment in Brazil. A detailed analysis of the GPS-PWV time series was carried out, and strong and sudden sharp increase composed predominantly by positive derivatives, before the precipitation events were identified and called as GPS-PWV jumps. In this process, a crest in the PWV series is remarkable before the precipitation events. Although this sharp increase can be observed for any precipitation event, it is preponderant before more intense and extensive precipitation events.

The wavelet analysis for the GPS-PWV time series was explored to characterize the strong changes in the power spectrum between different time scales during precipitation events generated by the occurrence of the GPS-PWV jumps. Additionally, the application of wavelet cross-correlation between the PWV and precipitation showed that important oscillations exist between these variables on the scale related to a time period from 32 to 64 minutes, which is stronger for events of large intensity and extension. These results corroborates with those reported by Adams et al. (2013), who showed that the strongest water vapor convergence is typically ~1 hour before heavy precipitation.

[revised manuscript text omitted]

Bock, O., M. N. Bouin,E. Doerflinger, P. Collard, F. Masson, R. Meynadier, S. Nahmani, M. Koité, K. Gaptia Lawan Balawan, F. Didé, D. Ouedraogo, S. Pokperlaar, J.-B. Ngamini, J. P. Lafore, S. Janicot, F. Guichard, M. Nuret. 2008: The West African Monsoon observed with ground-based GPS receivers during AMMA, J. Geophys. Res., 113 (D21105) doi: 10.1029/2008JD010327.

Boehm, J., B. Werl, and H. Schuh, 2006: Troposphere mapping functions for GPS and VLBI from ECMWF operational analysis data. J. Geophys. Res., 111, B02406, doi:10.1029/2005JB003629.

Brenot, H., A. Walpersdorf, M. Reverdy, J. van Baelen, V. Ducrocq, C. Champollion, F. Masson, E. Doerflinger, E., Collard, P., and Giroux, P. 2014: A GPS network for tropospheric tomography in the framework of the Mediterranean hydrometeorological observatory Cévennes-Vivarais (southeastern France), Atmos. Meas. Tech., 7, 553-578, doi:10.5194/amt-7-553-2014.

Calheiros, A. J. P., and L. A. T. Machado, 2014: Cloud and rain liquid water statistics in the CHUVA campaign. Atmosph. Research, 144, 126-140, doi:http://dx.doi.org /10.1016/j.atmosres.2014.03.006.

Chan, P. W., 2009: Performance and application of a multi-wavelength, ground-based microwave radiometer in intense convective weather. Meteorol. Z., 18, 3, 253-265, doi:10.1127/0941-2948/2009/0375.

Cucurull, L., F. Vandenberghe, D. Barker, E. Vilaclara, and A. Rius, 2004: Three-Dimensional Variational Data Assimilation of Ground-Based GPS ZTD and Meteorological Observations during the 14 December 2001 Storm Event over the Western Mediterranean Sea. Mon. Wea. Rev., 132, 749–763, doi:http://dx.doi.org/10.1175/1520-0493(2004)132<0749:TVDAOG>2.0.CO;2.

Daubechies, I., 1992: Ten Lectures on Wavelets. Society for Industrial and Applied Mathematics, 357 pp.

Davis, J. L., T. A. Herring, I. Shapiro, A. E. Rogers, and G. Elgened, 1985: Geodesy by radio Interferometry: Effects of Atmospheric Modeling Errors on Estimates of BaseLine Length. Radio Sci., 20, 1593-1607, doi: 10.1029/RS020i006p01593.

Giangrande, S. E., T. Toto, A. Bansemer, M. R. Kumjian, S. Mishra, and A. V. Ryzhkov, 2016: Insights into riming and aggregation processes as revealed by aircraft, radar, and disdrometer observations for a 27 April 2011 widespread precipitation event, J. Geophys. Res. Atmos., 121, 5846–5863, doi:10.1002/2015JD024537.

Gematronik, 2007: Dual-polarization weather radar handbook. In: Bringi, V.N., Thurai, M., Hannesen, R. (Eds.), Selex-SI gematronik, 2nd edition (163 pp.).

Gregorius, T., 1996: GIPSY-OASIS II How it works. Department of Geomatics, University of Newcastle upon Tyne, 167 pp. [Available online at: http://web.gps.caltech.edu/classes/ge167/file/gipsy-oasisIIHowItWorks.pdf.].

Guerova G, J. Jones, J. Dousa, G. Dick, S. De Haan, E. Pottiaux, O. Bock, R. Pacione, G. Elgered, H. Vedel, M. Bender, 2016: Review of the state-of-the-art and future prospects of the ground-based GNSS meteorology in Europe. Atmos Meas Tech Discuss. doi:10.5194/amt-2016-125

de Haan, S. D., S. Barlag, H. K. Baltink, F. Debie, and H. V. Marel, 2004: Synergetic Use of GPS Water Vapor and Meteosat Images for Synoptic Weather Forecasting. J. Appl. Meteoro., 43, 514-518, doi:http://dx.doi.org/10.1175/1520-0450(2004)043<0514: SUOGWV>2.0.CO;2.

de Haan, S. D., 2006: Measuring Atmospheric Stability with GPS. J. Appl. Meteoro. Climatol., 45, 467-475, doi: http://dx.doi.org/10.1175/JAM2338.1.

de Haan, S. D., and I. Holleman, 2009: Real-Time Water Vapor Maps from a GPS Surface Network: Construction, Validation, and Applications. J. Appl. Meteoro. Climatol., 48, 1302-1316, doi: http://dx.doi.org/10.1175/2008JAMC2024.1.

Holloway, C. and J. Neelin, 2010: Temporal Relations of Column Water Vapor and Tropical Precipitation. J. Atmos. Sci., 67, 1091–1105, doi: http://dx.doi.org/10.1175/2009JAS3284.1.

Iwabuchi, T., C. Rocken, Z. Lukes, L. Mervat, J. Johnson, and M. Kanzaki, 2006: PPP and Network True Real-time 30 sec Estimation of ZTD in Dense and Giant Regional GPS Network and the Application of ZTD for Nowcasting of Heavy Rainfall. Proceedings of the 19th International Technical Meeting of the Satellite Division of The Institute of Navigation (ION GNS 2006), 1902-1909.

IERS Conventions, 2010: Gérard Petit and Brian Luzum (eds.). (IERS Technical Note ; 36) Frankfurt am Main: Verlag des Bundesamts für Kartographie und Geodäsie, 2010. 179 pp., ISBN 3-89888-989-6 https://www.iers.org/IERS/EN/Data Products/ Conventions/conventions.html.

Jerrett, D., and J. Nash, 2001: Potential Uses of Surface Based GPS Water Vapour Measurements for Meteorological Purposes. Phys. Chem. Earth (A), 26, 457-461, doi:10.1016/S1464-1895(01)00083-7.

Joss, J., and A. Waldvogel, 1967: Ein Spektrograph für Niederschlags-tropfen mit automatischer Auswertung (A spectrograph for rain drops with automatical analysis). Pure Appl. Geophys., 68, 240–246, http://dx.doi.org/10.1007/BF00874898.

Kinnell, P. I. A., 1976: Some Observations on the Joss-Waldvogel Rainfall Disdrometer, J. Appl. Meteor., 15, 499–502, doi:http://dx.doi.org/10.1175/1520-0450(1976)015<0499:SOOT JW>2.0.CO;2.

Kursinski, E. R., R. A. Bennett, D. Gochis, S. I. Gutman, K. L. Holub, R. Mastaler, C. Minjarez Sosa, I. Minjarez Sosa, and T. van Hove (2008), Water vapor and surface observations in northwestern Mexico during the 2004 NAME Enhanced Observing Period, Geophys. Res. Lett., 35, L03815, doi:10.1029/2007GL031404.

Lyard, F., F. Lefèvre, T. Letellier, and O. Francis, 2006: Modelling the global ocean tides: a modern insight from FES2004, Ocean Dynamics, 56, 394-415, doi:http://dx.doi.org/10.1007/s10236-006-0086-x.

Machado, L. A. T., M. A. F. Silva Dias, C. Morales, G. Fisch, D. Vila, R. Albrecht, S. J. Goodman, A. J. P. Calheiros, T. Biscaro, C. Kummerow, J. Cohen, D. Fitzjarrald, E. L. Nascimento, M. S. Sakamoto, C. Cunningham, J. P. Chaboureau, W. A. Petersen, D. K. Adams, L. Baldini, C. F. Angelis, L. F. Sapucci, P. Salio, H. M. J. Barbosa, E. Landulfo, R. A. F. Souza, R. J. Blakeslee, J. Bailey, S. Freitas, W. F. A. Lima, and A. Tokay, 2014: The Chuva Project: How Does Convection Vary across Brazil? Bull. Amer. Meteor. Soc., 95, 1365–1380, doi:http://dx.doi.org/10.1175/BAMS-D-13-00084.1.

Madhulatha, A., M. Rajeevan, M. Venkat Ratnam, J. Bhate, and C. V. Naidu, 2013: Nowcasting severe convective activity over southeast India using ground-based microwave radiometer observations. J. Geophys. Res., 118, doi:10.1029/2012JD018174.

Mazany, R. A., S. Businger, S. I. Gutman, and W. Roeder, 2002: A lightning prediction index that utilizes GPS integrated precipitable water vapor. Wea. Forecasting, 17, 1034–1047, doi:10.1175/1520-0434(2002)017<1034:ALPITU>2.0.CO;2.

Montenbruck, O., R. Schmid, F. Mercier, P. Steigenberger, C. Noll, R. Fatkulin, S. Kogure, A.S. Ganeshan, 2015: GNSS satellite geometry and attitude models, Advances in Space Research, 56, 6, 1015-1029. Doi:http://dx.doi.org/10.1016/j.asr.2015.06.019.

Moore, A., I. Small, S. Gutman, Y. Bock, J. Dumas, P. Fang, J. Haase, M. Jackson and J. Laber, 2015: National Weather Service Forecasters Use GPS Precipitable Water Vapor for Enhanced Situational Awareness during the Southern California Summer Monsoon. Bull. Amer. Meteor. Soc., 96, 1867–1877, doi: 10.1175/BAMS-D-14-00095.1. Muller, C. J., L. E. Back, P. A. O'Gorman, and K. A. Emanuel, 2009: A model for the relationship between tropical precipitation and column water vapor. Geophys. Res. Lett., 36, L16804, doi:10.1029/2009GL039667.

Percival, D. B., and A. T. Walden, 2000: Wavelet Methods for Time Series Analysis. Cambridge University, 594 pp.

Raymond, D. J. 1987: A Forced Gravity Wave Model of Serf-Organizing Convection. J. Atmo. Sci., 44, 23, 3528-3543.

Rocken, C., T. VanHove, M. Rothacher, F. Solheim, R. Ware, M. Bevis, S. Businger, and R. Chadwick, 1994: Towards near-real-time estimation of atmospheric water vapor with GPS. Eos Trans. AGU, 7544, Fall Meet. Suppl., 173.

Sapucci, L. F., 2014: Evaluation of Modeling Water-Vapor-Weighted Mean Tropospheric Temperature for GNSS-Integrated Water Vapor Estimates in Brazil. J. Appl. Meteor. Climatol., 53, 715–730, doi:http://dx.doi.org/10.1175/JAMC-D-13-048.1.

Sapucci, L. F., L. A. T. Machado, J. F. G. Monico, and A. Plana-Fattori, 2007: Intercomparison of Integrated Water Vapor Estimative from multi-sensor in Amazonian Regions. J. Atmos. Oceanic Technol., 24, 1880–1894, doi:http://dx.doi.org/10.1175/JTECH2090.1.

Serra, Yolande L., David K. Adams, Carlos Minjarez-Sosa, James M. Moker, Jr., Avelino Arellano, Christopher Castro, Arturo Quintanar, Luis Carlos Alatorre, Alfredo Granados-Olivas, Enrique Vazquez, Kirk L. Holub, Charles DeMets, 2016: The North American Monsoon GPS Transect Experiment 2013. *Bull. Amer. Meteor. Soc.,* **97**, 2103–2115, doi: 10.1175/BAMS-D-14-00250.1. Sato, T and F. Kimura, 2005: Diurnal Cycle of Convective Instability around the Central Mountains in Japan during the Warm Season. J. Atmo. Sci., 62, 1626-1636.

Schmid, R., P. Steigenberger, G. Gendt, M. Ge, M. Rothacher, 2007: Generation of a consistent absolute phase-center correction model for GPS receiver and satellite antennas. Journal of Geodesy, 81: 781. 81, 12, 781–798, doi:10.1007/s00190-007-0148-y

Shangguan, M., S. Heise, M. Bender, G. Dick, M. Ramatschi, and J. Wickert, 2015: Validation of GPS atmospheric water vapor with WVR data in satellite tracking mode, Ann. Geophys., 33, 55-61, doi:10.5194/angeo-33-55-2015.

Shi, J., X. Chaoqian, G. Jiming and G. Yang, 2015: Real-Time GPS precise point positioning-based precipitable water vapor estimation for rainfall monitoring and forecasting, IEEE Trans. Geosci. Remote Sens, vol.53, pp. 3452-3459, doi: 10.1109/TGRS.2014.2377041.

**Comentário [s28]:** Reviser1E16

Solheim, F. S., J. Vivekanandan, R. H. Ware, and C. Rocken, 1999: Propagation delays induced in GPS signals by dry air, water vapor, hydrometeors, and other particulates. J. Geophys. Res., 104, 9663–9670, doi:10.1029/1999JD900095.

Testud, J., E. L. Bouar, E. Obligis, and M. Ali-Mehenni, 2000: The Rain Profiling Algorithm Applied to Polarimetric Weather Radar. J. Atmos. Oceanic Technol., 17, 332–356, doi:http://dx.doi.org/10.1175/1520-0426(2000)017<0332:TRPAAT>2.0.CO;2.

Torrence, C., and G. P. Compo, 1998: A Practical Guide to Wavelet Analysis. Bull. Amer. Meteor. Soc., 79, 61-78.

Whitcher, P., D. B. Guttorp, and B. Percival, 2000: Wavelet analysis of covariance with application to atmospheric time series. J. Geophys. Res., 105, 14941–14962, doi::10.1029/2000JD900110.

Wolfe, D. E., and S. I. Gutman, 2000: Developing an operational, surface-based, GPS, water vapor observing system for NOAA: Network Design and Results. J. Atmos. Ocean. Technol., 17, 426-440.

Zumberge, J. F., M. B. Heflin, D. C. Jefferson, M. M. Watkins, and F. H. Webb (1997), Precise point positioning for the efficient and robust analysis of GPS data from large networks, J. Geophys. Res., 102(B3), 5005–5017, doi:10.1029/96JB03860.

**Tables**

**Table 1: Configuration items associated with ZTD estimates and respective values used GPS data processing using GOA-II software.**

| Configuration item | Parameter used | Comments |
|---|---|---|
| GPS data file for GOA-II | Rinex format in sampling rate of 1 second | Data collection from CHUVA |
| Mapping function | Global Mapping Function (Boehm et al. 2006) | Selected in the data processing |
| Cut-off elevation angle: | 10° | Selected in the data processing |
| Elevation weight in the Mapping function | Elevation dependent and apply 3 coefficients depend on the latitude and height above sea level of the observing site and on the DoY | (Boehm et al. 2006) |
| ZHD a priori | Constant using a Tropospheric model in function of high of the GPS station (2.126 m) | Default values |
| ZWD a priori | Constant (0.10 m) | Default values |
| Process noise for tropospheric delay | Random walk | Default values |
| Maximum drift for tropospheric delay | 8.333E-8 (km per square-root second); | Default values |
| Tropospheric Gradients estimates | yes | Default values |
| Maximum drift for Gradients estimates | 8.333E-9 (km per square-root second) | Default values |
| Sampling rate of the ZTD estimates | 60 seconds | Selected in the data processing |

**Table 2: Precipitation events observed by radar during CHUVA Vale experiment in different extension terciles as a function of the precipitation fractions above 35 mm h$^{-1}$.**

| Event | DoY | Maximal radar precipitation UTC Time (hh:mm) | Precipitation fraction observed by XPol radar (%) | | | Terciles |
|---|---|---|---|---|---|---|
| | | | Above 50 mm h$^{-1}$ | Above 35 mm h$^{-1}$ | Above 20 mm h$^{-1}$ | |
| 1 | 348 | 02:42 | 73 | 85 | 95 | |
| 2 | 354 | 21:12 | 28 | 45 | 63 | |
| 3 | 341 | 18:36 | 36 | 41 | 45 | Upper |
| 4 | 315 | 17:12 | 26 | 41 | 49 | tercile |
| 5 | 335 | 19:24 | 30 | 38 | 42 | |
| 6 | 352 | 20:00 | 2 | 33 | 84 | |
| 7 | 342 | 16:36 | 24 | 27 | 28 | |
| 8 | 326 | 21:18 | 8 | 26 | 45 | |
| 9 | 343 | 01:06 | 4 | 9 | 15 | Middle |
| 10 | 338 | 20:18 | 0 | 8 | 19 | tercile |
| 11 | 332 | 19:18 | 5 | 5 | 5 | |
| 12 | 333 | 19:42 | 0 | 2 | 8 | |
| 13 | 314 | 21:06 | 2 | 2 | 7 | |
| 14 | 327 | 00:36 | 0 | 1 | 25 | |
| 15 | 358 | 23:00 | 0 | 1 | 10 | Lower |
| 16 | 317 | 21:48 | 0 | 1 | 2 | tercile |
| 17 | 318 | 08:48 | 0 | 0 | 19 | |
| 18 | 331 | 17:12 | 0 | 0 | 2 | |

**Table 3. Statistical measurements of the GPS-PWV derivative for different extension terciles of precipitation events.**

| Statistical Measurements | Other cases | Terciles | | |
|---|---|---|---|---|
| | | Lower | Middle | Upper |
| Average value (mm h$^{-1}$) | +0.04 | -0.18 | -0.38 | 0.13 |
| Standard deviation (mm h$^{-1}$) | ±2.52 | ±3.18 | ±4.76 | ±5.57 |
| Median (mm h$^{-1}$) | 0.00 | 0.00 | +0.29 | +0.65 |
| Mode (mm h$^{-1}$) | 0.00 | 0.00 | +1.00 | +1.00 |
| Maximal value (mm h$^{-1}$) | +21.13 | +8.42 | +11.00 | +13.25 |
| Minimal value (mm h$^{-1}$) | -19.07 | -6.99 | -17.30 | -14.15 |
| % > +9.5 mm h$^{-1}$ | 0.21% | 0.00% | 0.78% | 7.81% |
| % < -9.5 mm h$^{-1}$ | 0.38% | 0.00% | 4.68% | 5.47% |

**Figures**

[Figure]

**Figure 1: Illustration of the CHUVA Vale experiment, in which the sites where the XPol radar, GPS receiver and disdrometer were installed are indicated. The area of 4.4 km per 4.4 km around the GPS station is highlighted in this figure over the precipitation field observed by XPol radar on December 14th (DoY 348) of 2011. Some details about the composition of this area using the points of the XPol gridded map are additionally presented.**

[Figure]

**Figure 2: Time series of the precipitation and GPS-PWV obtained during CHUVA Vale campaign: (a) GPS-PWV time series; (b) precipitation intensity observed by Joss disdrometer; (c) 95th percentile of the precipitation intensity observed by XPol radar in the area of 4.4 km per 4.4 km centered on the GPS antenna; and (d) precipitation fraction in the area of 4.4 km per 4.4 km centered on the GPS antenna, where the black bar is the fraction above 20 mm h-1, the dark gray bar is the fraction above 35 mm h-1 and the light gray bar is the fraction above 50 mm h-1.**

[Figure]

**Figure 3: GPS-PWV jump observed in the 2-hour period before a heavy storm occurred during DoY 341 (1836 UTC 7 December 2011).**

**Comentário [s29]:** Reviser1E15

[Figure]

**Figure 4: Other cases of GPS-PWV jumps observed before precipitation events with fraction above 35 mm h⁻¹ of different extension occurred during (a) DoY 315 (1712 UTC), (b) DoY 332 (1918 UTC) and (c) DoY 358 (2300 UTC).**

**Comentário [s30]:** Reviser1E17

[Figure]

**Figure 5: Composite the GPS-PWV time series 60 minutes before and 60 minutes after of 18 precipitation events listed by Table 2 and the distribution of the time lag between the maximum GPS-PWV and the time of the maximum rainfall of these events (precipitation fraction above 35 mm h⁻¹ in the area of 4.4 km per 4.4 km centered on the GPS).**

Comentário [s31]: Reviser1E17

[Figure]

**Figure 6: GPS-PWV time series at IEAV station (a) and wavelet power spectrum analysis (b). The cone of influence is plotted in green dashed line to avoid interpretations in border regions.** The precipitation intensity values observed by XPol radar (95[th] percentile) in the area of 4.4 km per 4.4 km around the GPS antenna are included in the bottom of this plot.

[Figure]

**Figure 7: Similar to Fig. 6 for a shorter period (from 0000 UTC on DoY 340 to 0000 UTC on DoY 344) to emphasize the details in the wavelet power spectrum during the occurrence of precipitation events. This period is inside of the cone of influence. A vertical line was drawn for each precipitation maximum to make easier the analysis.**

Comentário [s32]: Reviser1E12

[Figure]

**Figure 8: Wavelet correlation values between GPS-PWV and 95th percentile of the precipitation intensity from XPol radar time series as a function of the different wavelet scales represented by their respective time periods. The correlation is estimated considering the lag zero.. The 95% Confidence Interval for each WCC is estimated considering a Gaussian Distribution after applying the Fisher`s Z Transformation (Whitcher et al. 2000).**

Comentário [s33]: Reviser1E14

[Figure]

**Figure 9: Wavelet correlation values between GPS-PWV and precipitation time series in different wavelet scales (represented by their respective time periods) for (a) percentage of points above 20 mm h$^{-1}$ observed by radar around the GPS antenna; (b) the same above 35 mm h$^{-1}$; (c) and the same above 50 mm h-1. The correlation is estimated considering the lag zero. The 95% Confidence Interval for each WCC is estimated considering a Gaussian Distribution after applying the Fisher`s Z Transformation (Whitcher et al. 2000).**

Comentário [s34]: Reviser1E14

[Figure]

**Figure 10: Wavelet cross-correlation values between GPS-PWV and (a) percentage of points above 20 mm h⁻¹ observed by radar around the GPS antenna; (b) the same above 35 mm h⁻¹; (c) and the same above 50 mm h-1; as function of lead-lag for different time periods. The dotted line represents the 95% confidence interval.**

[Figure]

5   **Figure 11: Spearman correlation histograms of maximum positive (a) and minimum negative (b) correlations as functions of the lag of occurrence for GPS-PWV values and precipitation events. All 18 precipitation events listed in Table 2 were taken into account in this analysis.**

[Figure]

**Figure 12: Frequency polygons of the GPS-PWV derivatives calculated over the period of 60 minutes before precipitation events for different terciles of the precipitation fraction above 35 mm h$^{-1}$ observed using XPol radar.**

[Figure]

**Figure 13: Time series of GPS-PWV derivative before precipitation for upper terciles. The mean values of GPS-PWV derivative, precipitation and GPS-PWV for this events are also shown in this figure.**

---

## Author Comment (AC2) · 25 Mar 2017

**Response latter for Referee 3**

**General comments from Referee#3**

This manuscript presents an analysis of time series of PWV from a GPS station and of precipitation from a nearby X-band radar for a period of 56 days in Brazil. The characteristics of PWV variations before and after the peak in precipitation rate are described with wavelet analysis and more classical tools (Spearman correlation coefficient and histograms). The discussion is focused on the rapid variation of PWV preceding the

peak in precipitation (increase followed by decrease). Though this feature is quite well known, both for the Amazon and for other regions of the world, the main novelty I see from this work is the fact that the increasing phase in PWV can be built upon successive pulses as suggested by Fig. 7. This subtle modulation of PWV was detected in this study thanks to the high temporal resolution of the PWV time series (1-minute). The potential for predicting the occurrence of intense precipitation with the help of real-time GPS PWV data is also suggested but not demonstrated. The topic of this study is interesting and might contribute to a better understanding of the moist processes involved in deep convection over land in the tropics (though such physical interpretation is beyond the scope of this paper). However, in its present form, this manuscript cannot be accepted for publication. A major rework of the data analysis and presentation is required. Major and specific comments are given below.

Response: Taking into count the comments, corrections and suggestions from the Reviser, the new version of the manuscript was significantly changed, particularly in the analysis of the results: discussion about the GPS-PWV jump occurrence before precipitation, wavelet, and derivative analysis. A new data processing of the time series had to be done to include 4 new figures in the text. We agree that these figures are very important for better discussing the results and to evidence some obtained results, which were not suitably emphasized in the previous version. The additional analyses improved substantially the scientific quality of the results. Thanks for these very interesting suggestions. About the specific items presented by Reviser, some aspects of this revision deserve some general comments to explain the changes done in the new version of manuscript. These general comments are presented here:

- (I) Data processing and interpretation of results:

- The relationship between GPS-PWV jump and PWV pulses was clarified in the new version of the manuscript and some inconsistencies were removed. The jump is composed by several pulses; the jump is evidenced in a scale that represents a time period of 32-64 minutes and the pulse is shorter and more variable.

- Some inconsistencies about the intensity and intensity/extension of the precipitation events were removed in several parts of the manuscript.

- The most important aspects associated to GPS data process, particularly those associated with ZTD estimates, were better detailed. Several information about the configuration adopted to use GOA-II software, such as information about the mapping function and Stochastic model for tropospheric delay (Random walk method) and its constrains were included.

- (II) Wavelet analysis:

- The wavelet analysis was revised and we included the wavelet correlations for different lags, which was very important to emphasize the lag of 30 minutes and show the relevance of interrelations between the time series in scales related to time periods smaller than 32 minutes where the PWV pulses are contained.

- We improved and reorganized considerably the introductory section about the wavelet, changing some notation to make the text easier to readers, excluding some technical information, including some synonymous, and explanation about the evaluation and choice of different wavelet mothers.

- Results of significance tests, such as the cone of influence, was included in the wavelet power Spectrum in the new version of manuscript.

- (III) Result analysis:

- Based on your specific comments about the GPS-PWV jump discussion, two new figures were organized and included in the manuscript. The Fig. 4 shows other GPS-PWV jumps observed before different extension rain events and in Fig. 5 the composite

of time series was done in order to highlight the most prominent features of the PWV jumps, as suggested. This composite was normalized by maximum GPS-PWV values before maximum precipitation and we agree that this figure was very suitable to emphasize the jump occurrence and the time scale that it is more frequent.

- A histogram of the time lag between the maximum GPS-PWV and the time of the maximum rainfall is presented and clearly shows that there is a range of lag time between these two maximums. Certainly, the precipitation area and the temporal resolution of GPS-PWV here employed have an important impact on this lag time. For instance, Adams et al (2013) using rain gauge and PWV with sampling rate of 30 minutes found out lag zero, however, in this study, we show that there are PWV oscillation in higher frequency than 30 minutes, which can represent an area around GPS antenna and not only a punctual measurement.

- The GPS-PWV derivative time series before severe precipitation of the upper tercile was done and included in the manuscript, which allows the analysis of the occurrence of stronger PWV derivatives and associating them to PWV pulses.

- (IV) GPS-PWV potential for predicting the occurrence of precipitation

- The present study has the aim to show the GPS-PWV jump, to discuss the time variability, the jump as function of the rainfall intensity and discuss the possible associated physical mechanism. As the number of precipitation events used in this study is not suitable to show the skills of the model, we opted in not presenting the suggested short hindcast. Developing this model is a proposal for future article (in development), which should be involved a larger dataset with independent events to building and testing appropriately the model. Besides, the data processing used here are not in real time and there are some aspects that deserve attention in sensibility experiments (mentioned by revisers) such as: the size of the time-windows before severe precipitation, the constrains of temporal evolution of the ZTD estimation, precipitation area around the GPS

antenna, and others.

Others minor corrections pointed by reviser, such as inconsistencies, use of not conventional scientific terminology, and technical corrections, were taken in count in the elaboration of the new version of the manuscript. Specific comments point-by-point are presented below. The numbers (in the new version of the manuscript) of the changed lines and respective page in each point are listed and highlighted in green. The numbers of figure in the response refer to 13 figures of the new version of the manuscript.

**Specific comments**

**(I) Wavelet analysis:**

1) **Referee#3:** The wavelet correlation method and results are not properly used and interpreted. Several times the authors interpret the period bands for which the wavelet correlation values are displayed as lag times (P2L29, P620, P6L29, P9L19). For example (P6L29): "an interrelation analysis can be performed with the WCC of GPS-PWV and precipitation time series to evaluate the correlation at different lags in time, scale by scale". The results provided in figure 5 and 6 are correlation values for a range of time periods indicated in the x-axis and wrongly labelled "wavelet scale". This kind of plot is typically given for a zero-lag (see e.g., Whitcher et al., 2000). Where are the wavelet correlations for different lags? The analysis could be complemented with wavelet cross-correlations as a function of lag for different scales such as in Whitcher et al., 2000. Proper use of wavelet cross-correlations for lead/lag analysis would probably not require using the classical Spearman correlation (Fig. 8).

**P11L28-P11L32 Fig.10 Response:** We agree that explore wavelet cross-correlations as a function of lag for different scales is useful for analysis of GPS-PWV jumps. We included the wavelet correlations for different lags complementing the graphics of the Fig 8 and 9. The wavelet cross-correlations (WCC) as a function of lag for different scales are presented in Fig 10. We corrected the x-label, where we preferred to keep the time periods instead of wavelet scales. We also improved the information in the Caption of Fig 8 and 9. The inconsistencies in the text were corrected.

2) **Referee#3:** The significance of wavelet spectra (Fig. 3 and 4) and correlations (Fig. 5 and 6) is not addressed, though mention to "significant" frequencies is made in several places in the text. Most tools used for wavelet analysis provide also results of significance tests or confidence intervals. These should be used to highlight objectively the significant frequencies.

**Caption Fig. 6 Fig.7 Response:** Unfortunately this information was missing, which was included in this new version of the manuscript. The cone of influence was included in the wavelet power spectrum of the Fig. 6 and 7. The error bars in Fig. 8 and 9 represents the confidence interval, which means if the zero is outside the interval, the WCC is statistically significant. Some information about the statistic significance of the results are presented in the text and in the caption of these figures.

3) **Referee#3:** It is not said what the error bars in Fig. 5 and 6 represent.

**P9L31-P33 Fig. 8 Fig.9 Response:** Information about the error bar were included in the Caption of this figure, as follows: "The 95% Confidence Interval for each WCC is estimated considering a Gaussian Distribution after applying the Fisher's Z Transformation (Whitcher et al 2000)".

4) **Referee#3:** The description of wavelet analysis methodology in section 3 is not sound. Only fragmented information is given on the used methods, without justification of the choices (e.g. of mother wavelets, continuous vs. discrete wavelet transforms). On the other hand very highly-technical information is given (e.g. use of non-decimated discrete wavelet transform, use of pyramidal algorithm...) which won't be explicit to the general readers of AMT journal. So I recommend that this section explains a bit more about the general principles and choices that were made, and the implication of the choices that are made. For example, one might question about the robustness of the results regarding the choice of mother wavelets. For the more technical aspects, refer to the proper literature where appropriate.

**P9L7-P33 Response:** We improved and reorganized considerably the introductory section about the wavelet, changing some notation to make the text easier to readers of Whitcher et al(2000). We excluded some technical information and included the synonymous of non-decimated discrete wavelet transform (NDWT): Maximal Overlap DWT (MODWT) and, instead of Symlet we used the synonymous Least Asymmetric wavelet with the number of the filter length. We included briefly a comment about the evaluation and choice of different wavelet mothers. Some information about the choice of the discrete wavelets was discussed showing the evaluated wavelets, as well as which one was selected and the consequences of this choice.

**(II) Analysis of data and interpretation of results**

5) **Referee#3:** The GPS data processing options should be more detailed (see specific comments below). Some fundamental aspects are missing, such as the constraints of the stochastic model of the ZTD parameters. They impact directly the magnitude of variability in the retrieved PWV time series.

**P5L1-L3 Table 1 Response:** We agree that the most important aspects associated

to GPS data process not had been presented suitably in the previous version of the manuscript, particularly those associated with ZTD estimates. The section "2.2 High temporal resolution GPS-PWV time series" was drastically changed with the inclusion of the several information in the configuration adopted to use GOA-II software, such as information about the mapping function, ZHD and ZWD values used *a priori*; Stochastic model for tropospheric delay (Random walk method) and its constrains. These information were organized in Table 1.

6) **Referee#3:** The dimensions of the area around the GPS station used to study the PWV-precipitation relationship is said to be a "key factor" (P5L5). In consequence, some preliminary tests should be presented which give insight into the sensitivity of results to this parameter. It is not clear how the representativeness (P5L11) is estimated.

**P5L28-P6L7 Response:** The rainfall area employed in this study is only a reference for the description of GPS-PWV jump. For instance, if a raingauge is employed to only an area of 20 cm radius is recorded and it is considered represents the rainfall from a large region. From another side, if the whole radar area is employed a rainfall over 100 km radius is recorded and could be associated to the GPS-PWV. In both cases representativeness and the lead time (for a nowcasting application) should be considered. In the raingauge case, rainfall will be underrepresented because only rainfall over the raingauge will be considered and for the radar case rainfall far from the GPS-PWV will be considered and cannot be associated to the local increase in PWV. The representativeness of the GPS measurement is still an open question because it depends on the vertical distribution of the water vapor and mainly from the combination of elevation angles of the GPS satellite and the elevation angle threshold employed in the PWV processing. Actually, the area employed was the one having the best correlation with PWV and nearly the expected representativeness of the GPS receiver for this location, but it is expected to vary as function of the region and satellite

configuration. This information was included in the manuscript.

7) **Referee#3:** Fig. 7 shows one case of heavy precipitation where PWV shows a strong peak achieved in several steps (so-called "pulses" in the text and inconsistently labelled "jumps" in the Figure). This kind of analysis should be made for other cases to establish if the pulses are a robust feature of the heavy precipitation events. It is not clear if the authors associate the 32-64 min periodicities detected by the WCC with these pulses? It would also be interesting to add a composite of time series to highlight the most prominent features of the PWV jumps. The authors refer extensively to Adams et al. (2013) regarding a maximum of PWV 1-h before the maximum of precipitation. Inspecting this reference carefully reveals that on average the peak in PWV is rather coincident with the peak in precipitation (Fig. 2 in Adams et al., 2013) or slightly ahead (Fig. 3 and 4), in accordance with other results (e.g. Holloway and Neelin, 2010). The case illustrated in Fig. 7 shows indeed a lead time of 1 hour but about 50% of the cases show lead times between 0 and 30 min (Fig. 8). So the 1-h lead time should not be considered as a general rule.

**P7L1-P8L24 Fig.4 Fig5 Response:** The relationship between jump and pulse was clarified and the mentioned inconsistency (Fig.3) was removed. The jump is composed by several pulses and is better represented in the scale related to the time period of 32-64 minute. The pulse is shorter and more variable. This was included in the manuscript. The Fig. 3 present the indication of pulses and the PWV-GPS jump before the precipitation. Based on this comment, two new figures was included in the manuscript. The new Fig. 4 shows other GPS-PWV jumps observed before different extension rain events and in the Fig. 5 the composite of time series was done in order to highlight the most prominent features of the PWV jumps, as suggested. All the 18 events listed by Table 2 are taken into account in this analysis. The analysis of this results is presented to highlight the hardiness of the GPS-PWV jumps feature before
heavy precipitation events. The results obtained to follow this suggestion was very interesting and improve significantly this section of the manuscript. The time lag between the maximum GPS-PWV and the time of the maximum rainfall is presented in a sub plot in Fig.5. It is important to highlight that the there are many precipitation events of lower intensity and extension in which the GPS-PWV jumps are not observed. In these cases, the maximum GPS-PWV is observed in the maximum precipitation, consequently, the time lag is zero (37% of the cases evaluated). In this histogram, it is clear there is a range of lag time between these two maximum. Certainly, the precipitation area and the temporal resolution of GPS-PWV here employed have an impact on this lag time. For instance, Adams et al (2013) using rain gauge and PWV with sampling rate of 30 minutes find out lag zero, however, in this study we show that the GPS-PWV signal can represent an area and not only a punctual measurement besides the GPS. This discussion was included in the manuscript after present the Fig. 5.

8) **Referee#3:** In section 4.1, show an example of correlation function to help explaining what is meant by the "positive and negative correlations" (in fact maximum positive/minimum negative correlations) and specify the time window of analysis (+/- 1 hour). Again, the histograms in Fig. 8 suggest that the case in Fig. 7 might be a specific case because the minimum correlation is probably reached for a lag time around 0. Illustrating other cases and providing composite plots would help to better catch the general situation. Moreover, Fig. 7 is a case where the pulses preceding the peak in PWV are outside of the correlation window. The link between the pulses and the peak in PWV and in precipitation should be investigated in more detail as this is to my opinion the real innovation of this work.
**P7L22-L26 Fig.11 Response:** The composite of PWV-GPS and precipitation time series and a new figure with other cases of jumps was included. The relation between the water pulses and GPS-PWV jumps was better discussed. The derivatives larger than 9 mm/h obtained from GPS-PWV in high temporal resolution were associated with

this pulses, and these results were appropriately emphasised, following this comment. Based on your comment about the histogram correlation a mistake was identified and corrected. The minimum negative correlation in zero was not been taken in account, which was included and the problem corrected. The Fig.11 was updated. Thanks so much for that. In fact, the case showed in Fig.3 is the special case because it was one of the strongest events registered during the CHUVA Vale experiment, the lag time zero in the minimum negative correlation maybe a consequence of this intense precipitation. A similar result can be observed in Fig.4a, for the precipitation occurred during the 315 DoY. The title of figures of the Spearman correlation histograms were corrected, as suggested.

9) **Referee#3:** In section 4.2, the PWV derivatives are computed in a 1-hour window preceding the peak in precipitation. This window is too small, namely for the case of Fig. 7, as it does not include the part of time series with the pulses. Extend the time window to analyse this feature in more detail and also to not give too much weight to the decreasing phase after the peak in the statistics computed in Table 2. Some sensitivity tests should be made to choose the best window size.
**P13L5-L7 Response:** In this conceptual study, the size of the time-window used to analysis of the PWV derivatives was defined based on wavelet correlation results, which given the clear indication that PWV-GPS oscillation for the scale related to period band between 32 and 64 minutes are associated with precipitation events. The time window used here is 60 minutes. The histogram of the time lag between the maximum GPS-PWV and the time of the maximum rainfall presented in the new figure 5, suggest that this time-window is suitable for this data set. We agree that others time-windows should be tested in sensitivity experiment, particularly when a large data set be used to developing a nowcasting tool based on GPS-PWV values. In this future study probably this time window can be adaptive, which should be larger depend on intensity and frequency of the stronger PWV positive derivative (associated with water

vapor pulses), as before precipitation events of the Fig. 3. The reasons about the size of the time-window used in this study were included in the manuscript.

10) **Referee#3:** I suggest to add also a figure with the time series of PWV derivatives for all 6 cases of the upper tercile (e.g. superposed on one graph showing also the average PWV derivative and precipitation).
**P14L1-L5 Fig.13 Response:** The suggested figure was done and included in the manuscript (new Fig. 13), which improved the analysis of GPS-PWV derivative before severe precipitation and allowed the analysis of the occurrence of the stronger PWV derivative and associating them to PWV pulses.

11) **Referee#3:** The sentence P2L29 announces a "study of correlation and lags with rainfall events to form a conceptual model with predictive capacity that can hence be used in developing a nowcasting tool for strong precipitation events". However, section 4.3 doesn't provide any proof of concept of this model. I suggest a short hindcast study is performed, based on the data from this campaign, to evaluate the skill of the proposed detection method.
**P3L15 Response:** The number of precipitation events used in this study are not suitable to show the skill of the model. This phrase was rewritten to avoid the false expectancy in the reader. The new phrase is "... capacity, which can be useful in nowcasting tool for strong precipitation events." Developing this model is a proposal for future article (in development), which involve a larger dataset with independent events to build and test the model. All precipitation events from several CHUVA campaigns is being explored. The studied model should be adjusted for different precipitation regime to obtain the acceptable skill. The present study has the aim to show the GPS-PWV jump, to discuss the time variability, the jump as function of the rainfall intensity and discuss the possible associated physical mechanism.

12) **Referee#3:** Some interpretation is made about the underlying physical processes leading to the observed PWV jumps:

- (1) P8L25-29: "The presence of low-level water vapor convergence can be attributed to mechanisms such as gravity wave forcing (Raymond 1987) or other larger-scale forcing mechanisms. The increased water vapor convergence (i.e., positive PWV derivatives) may also simply be a reflection of the unstable surface parcels accelerating upwards, thereby vertically advecting a larger surface specific humidity to higher levels in the atmosphere without necessarily any larger-scale dynamical forcing"

- (2) P10L31: "This increase in the stronger negative derivative frequency before more intense events (upper and middle terciles) is associated with the conversion process from water vapor to liquid water…"

- (3) P11L5: 'the processes responsible for the maintenance of precipitable water suspended in the atmosphere are very complex and highly nonlinear"

- (4) P11L8: "The increase in moisture convergence appears to be due to an increase in the frequency of convergence pulses."

These assertions relate either to very general atmospheric processes or are highly speculative as recognized by the authors, P8L29, "Given the limitations of the observations, our interpretation of the physical mechanisms responsible for the jumps remains speculative." I suggest to remove sentences which cannot be supported by other studies of similar phenomena or provide the necessary proofs, e.g. by analysing additional data.

**Response:** We agree some of these sentences are speculative, but we also should open the discussion to possible physical effects related to this feature and at the same be precise. We changed the above sentences as:

- (1) P8L5L7 The sequence of pulses of positive increases in the PWV could be a result of several physical processes. Some of the physical process that can explain these pulses could be low-level water vapor convergence forced by gravity wave (Raymond 1987) or simply unstable surface parcels accelerating upwards... P8L22L24 To understand what the physical mechanisms responsible for these pulses are a specific field campaigns design are needed.

- (2) P13L22L24 One possible reason for this increase in the stronger negative derivative frequency before more intense events (upper and middle terciles) is the conversion process from water vapor to liquid water.

- (3) P14L8-L9 Therefore the use of only a maximum threshold from a GPS-PWV derivative, as suggested by Iwabuchi et al. (2006), is not sufficient to predict intense rainfall , several atmosphere processes are very complex and highly non-linear.

- (4) P14L11 was deleted.

**(III) Inappropriate and vague terminology**

13) **Referee#3:** The PWV jumps which are the main topic of this work (as advertised in the title) are defined inconsistently in several places in the manuscript: P2L28 ("sharp increase in PWV"), P8L22 (a "pattern" of maximal PWV), P9L14 ("oscillations"), and Fig. 7 (pulses defined P8L25 are labelled "jumps"). Please be consistent. P7L3-L8 P12L6 P15L3-L7 Fig.3 **Response:** The appointed inconsistency was removed. The definition used in Abstract and introduction using the term "sharp increase" was used in the section 3 (replacing "pattern") and conclusion (replacing "oscillation")

to describing correctly the GPS-PWV jump. The sentence mentioned, which the term "oscillation (jumps)" appear is not a definition. It was rewritten to maintain the consistency. The Fig. 3 the mistake about the label "jumps" instead of "pulses" was corrected, as well suggested. A label "GPS-PWV jump" was included in this figure.

14) **Referee#3:** Precipitation intensity is measured as the 95th percentile of all precipitation data in the area (P5L22). However, later, precipitation fractions are referred to as precipitation intensity (Table 1 and 2). Intensity terciles are referred to in section 4.2 and 5. Please be consistent.
**P1L18-L20 P7L16 and many others Table 2 3 Response:** The reviser is right, some inconsistencies about the intensity and intensity/extension of the precipitation events were present in several part of the previous version of the manuscript, which were removed. The term "precipitation intensity" is used for analysis of 95% percentiles of precipitation time series. The term "precipitation intesity/extension" is used for analysis of rain fraction above different thresholds of precipitation, for example wavelet cross correlation analysis. The term "precipitation extension" is used for analysis of percentage of precipitation above of one specific selected thresholds, for example: precipitation extension terciles explored in the derivative analysis (table 2 and table 3), which are defined in function of the precipitation fractions above 35 mm h$^{-1}$.

15) **Referee#3:** It is not clear if the precipitation fractions are computed from the 95th percentile of precipitation or from all data and grid points?
**P6L21 Response:** The precipitation fraction are computed as the fraction of the area of 4.4 km per 4.4 km around the GPS antenna with precipitation rates above some chosen threshold. This phrase was rewritten in the manuscript.
16) **Referee#3:** Expressions such as "expressive", "expressive changes", "evident", "more evident" are not adequate. Please use conventional scientific terminology.
**Response:** As also suggested by reviser 1, all occurrences for the terms "expressive" were replaced by other more suitable (see Specific comment 11-Referee#1). The term evident were removed from manuscript using the following word (underline):
P10L27 "... these cases makes the result discussed for Fig. 6 more understandable"
P12L14 "...the correlation between the GPS-PWV and precipitation time series is more higher."
P13L20 "Notably, this effect is observed for the middle and uppe..."
P15L11 "...which is more stronger for events of large intensity and extension."

17) **Referee#3:** What is a positive oscillation? (P8L18, P12L3. . .) or a positive increase?
**P15L3 Response:** The term "predominantly positive" was excluded of phrase in the section 4 to avoid the inconsistency pointed by reviser. In conclusion section the phrase was rewritten associating with PWV derivatives, which are predominantly positive before intense precipitation.

**Technical corrections**

18) **Referee#3:** P4L7: specify which "possible noise sources" are taken into account and how.
**P4L23-L25 Response:** This phrase was rewritten in the new structure of this section and the term "possible noise sources" was excluded because is impossible eliminate the "possible noise sources" in the geophysics process. We would like to highlight in this phase that in the GPS data processing used to generate PWV time series with high temporal resolution is necessary to taking into consideration several known uncertainty

sources in order to minimize them. This phrase was rewritten in new structure of this section, which this rigorous data-processing was demonstrated with details through the items listed and suggested by reviser in his specific comment (5-Referee#3).

19) **Referee#3:** I understand that you used the regional model for Tm of Sapucci 2014. So it is not the Bevis 1992 relationship that is used (P4L13). Please correct.
**P5L5-L9 Response:** The methodology used to converting ZTD into PWV is from Bevis et al (1992), but the TM values were calculated using the regional model suggested by Sapucci (2014). This was clarified in the manuscript.

20) **Referee#3:** This Tm model requires RH. It is not said what RH data are used to compute Tm.
**P5L5-L9 Response:** The TM model suggested by Sapucci (2014) for CHUVA-Vale region (Subtropical ocean) are not requires RH values. Only temperature and pressure values are required in this case. The RH values are required in other region such as subtropical continent, South and Northeast (Sapucci, 2014).

21) **Referee#3:** P4L17: "3.1% of data are missing in 2 days". This represents about 1.7 days for the total period of 56 days. I suggest that you completely remove these 2 days from the analysis because the spline interpolation won't give correct results.
**P5L12-L14 Response:** In fact, these period were removed from analysis results. The results from spline interpolation were used only in wavelet analysis, which require time series without failures. The phrase was rewritten in the manuscript.

22) **Referee#3:** It is important to specify the tropospheric model used in the GPS data processing: is it a random walk? Are both ZWD and gradients estimated? What are the constraints of temporal evolution of these two parameters? Did you make some tests with different constraints? (the analysis of 6-min time differences in section 4.2 might be strongly impacted if the constraints parameters are too small or too big).
P5L1-L3 P14L18-L20 Table 1 Response: In the new section 2.2 the information about the stochastic model used in the data processing were presented, as well as information about azimuthal gradient estimated. The constrains of temporal evolution of these parameters used were the default value suggested by JPL: 8.333E-8 (km per square-root second) and 8.333E-9 (km per square-root second), respectively. We agree that this values might impact the variability of the PWV estimates in high frequency, and others values should be tested ahead in this research. This information was included in the manuscript.

23) **Referee#3:** P6L8-10 and in many other places, results from Adams et al., 2013, should be rather described in the introduction.
P3L11-L13 Response: Adams at al. 2013 in this paragraph and other places were moved to introduction section, which was better described, as also suggested by reviser 1 (9-Referee#1) and reviser 2 (16-Referee#2).

24) **Referee#3:** P7L17: what do you mean by "amplification"?
P10L25 Response: The Fig. 7 is a aggrandizement or enlargement of the FIg.6 in short specific period. The term was replaced by "enlargement" in the new version of manuscript.

25) **Referee#3:** P7L31: why is quasi-symmetry ("a good property for the mother wavelet") important to this study?

**P11L5 Response:** One important advantage of the quasi-symmetry is related to the reconstruction. As this is not the case for this paper, this information was omitted.

26) **Referee#3:** P8L19:20: "maximum peak" replace with "maximum"
**P7L7 Response:** Replaced as suggested.

27) **Referee#3:** P10L12: "the lowest threshold" complete with "among the 3 that were tested".
**P13L8 Response:** Completed as suggested.

28) **Referee#3:** P10L16: what are rainfall events with periods without precipitation?
**P13L11-L13 Response:** In the statistic analysis from GPS-PWV derivative besides the three threshold of precipitation intensity, the derivative histogram was calculated for the other rainfall events with precipitation intensities smaller than 20 mm h-1 and periods without precipitation. This histogram is called by "other cases" in Figure 9. This is better described in the new version of manuscript.

29) **Referee#3:** P11L11-19: description of results from Fig. 9 should go to section 4.2
**P13L30-P14L1 Response:** The partial analysis of Fig. 12 presented in the section 4.3 was moved to section 4.2, as suggested.

30) **Referee#3:** P11L21-25: results from past studies about nowcasting should go to the introduction. Note that these references can probably be updated.
**P14L22-L30 Response:** In this study the PWV estimates used are post processed and

the following question naturally appear: Are PWV estimates in real time able to capture the jumps before the precipitation observed? These previous studies shows that the quality of PWV obtained in real time is similar the post-processed, which are mentioned in this context. These references was updated with the inclusion of the results reported by Shi et al (2015) (doi: 10.1109/TGRS.2014.2377041).

Please also note the supplement to this comment:
http://www.atmos-meas-tech-discuss.net/amt-2016-378/amt-2016-378-AC2-supplement.pdf

**Supplement:**

**GPS-PWV** jumps before intense rain events**

Luiz F. Sapucci1, Luiz A. T. Machado1, Eniuce Menezes de Souza2, Thamiris B. Campos3

1Centro de Previsão de Tempo e Estudos Climáticos, Instituto Nacional de Pesquisas Espaciais, Cachoeira Paulista, Postal Code: 12630-000, Brazil.

2Departamento de Estatística - Universidade Estadual de Maringá, Maringá, Postal Code: 87020-900, Brazil 3Programa de Pós-Graduação em Meteorologia, Instituto Nacional de Pesquisas Espaciais, São José dos Campos, Postal Code: 12227-010, Brazil

Correspondence to: Luiz Sapucci (luiz.sapucci@cptec.inpe.br)

- 10 Abstract. A rapid increase in atmospheric water vapor is a fundamental ingredient for many intense rainfall events. Highfrequency precipitable water vapor (PWV) estimates (one minute) from a Global Positioning System meteorological site (GPS) are evaluated here for intense rainfall events during the CHUVA Vale field campaign in Brazil (November-December 2011), in which precipitation events of differing intensities and spatial dimensions, as observed by an X-band radar, have been explored. A sharp increase in the GPS-PWV prior to the more intense events has been found and termed GPS-PWV
- 15 "jumps". These jumps are associated with water vapor convergence and the continued formation of cloud condensate and precipitation particles. The correlation and lags between the high temporal resolution GPS-PWV time series and rainfall events are evaluated. A wavelet cross-correlation analysis shows that there are important spikes in the PWV that precede the more intensity/extension rainfall events on scales related to time periods from about 30 to 60 minutes. The GPS-PWV timederivative histogram for the period of 60 minutes before the rainfall event reveals different distributions for higher intensity
- 20 and extension events. This feature could indicate the occurrence of severe precipitation and consequently has the potential for application in nowcasting activities.

**1** Introduction**

The application of the Global Positioning System (GPS) tropospheric-induced signal delay to estimate the precipitable water vapor (hereafter, GPS-PWV) is a good example of an indirect solution for quantifying atmospheric humidity. The magnitude

- 25 of this delay is related to the integral of the refractivity index of the air as a function of temperature, pressure and water vapor (Bevis et al. 1992) on the optical path followed by the GNSS signal. The wet component of this delay provides the precipitable water vapor (PWV) (Bevis et al. 1994), with an error of approximately 5% under all weather conditions (Wolfe and Gutman 2000) relative to other measurement techniques (Sapucci et al. 2007) and in near real time (Rocken et al. 1994). The methodology employed in GPS data processing has been improvement continually to minimize the uncertainty and the
- 30 PWV estimate has been determined with an accuracy better than 2 mm (Moore et al. 2015; Shangguan et al. 2015) Although the vertical humidity structure is not captured in GPS-PWV estimates, the great advantage of GPS-PWV (in addition to its

all-weather capacity) is its high temporal resolution (minutes) (Zumberge et al. 1997). An important application of the GPS-PWV estimate is its assimilation into the Numerical Weather Prediction process, which has a positive impact on short-range forecasts of humidity fields and, consequently, better precipitation forecasts for heavy rainfall events (Cucurull et al. 2004; Bennitt and Jupp 2012). Other applications become viable in dense networks and transects. GPS-PWV has been useful for:

- 5 studying the diurnal cycle of convective instability in Japan (Sato and Kimura 2005); investigating water vapor variability during a mistral/sea breeze event in southeastern France exploring GPS-PWV tomography (Bastin et al. 2005); studying water vapor diurnal cycle over African continent (Bock et al. 2008); evaluation of the GPS-PWV values before precipitation (Kursinski et al. 2008); tracking water vapor advection over Amazonian (Adams et al. 2011); GPS-PWV tomography have been used to investigate the water vapor distribution related to convective rainfall events and better understanding and
- 10 quantification of the hydrological cycle in southeastern France (Brenot et al. 2014); it has been used to propagate convective events and determining topographic effects on the evolution of convection (Adams et al. 2015); studies of convective mesoscale events during the North American Monsoon (Serra et al. 2016) and nowcasting thunderstorm activities, as foreseen by Jerrett and Nash (2001). Guerova et al. (2016) showed development and test of new multi GNSS (Global Navigation Satellite System) products to forecasting of severe weather and emphasized the fact that for short-term, high-15 resolution forecasting or nowcasting models require more detailed humidity observation, e.g. GPS-PWV estimates.
- The relationship between the occurrence of intense rainfall and high concentrations of atmospheric water vapor is well known and has been explored in nowcasting applications. Muller et al. (2009) described a model for the relationship between the PWV and convective precipitation event in the tropics. PWV data from a microwave radiometer (MWR) with high temporal resolution have been used to analyse and better understand the temporal relations of column water vapor and
- 20 tropical precipitation (Holloway and Neelin, 2010). Chan (2009) evaluated the performance of a ground-based microwave radiometer in intense convective weather and reported an increasing degree of instability of the troposphere before the occurrence of heavy rain. In addition, he compared PWV from GPS and radiometer, the result showed that radiometer exhibited rather rapid fluctuations during intense precipitation, which are not observed in the GPS-PWV data. Madhulatha et al. (2013) reported a sharp increase in the PWV values approximately 2–4 hours prior to the occurrence of thunderstorms
- 25 and developed a nowcasting technique using PWV values and 7 other thermodynamic indices from microwave radiometer observations. The relationship between deep convective activity and occurrence of intense rainfall and the temporal evolution of GPS-PWV has been studied for well over a decade. Mazany et al. (2002) developed a lightning prediction index for Florida based on the GPS-PWV magnitude and its temporal evolution. This index is a binary logistic regression model based PWV-GPS and other two variable predictors. The plot of the GPS lightning index time series showed a pattern several
- 30 hours prior to a lightning, which was tested as forecasting tool of this events. Nowcasting employing GPS-PWV was reported by de Haan et al. (2004), who demonstrated its viability in improving thunderstorm and heavy precipitation forecasts during a cold front passage. De Haan (2006) developed a method for inferring the atmospheric stability from a nonisotropic GPS path-delay signal (slant delay). Book et al. (2008) carried out studies of the West African Monsoon using

ground-based GPS receivers and demonstrated the correlation between PWV and precipitation in seasonal and intraseasonal timescales. More recently, in the tropics, Adams et al. (2013) utilized 3.5 years of Amazon GPS-PWV to derive a water vapor convergence time scale associated with the shallow-to-deep convective transition. Given this one-hour lead time, near real-time GPS-PWV could, in principle, be employed to improve the nowcasting of these events. This study shown the PWV increase prior to heavy deep convective precipitation principally results from water vapor convergence, given that the surface evaporation is small in the cloudy, showery conditions preceding the event.

The motivation of the present study is to evaluate the large and rapid increases in the PWV characteristics prior to the onset of deep convective rainfall events observed during the CHUVA (Cloud processes of the main precipitation systems in Brazil: a contribution to cloud resolVing modeling and to the GPM (GlobAl Precipitation Measurement)) Vale experiment

- 10 in Brazil in 2011 (Machado et al. 2014) using PWV estimates with high temporal resolution (one minute) from a GPS meteorological site. Adams et al. (2013) showed that prior to deep convective events in the central Amazon, a 4-hour "ramp up" in the time derivative of GPS-PWV is observed, reaching a maximum approximately one hour before heavy precipitation. This sharp increase in the GPS-PWV values before the occurrence of more intense rainfall events is hereafter termed the GPS-PWV jump. The GPS-PWV time series was evaluated using wavelet analysis in a study of correlation and
- 15 lags with rainfall events to form a conceptual model with predictive capacity, which can be useful in nowcasting tool for strong precipitation events.

In section 2, we present the data collected during the CHUVA Vale experiment in Brazil and describe the GPS data processing technique for obtaining PWV values. An X-band radar was utilized to quantify the spatial-temporal distribution of the precipitation events. In section 3, the GPS-PWV jumps are defined and characterized. Section 4 presents the high

20 temporal resolution GPS-PWV time series analysis, including wavelet analysis and time lag correlations between the precipitation and GPS-PWV time series, as well as the evaluation of GPS-PWV derivatives before precipitation events of different extensions. Finally, Section 5 presents the conclusions.

**2 Data collection design and processing method**

5

The data employed in this study were collected during the CHUVA Vale campaign, one of the field campaigns of the 25 CHUVA project (Machado et al. 2014). The CHUVA project was designed to investigate cloud microphysical and precipitation processes through six intensive field campaigns covering various precipitation regimes in Brazil. The CHUVA Vale campaign was carried out in São José dos Campos City in São Paulo State (23° 12' 30" S and 45° 57' 08" W) in an elevated valley between the Mantiqueira and Serra do Mar mountain ranges. Fig. 1 shows the geographic location of the CHUVA Vale campaign. This region is dominated by deep convection with typical rainfall systems that are forced by sea

30 breeze-mountain convergence zones as well as squall lines associated with cold front penetration (Machado et al. 2014).

**Comentário [s1]: Reviser3E11**

**2.1 Data from the CHUVA Vale experiment**

The CHUVA Vale experiment consisted of an intensive observation period from November 3rd to December 28th of 2011. The instruments used in this study were a dual-frequency GPS receiver for scientific applications, a disdrometer and a mobile X-band dual polarization radar (XPol). The TRIMBLE brand GPS receiver, model NETR8 with Dorne-Margolin

- 5 Choke Ring antenna, utilized in this study was installed 11 km from the XPol radar at the site denominated the Institute of Advanced Studies from the Department of Aerospace Science and Technology (IEAV) site. Any sky obstructions around the GPS receiver were avoided to minimize the multi-path effect in GPS signal propagation. See Fig. 1 for more details about the sites, at which the instruments were placed.
- The GPS satellite signals were sampled at one-second frequencies, whereas the collocated meteorological sensor captured 10 the pressure and temperature at one-minute frequencies. A Joss-Waldvogel brand acoustic impact disdrometer (Joss and Waldvogel 1967), model RD 80, was installed a few meters from the GPS receiver. The XPol radar scan strategy collected one volume scan every 6 minutes at 13 elevations, from 1° to 25°, with 1° and 150 m angular and radial resolutions, respectively. Due to a technical problem the radar was turn off in the period since 12:36 UTC of 14th November until 19:41 UTC of 15th November of 2011. The radar data were pre-processed using the attenuation correction of the reflectivity,
- 15 employing the algorithm for ground-based polarimetric radars (ZPHI algorithm) proposed by Testud et al. (2000). For a detailed description of the radar and disdrometer pre-processing, see Calheiros and Machado (2014).

**2.2 High temporal resolution GPS-PWV time series**

20

The zenith total delay (ZTD) was obtained by processing the GPS data using GOA-II [(Gipsy, GPS Inferred Positioning System) (OASIS, Orbit Analysis and Simulation Software II) Gregorius 1996] software by applying the precise point positioning method in post-processing mode with the precise ephemerides of the GPS constellation provided by the NASA Jet Propulsion Laboratory. Sampling rate of the used GPS satellites ephemeris is 15 minutes for orbits and 5 minutes for

GPS satellite crock. To ensure the quality of the PWV time series with high temporal resolution required in this study, in the data-processing strategy adopted the known uncertainty sources were taken into consideration applying the recommended models and

- 25 adjustment of parameter exploring available stochastic models. The latest version of the GOA-II software (version 6.3) was used, which estimates parameter with high temporal resolution exploring the sophisticated orbit integrator package to estimate GPS satellite position in each epoch. The ocean tide model FES 2004 (Lyard et al. 2006), recommended by the International Earth Rotation and Reference Systems Service (IERS Conventions 2010) was applied in this processing. Method of antenna absolute calibration (Schmid et al. 2007) was applied by GOA-II to ensure the correct phase center
- 30 variation of the satellites and receiver antennas using parameters provide by IGS web site (Montenbruck et al. 2015). The data processing with GOA-II software to obtain ZTD estimates was done selecting the Global Mapping Function (Boehm et al. 2006) and the sampling rate of the ZTD estimates of 60 seconds. The others possible parameters that can be selected were

Comentário [s2]: Reviser3E18

used the configuration basic suggested by JPL. As the configuration items associated with ZTD estimates and respective values used can impact in the variability of the ZTD in high temporal resolution (basic information used in this study), they are listed in Table 1 in order to highlight them.

The zenith wet delay was obtained from the ZTD after removing the zenith hydrostatic delay obtained through the application of a representative tropospheric temperature model and a surface pressure measurement (Davis et al. 1985). The zenith wet delay was converted into the PWV using the relationship suggested by Bevis et al. (1992). The mean tropospheric temperatures (Tm) with sampling rate of 1 minute were obtained from the temperature and pressure measured at the GPS antenna by applying the regional model suggested by Sapucci (2014), which is the most suitable for this region. The sampling rate of the GPS-PWV values was 1 minute. The GPS-PWV time series suffered some short failures due to

10 interruptions in data collection, problems with the incomplete satellite ephemerides file from JPL website and unavailable pressure measurements. These time series interruptions occurred in 3,183 epochs (3.1% of the total period), and the missing values were filled by a cubic spline interpolation method specific for application of wavelet analysis, which requires complete time-series without failures. These interpolation were concentrated in two specific days [Day of Year (hereafter called DoY) 331 and 348] and these periods were removed of the data analysis.

**15 2.3 Precipitation time series from disdrometer and XPol radar data**

[revised manuscript text omitted]

The high temporal resolution obtained with the GPS-PWV enables the evaluation of high frequency variations and their relationship with intense precipitation events. The GPS-PWV time series shows a well-defined sharp increase before the occurrence of precipitation, as reported by Kursinski et al. (2008) as a rapid rise in PWV preceding the rain events. Shi et al.

- 5 (2015) using GPS -PWV to monitoring the water vapor variation shown that ascending and descending patterns of GPS-PWV can be identified before and after each rainfall event. There are strong oscillations, generating a significant increase in the total water vapor content until a maximum is reached. Subsequently, a strong GPS-PWV reduction is observed, and after a short interval, the precipitation also reaches a maximum peak. Here, this sharp increase is called GPS-PWV jump. Fig. 3 shows a typical case exemplifying the PWV behavior before precipitation occurs on DoY 341; this was one of the strongest
- 10 events registered during the CHUVA Vale experiment. Before the severe precipitation begins, the GPS-PWV follows several pulses, increasing the value and forming the PWV jump, until it reaches a peak of maximum value. After the GPS-PWV crest, a decreasing period is observed some minutes before severe precipitation. Fig. 3 clearly shows this configuration of a crest in the GPS-PWV time series around precipitation (composed of several pulses) and its subsequent decrease immediately before the beginning of stronger precipitation.
- 15 This GPS-PWV behavior before precipitation occurs not only for more intense events but also for lower rainfall rates. Fig. 4 shows other GPS-PWV jumps observed before rain events with different intensity/extension occurred on: DoY 315 (41% of precipitation fraction above 35 mm h-1), DoY 332 (5%) and DoY 358 (only 1% of precipitation fraction above 35 mm h-1). This figure shows the intensity of these sharp increases in GPS-PWV are larger before more extensive precipitation. Table 2 presents all precipitation events in which the rainfall above 20 mm h-1 was observed by XPol radar in the area of 4.4 km per
- 20 4.4km around GPS antenna. This table presents the DoY and time of maximal radar precipitation of the each event and the respective fraction rain above 20 mm h-1, 35 mm h-1 and 50 mm h-1. In order to obtain an overview of the hardiness of the GPS-PWV jumps feature before precipitation, Fig. 5 show the composite mean of GPS-PWV time series from 60 minutes before to 60 minutes after maximum observed precipitation (fraction rain above 35 mm h-1) for 18 events listed by Table 2. The composite presented in Fig. 5 is normalized by maximum GPS-PWV values before precipitation. The composite mean
- 25 shows that the GPS-PWV jump is strongly remarkable before the maximum precipitation, which the maximum of the composite mean is observed in 30 minutes and lower dispersion in 25 minutes before the maximum precipitation. The time lag between the maximum GPS-PWV and the time of the maximum rainfall is presented in a sub plot in Fig.5. It is important highlight that the there are many precipitation events of lower intensity and extension in which the GPS-PWV jumps are not observe. In these cases, the maximum GPS-PWV is observed in the maximum precipitation, consequently the time lag is
- 30 zero (37% of the cases evaluated). In this histogram, it is clear there is a range of lag time between the GPS-PWV and precipitation maximums, the precipitation area and the temporal resolution of GPS-PWV here employed have an impact on this lag time. For instance, Adams et al. (2013) using rain gauge and PWV with sampling rate of 30 minutes over Amazonian region found out lag zero, however, in this study we shows that the GPS-PWV signal can represent a precipitation area and

**Comentário [s7]: Reviser 3E17 Comentário [s8]: Reviser3E26 Comentário [s9]: Reviser3E13**

**Comentário [s10]: Resiser3E14**

Comentário [s11]: Reviser3E7

**Comentário [s12]: Reviser3E7**

Comentário [s13]: Reviser3E8

|    | not only a punctual measurement over the GPS antenna. An analysis of the time lag correlation is necessary to define this      | Comentário [s14]: Reviser3E8  |
|----|--------------------------------------------------------------------------------------------------------------------------------|-------------------------------|
|    | interval between the maximal GPS-PWV and precipitation, which is carried out in the next section.                              |                               |
|    | The physical explanation for this behavior could be explained by different physical processes. First, the water vapor may      |                               |
|    | increase through low-level moisture convergence. The variation of the moisture convergence generates a sequence of pulses      |                               |
| 5  | of positive increases in the PWV value. The sequence of pulses of positive increases in the PWV could be a result of several   |                               |
|    | physical processes. Some of the physical process that can explain these pulses could be low-level water vapor convergence      |                               |
|    | forced by gravity wave (Raymond 1987) or simply unstable surface parcels accelerating upwards. After the increase of GPS-      | Comentário [s15]: Reviser3E12 |
|    | PWV at the crest of the jump, rainfall starts and PWV starts to decrease. The lag time between the crest and the maximum       |                               |
|    | precipitation can vary from one region to another, from one rainfall cell to another because it depends on the cloud           |                               |
| 10 | condensation nuclei and the precipitation efficiency, normally a function of the wind shear. In this study using precipitation |                               |
|    | measures based on area about the GPS antenna, we found a time lag between these maximum. Addams et al. (2013)                  |                               |
|    | considered that the conversion of water vapor to liquid water and precipitation are of second order during the process of      |                               |
|    | PWV increasing. It is probably true during the phase of cloud formation, however, when the precipitation starts, water vapor   |                               |
|    | decreases due to the formation of liquid water. The conversion of the water vapor to liquid water changes the dielectric       |                               |
| 15 | medium, where the refractivity is induced by the displacement of charge (Solheim et al. 1999). While the refractivity from     |                               |
|    | water vapor is due to the polar nature of the water molecule, the GPS phase delay induced by liquid water (hydrometeor) is     |                               |
|    | proportional to the electric permittivity of the formed dielectric medium and, consequently, much lower than the delay         |                               |
|    | generated by water vapor. Another important physical mechanism is the storm downdraft that is dryer and colder that the        |                               |
|    | ascending moisture air and this downdraft can also contributes to the decrease in PWV. In addition, PWV decreases after        |                               |
| 20 | precipitation starts can be simply associated to the final process of surface convergence as function of the rainfall and      |                               |
|    | downdrafts on the surface, or by the advection process forced by the shear and storm movement. Given the limitations of the    |                               |
|    | observations, our interpretation of the physical mechanisms responsible for the jumps remains speculative. To understand       |                               |
|    | what the physical mechanisms responsible for the pulses and GPS-PWV jumps, a specific field campaigns design are               |                               |
|    | needed.                                                                                                                        |                               |
|    |                                                                                                                                |                               |

**25 4 High temporal resolution GPS-PWV time series analysis**

The high temporal resolution GPS-PWV time series and precipitation in different intensity and extension are evaluated and the PWV-GPS jumps are characterized. The wavelet analysis are explored to evaluating which the timescale of the GPS PWV oscillations are associated with a more intense rainfall occurrence and time lag correlation analysis are used to determine the lag between the rainfall intensity and the GPS-PWV time series. Additionally, PWV-GPS derivative analysis

30 is explored and its potential for nowcasting application is discussed.

**4.1 Wavelet analysis**

Wavelet analysis was used to perform a detailed analysis of the GPS-PWV time series and to evaluate the variability within different time scales (denoted here as intra-relation), as well as to assess the relationship between the GPS-PWV time series and the precipitation time series (denoted here as interrelation wavelet analysis). This methodology enables simultaneous

- decomposition of the PWV time series as a function of time and frequency (Daubechies 1992). Consequently, accessing to the information regarding the signal amplitude/frequency and its variation as a function of time becomes possible.
   To perform the intra-relation analysis evaluation of how spectral characteristics change over scales (*s*) and time (*t*), but with highly redundant information, the continuous wavelet analysis (Torrence and Compo 1998) was used to estimate the wavelet power spectrum. Thus, a decomposition of the GPS-PWV data into time-variability space allows an evaluation of the main
- 10 frequencies composing the GPS-PWV time series during intense rainfall events. With continuous analysis, some hidden features of the time series can be identified, e.g., in which scale are the most representative behaviors of the time series. Continuous analysis is often easier to interpret because its redundancy tends to reinforce the traits and makes all information more visible. However, for some specific choices of values for time and frequency, it is possible to apply a discrete wavelet transform, which does not lose important information and has advantages of implementation and computational effort. This
- 15 is the case of the non-decimated discrete wavelet transform (NDWT), also called Maximal Overlap Discrete Wavelet Transform, which can be seen as a compromise between the discrete wavelet transform (DWT) and CWT because of its redundancy, but not as redundant as CWT. The NDWT can be computed similarly to the ordinary DWT but without subsampling (decimation), ensuring the translational invariance, which is ideal for analysing time series, especially interrelations between different time series. A time-variant transform disrupts the lag-resolution in a cross-correlation
- 20 analysis. Furthermore, estimators calculated using the NDWT are considered more preferable because they are asymptotically more efficient than the estimator based on the DWT (Percival and Walden 2000). As the bivariate relationship between two time series is essential for this research, a wavelet cross-correlation (WCC) constructed from NDWT (Whitcher et al. 2000) is ideal for analysing different scale structures and the interrelations of the dynamic behaviour of two time series, as well as the lead-lag relationships. Some lead-lag relations that could not be distinguished in the usual
- 25 cross-correlation can be investigated in the WCC, which decomposes the cross-correlation on a scale-by-scale basis. Thus, a specific scale may be associated with water vapor convergence before precipitation occurrence, also providing a lead time for nowcasting, a specific time scale for calculations and additional information to understand the physical mechanisms associated with intense rainfall events in this region.

Considering the large quantity of available discrete mother wavelets, some of the most used in the literature were evaluated:
Daubechies with 4, 6, 8, and 16 coefficients, denoted by D4, D6, D8, D16, and Daubechies Least Asymetric with 8, 16, and 20 coefficients, denoted by LA8, LA16, and LA20, respectively. To verify the statistically significance of the estimated wavelet correlations, the 95% confidence interval was estimated considering a Gaussian Distribution after applying the Fisher's Z Transformation (Whitcher et al. 2000).

Comentário [s16]: Reviser3E4

Because of implementation aspects considered in this study, the two time series were restricted to a power of two length, with 65,536 ( $2^{16}$ ) observations, excluding one day at the beginning and another at the end of the GPS-PWV series. The time span corresponds to 45 days, 12 hours and 16 minutes, beginning on DoY 314 (00 UTC 10 November) and finishing on DoY 359 (1216 UTC 27 December). The same time series length was used for both intra- and interrelation analyses. Because each wavelet scale *j* in the interrelation analysis corresponds to a frequency band from  $2^j$  to  $2^{j+1}$ , its inversion allows the interpretation in terms of a period of time also in a dyadic interval. Thus, WCC enables the identification of the most important scale of the PWV oscillation during precipitation events. Furthermore, the WCC of GPS-PWV and precipitation time series also allows evaluating the lead-lag correlation that may exist between these time series for the different time

10 periods.

5

**4.1.1 Wavelet power spectrum analysis**

For an intra-relation analysis, the wavelet power spectrum of GPS-PWV and 95th percentile of the precipitation intensity time series are presented in Fig. 6. To emphasize the highest-frequency oscillations, the power spectrum in Fig. 6 shows the scales that represent the period below 512 minutes (~8.5 h). The PWV diurnal cycle presents strong power along all time

15 series, and consequently, it was not taken into consideration in this analysis. The range of power spectrum associated with color scale used in Fig. 6 is for scales related to time periods larger than 16 minutes.

The PWV series with a one-minute temporal resolution presents oscillations of high frequency; however, the frequency of occurrence of rainfall events is very low. For this reason, it is necessary to take into consideration a long time period (with several precipitation events) and a short time step, e.g., the one-minute interval used in this study. Consequently, in a general

- 20 analysis of the PWV wavelet power spectrum, it is difficult to clearly discern which power is associated with each time step. However, a more specific analysis of the wavelet power spectrum during precipitation events indicates that there are, as expected in function of described GPS-PWV jump, strong changes in the power between different scales; cases with an increase in the power of the oscillation from low to high frequency are observed. This result indicates that PWV oscillations on scales related to time periods smaller than 128 minutes occur more frequently during precipitation events than in periods
- 25 without rain. Fig. 7 presents the same wavelet power spectrum presented in Fig. 6 but with an enlargement applied for precipitation events observed during the period from DoY 340 to DoY 343. The analysis of the power spectrum in these cases makes the result discussed for Fig. 6 more understandable. For clarity, a vertical line was drawn for each precipitation maximum. Fig. 7 shows that the power of the PWV oscillations between 128 and 32 minutes is stronger during more intense precipitation events (for example, the events that occurred on DoY 341 at 1836 UTC and DoY 342 at 1636 UTC) than
- 30 during light rainfall events (events observed during DoY 343) and periods without rain (DoY 340). It is interesting to observe that GPS-PWV presents a jump at the end of the DoY 357 with a strong signal in the wavelet power spectrum, but XPol radar did not detect any precipitation. The wavelet power spectrum (Fig. 6) also shows the impact of the GPS failures

Comentário [s17]: Reviser3E24

Comentário [s18]: Reviser3E16

that occurred during DoY 331 and 348, which are unfortunately very close to the other intense precipitation events that occurred on DoY 332 and 348. For this reason, these cases are not spotlighted in the wavelet analysis.

**4.1.2 Wavelet cross-correlation analysis**

Before presenting the inter-relation results, one might question about the robustness of the method regarding the choice of mother wavelets. We estimated the wavelet correlations using Daubechies and Least Asymetric wavelets with different filter lengths (coefficient number) as presented in Section 4.1. The results were quite similar independently of the mother wavelet, but the correlations were maximized when the mother wavelet has a larger filter or more coefficients (D16, LA16, and LA20). Thus, all estimated wavelet correlations were presented using the LA20 mother wavelet. Fig. 8 shows the wavelet correlation and its 95% confidence interval between the GPS-PWV and the 95th percentile of the precipitation intensity as a

- 10 function of the wavelet scale, represented by the respective time periods, considering the lag zero. In this analysis, the precipitation data from XPol radar, originally with sampling rate of 6 minutes, were linearly interpolated to one-minute rate. The results show that the wavelet correlation between the PWV and precipitation intensity is stronger for the scale related to the period between 32 and 64 minutes, indicating the scale on which the most important GPS-PWV oscillations associated with precipitation events occur. After this scale, the correlation decreases, followed by another increase due to the influence
- 15 of the diurnal cycle. Although the correlations are not very high, 95% confidence interval showed that these results are statistically significant.

To evaluate the results presented in Fig. 8 for different intensity and extension of precipitation events, Fig. 9 shows the wavelet correlation for lag zero between GPS-PWV and precipitation fractions as a function of the period bands for different rain fraction intensities (>20 mm h-1, Fig. 9a; >35 mm h-1, Fig. 9b; and >50 mm h-1, Fig. 9c). The 95% percentiles give an

- 20 information about the maximum rain rate (which can be only one point) on the area of 4.4 km per 4.4 km around the GPS antenna. The rain fraction gives an information about the fraction of these studied area covered by rain rate above of these thresholds. There are some events where the rain intensity can be high and the fraction small, for instance, the cases of isolate clouds. Therefore, the area fraction presents a more close representation of rainfall events related to low level convergence because it gives information about the amount of liquid water in the area. The peak of the wavelet correlation
- 25 observed in the time period from 32 to 64 minutes is significant for the three rain thresholds. The plots of Fig. 9 also show when only heavy to torrential precipitation events are taken into consideration (hereafter called intense rain events), stronger wavelet correlation is observed.

To perform the lead-lag analysis, the WCC is showed in Fig 10 between GPS-PWV and rain fraction for different intensities (>20 mm  $h^{-1}$ , Fig. 10a; >35 mm  $h^{-1}$ , Fig. 10b; and >50 mm  $h^{-1}$ , Fig. 10c). Although the precipitation and GPS-PWV time

30 series present very distinct behaviors, the WCC permitted the identification of the correlation in lead-lag of about 30 minutes mainly for scales related to time periods from 32 to 64 minutes, indicating which GPS-PWV oscillations are important for predicting precipitation events. This results show also that considering the rain fraction for stronger intensities (>50 mm h-1), some statistically significant correlations in lead-lag of about 30 minutes also appear stronger from period bands of 4- 16 Comentário [s19]: Reviser3E2

Comentário [s20]: Reviser3E25

Comentário [s22]: Reviser3E3

Comentário [s23]: Reviser3E1

and 16-32 minutes. This result indicates that the GPS-PWV carries some information, mainly, on the scale related to the period from 32 to 64 minutes. That signals the occurrence of precipitation events of large intensity and extension, which suggests that the GPS-PWV jumps are, for these storms evaluated, concentrated in this scale emphasizing its potential in a nowcasting application. It is also important to highlight that the correlation is larger on this scale than on adjacent scales.

5 Furthermore, these results indicate that, although GPS-PWV jump before precipitation occurs not only for more intense events, the intensity of these oscillations associated to jumps is greater before the most intense and extensive ones. However, this feature should be considered as specific to each region because the time scale between the humidity convergence and rainfall may depend on many physical processes and environmental conditions (e.g., gravity wave-induced convergence, wind shear or thermodynamic instability).

**10 4.2 Time lag correlation analysis**

The relationship between the rainfall intensity (or rain fraction) and the GPS-PWV is different for each event; however, the GPS-PWV peak is a well-delineated pattern. The time interval between the moment of the PWV crest and the maximal precipitation can vary among cases. The WCC shows on which scale the correlation between GPS-PWV and precipitation time series is higher, as well as the lead-lag interrelation between them. However, evaluating the lag correlation for positive

[revised manuscript text omitted]

Comentário [s27]: Reviser 3E27

Comentário [s26]: Reviser3E9

**Comentário [s28]: Reviser3E28**

Comentário [s29]: Reviser3E16

Comentário [s31]: Reviser3E29

Comentário [s30]: Reviser3E12

| and are significant before the most precipitation-extensive events. Fig. 13 shows the time series of GPS-PWV derivative           |                               |
|-----------------------------------------------------------------------------------------------------------------------------------|-------------------------------|
| before precipitation events of the upper tercile, and emphasizes the occurrence of this stronger derivative (above +9.5 mm h      | Comentário [s32]: Reviser3E10 |
| 1) observed at least three of the evaluated events. These stronger derivatives, associated to PWV pulses, occur with larger       |                               |
| frequency in the period from 55 to 25 minutes before the maximum precipitation, which is GPS-PWV jump is observed and             |                               |
| the pick of PWV maximum also occur.                                                                                               |                               |
| Although the GPS-PWV pattern before precipitation described in the previous section is well defined, its use as an index for      |                               |
| the occurrence of severe storms is not simple. Several studies have taken into account the intensity of rainfall events.          |                               |
| Therefore the use of only a maximum threshold from a GPS-PWV derivative, as suggested by Iwabuchi et al. (2006) and Shi           |                               |
| et al. (2015), is not sufficient to predict intense rainfall, several atmosphere processes are very complex and highly nonlinear. | Comentário [s33]: Reviser3E12 |
| The intensity of the precipitation is highly correlated with the intensity of the PWV value, which is formed by a succession      |                               |
| of pulses of positive increases in the PWV value. Therefore, the occurrence of more intense rain is signaled not only by the      | Comentário [s34]: reviser3E12 |
| maximum derivative but also by the increase in the frequency distribution of the positive and negative derivatives before the     |                               |
| occurrence of precipitation.                                                                                                      |                               |
| The result found by GPS-PWV derivative analysis suggests that an algorithm for intense precipitation forecasting using the        |                               |
| GPS-PWV should consider the following points: (a) increases in GPS-PWV positive variations compared with negative ones            |                               |
| in which the median values of the variation in the last 60 minutes reach positive values and (b) a simultaneous increase in the   |                               |
| population of the GPS-PWV derivatives above +9.5 mm h -1 . The value of the stochastic constrains applied on temporal  |                               |
| evolution of ZWD during the GPS data processing can make influence the PWV variability in high temporal resolution. The           |                               |
| value used in this study was the default, but some tests with different constraint can be carried out to identify the impact of   | Comentário [s35]: Reviser3E22 |
| this parameter in the derivative analysis before intense precipitation.                                                           |                               |
| The GPS-PWV values evaluated in this study are post-processed, and an additional study is required to determine whether           |                               |
| these estimates in real time are able to capture the jumps before the precipitation reported. True real-time processing for       |                               |
| PWV estimates in dense and regional GPS networks has been explored in other studies related to nowcasting applications            |                               |
| (Iwabuchi et al. 2006). De Haan and Holleman (2009) reported the construction and validation of a real-time PWV map from          |                               |
| a GPS network combined with data from weather radar, a lightning detection network, and surface wind observations. They           |                               |
| tested a nowcasting algorithm for three thunderstorm case studies and concluded that the GPS-PWV in real time can be              |                               |
| helpful for the nowcasting of severe thunderstorms. Shi et al. (2015) studied the PWV estimates in real time for rainfall         | Comentário [s36]: Reviser3E30 |
| monitoring and forecasting and shown that this estimate has quality comparable with post-processed product. A significantly       |                               |
| reduction in the latency was obtained with GPS data processing proposed by Shi et al. (2015), which demonstrated                  |                               |
|                                                                                                                                   |                               |

30 promising perspective of the PWV-GPS data for rainfall forecasting.

**5 Conclusions**

This work evaluates the correlation between large and rapid increases in the GPS-PWV and the occurrence of rainfall events observed by radar during the CHUVA Vale experiment in Brazil. A detailed analysis of the GPS-PWV time series was carried out, and strong and sudden sharp increase composed predominantly by positive derivatives, before the precipitation

5 events were identified and called as GPS-PWV jumps. In this process, a crest in the PWV series is remarkable before the precipitation events. Although this sharp increase can be observed for any precipitation event, it is preponderant before more intense and extensive precipitation events.

The wavelet analysis for the GPS-PWV time series was explored to characterize the strong changes in the power spectrum between different time scales during precipitation events generated by the occurrence of the GPS-PWV jumps. Additionally,

10 the application of wavelet cross-correlation between the PWV and precipitation showed that important oscillations exist between these variables on the scale related to a time period from 32 to 64 minutes, which is stronger for events of large intensity and extension. These results corroborates with those reported by Adams et al. (2013), who showed that the strongest water vapor convergence is typically ~1 hour before heavy precipitation.

A time lag-correlation histogram shows that in 85% of the studied events, a crest in the PWV time series occurs between 15

- 15 and 60 minutes before the maximum precipitation. The GPS-PWV derivative histogram shows the distribution change for different precipitation extension terciles. The average values of the GPS-PWV derivatives present an increase in positive values as a function of the increase in the rainfall extension terciles. The results suggest that the derivative average values in the interval of 60 minutes before precipitation changes to positive values, and an increase in the frequency of the derivative above +9.5 mm h-1 can indicate the occurrence of severe precipitation. Consequently, a methodology based on the
- 20 monitoring of the GPS-PWV derivative histogram presents potential for exploration in nowcasting applications, but additional studies will be necessary to define an appropriate algorithm and characterize the skill of this tool using the GPS-PWV estimates in real-time.

Competing interests: The authors declare that they have no conflict of interest.

25

30

Acknowledgments: The authors are grateful to David Adams and three AMT's reviewers for their many constructive comments, which helped us to improve the technical and scientific quality of this paper. Thanks are given to the CHUVA team who were involved directly or indirectly in the data collection by XPol radar, disdrometer and GPS receiver during the CHUVA-VALE experiment in São José dos Campos-S.P. Special thanks is given to Thiago Souza Biscaro. This study was supported by the Fundação de Amparo a Pesquisa do Estado de São Paulo (FAPESP), which directly supported this experiment [Grant Process 2009/15235-8 (CHUVA project)], and the Contractual Instrument of the Thematic Network of Geotectonic Studies CT-PETRO (PETROBRAS) and INPE (Grant: 600289299), which provided the GPS receiver used in this experiment. The raw data (Level 0) from the XPol radar, disdrometer and GPS receiver used in this study and the values

Comentário [s37]: Reviser3E13 Comentário [s38]: Reviser3E17 Comentário [s39]: Reviser3E13

Comentário [s40]: Reviser3E16

[revised manuscript text omitted]

Giangrande, S. E., T. Toto, A. Bansemer, M. R. Kumjian, S. Mishra, and A. V. Ryzhkov, 2016: Insights into riming and aggregation processes as revealed by aircraft, radar, and disdrometer observations for a 27 April 2011 widespread

precipitation event, J. Geophys. Res. Atmos., 121, 5846–5863, doi:10.1002/2015JD024537. Gematronik, 2007: Dual-polarization weather radar handbook. In: Bringi, V.N., Thurai, M., Hannesen, R. (Eds.), Selex-SI gematronik, 2nd edition (163 pp.).

Guerova G, J. Jones, J. Dousa, G. Dick, S. De Haan, E. Pottiaux, O. Bock, R. Pacione, G. Elgered, H. Vedel, M. Bender, 2016: Review of the state-of-the-art and future prospects of the ground-based GNSS meteorology in Europe. Atmos Meas Tech Discuss. doi:10.5194/amt-2016-125

de Haan, S. D., S. Barlag, H. K. Baltink, F. Debie, and H. V. Marel, 2004: Synergetic Use of GPS Water Vapor and

- 30 Meteosat Images for Synoptic Weather Forecasting. J. Appl. Meteoro., 43, 514-518, doi:http://dx.doi.org/10.1175/1520-0450(2004)043<0514: SUOGWV>2.0.CO;2.
  - de Haan, S. D., 2006: Measuring Atmospheric Stability with GPS. J. Appl. Meteoro. Climatol., 45, 467-475, doi: http://dx.doi.org/10.1175/JAM2338.1.

Gregorius, T., 1996: GIPSY-OASIS II How it works. Department of Geomatics, University of Newcastle upon Tyne, 167 pp. [Available online at: http://web.gps.caltech.edu/classes/ge167/file/gipsy-oasisIIHowItWorks.pdf.].

de Haan, S. D., and I. Holleman, 2009: Real-Time Water Vapor Maps from a GPS Surface Network: Construction, Validation, and Applications. J. Appl. Meteoro. Climatol., 48, 1302-1316, doi: http://dx.doi.org/10.1175/2008JAMC2024.1.

Holloway, C. and J. Neelin, 2010: Temporal Relations of Column Water Vapor and Tropical Precipitation. J. Atmos. Sci., 67, 1091–1105, doi: http://dx.doi.org/10.1175/2009JAS3284.1.

5

- Iwabuchi, T., C. Rocken, Z. Lukes, L. Mervat, J. Johnson, and M. Kanzaki, 2006: PPP and Network True Real-time 30 sec Estimation of ZTD in Dense and Giant Regional GPS Network and the Application of ZTD for Nowcasting of Heavy Rainfall. Proceedings of the 19th International Technical Meeting of the Satellite Division of The Institute of Navigation (ION GNS 2006), 1902-1909.
- 10 IERS Conventions, 2010: Gérard Petit and Brian Luzum (eds.). (IERS Technical Note ; 36) Frankfurt am Main: Verlag des Bundesamts für Kartographie und Geodäsie, 2010. 179 pp., ISBN 3-89888-989-6 https://www.iers.org/IERS/EN/Data Products/ Conventions/conventions.html.

Jerrett, D., and J. Nash, 2001: Potential Uses of Surface Based GPS Water Vapour Measurements for Meteorological Purposes. Phys. Chem. Earth (A), 26, 457-461, doi:10.1016/S1464-1895(01)00083-7.

15 Joss, J., and A. Waldvogel, 1967: Ein Spektrograph für Niederschlags-tropfen mit automatischer Auswertung (A spectrograph for rain drops with automatical analysis). Pure Appl. Geophys., 68, 240–246, http://dx.doi.org/10.1007/BF00874898.

Kinnell, P. I. A., 1976: Some Observations on the Joss-Waldvogel Rainfall Disdrometer, J. Appl. Meteor., 15, 499–502, doi:http://dx.doi.org/10.1175/1520-0450(1976)015<0499:SOOT JW>2.0.CO;2.

20 Kursinski, E. R., R. A. Bennett, D. Gochis, S. I. Gutman, K. L. Holub, R. Mastaler, C. Minjarez Sosa, I. Minjarez Sosa, and T. van Hove (2008), Water vapor and surface observations in northwestern Mexico during the 2004 NAME Enhanced Observing Period, Geophys. Res. Lett., 35, L03815, doi:10.1029/2007GL031404.

Lyard, F., F. Lefèvre, T. Letellier, and O. Francis, 2006: Modelling the global ocean tides: a modern insight from FES2004, Ocean Dynamics, 56, 394-415, doi:http://dx.doi.org/10.1007/s10236-006-0086-x.

- Machado, L. A. T., M. A. F. Silva Dias, C. Morales, G. Fisch, D. Vila, R. Albrecht, S. J. Goodman, A. J. P. Calheiros, T. Biscaro, C. Kummerow, J. Cohen, D. Fitzjarrald, E. L. Nascimento, M. S. Sakamoto, C. Cunningham, J. P. Chaboureau, W. A. Petersen, D. K. Adams, L. Baldini, C. F. Angelis, L. F. Sapucci, P. Salio, H. M. J. Barbosa, E. Landulfo, R. A. F. Souza, R. J. Blakeslee, J. Bailey, S. Freitas, W. F. A. Lima, and A. Tokay, 2014: The Chuva Project: How Does Convection Vary across Brazil? Bull. Amer. Meteor. Soc., 95, 1365–1380, doi:http://dx.doi.org/10.1175/BAMS-D-13-00084.1.
  - Madhulatha, A., M. Rajeevan, M. Venkat Ratnam, J. Bhate, and C. V. Naidu, 2013: Nowcasting severe convective activity over southeast India using ground-based microwave radiometer observations. J. Geophys. Res., 118, doi:10.1029/2012JD018174.

Mazany, R. A., S. Businger, S. I. Gutman, and W. Roeder, 2002: A lightning prediction index that utilizes GPS integrated precipitable water vapor. Wea. Forecasting, 17, 1034–1047, doi:10.1175/1520-0434(2002)017<1034:ALPITU>2.0.CO;2.
Montenbruck, O., R. Schmid, F. Mercier, P. Steigenberger, C. Noll, R. Fatkulin, S. Kogure, A.S. Ganeshan, 2015: GNSS satellite geometry and attitude models, Advances in Space Research, 56, 6, 1015-1029. Doi:http://dx.doi.org/10.1016/ i.asr.2015.06.019.

 Moore, A., I. Small, S. Gutman, Y. Bock, J. Dumas, P. Fang, J. Haase, M. Jackson and J. Laber, 2015: National Weather Service Forecasters Use GPS Precipitable Water Vapor for Enhanced Situational Awareness during the Southern California Summer Monsoon. Bull. Amer. Meteor. Soc., 96, 1867–1877, doi: 10.1175/BAMS-D-14-00095.1, Muller, C. J., L. E. Back, P. A. O'Gorman, and K. A. Emanuel, 2009: A model for the relationship between tropical precipitation and

5

20

25

- column water vapor. Geophys. Res. Lett., 36, L16804, doi:10.1029/2009GL039667.
   Percival, D. B., and A. T. Walden, 2000: Wavelet Methods for Time Series Analysis. Cambridge University, 594 pp.
   Raymond, D. J. 1987: A Forced Gravity Wave Model of Serf-Organizing Convection. J. Atmo. Sci., 44, 23, 3528-3543.
   Rocken, C., T. VanHove, M. Rothacher, F. Solheim, R. Ware, M. Bevis, S. Businger, and R. Chadwick, 1994: Towards near-real-time estimation of atmospheric water vapor with GPS. Eos Trans. AGU, 7544, Fall Meet. Suppl., 173.
- 15 Sapucci, L. F., 2014: Evaluation of Modeling Water-Vapor-Weighted Mean Tropospheric Temperature for GNSS-Integrated Water Vapor Estimates in Brazil. J. Appl. Meteor. Climatol., 53, 715–730, doi:http://dx.doi.org/10.1175/JAMC-D-13-048.1.

Sapucci, L. F., L. A. T. Machado, J. F. G. Monico, and A. Plana-Fattori, 2007: Intercomparison of Integrated Water Vapor Estimative from multi-sensor in Amazonian Regions. J. Atmos. Oceanic Technol., 24, 1880–1894, doi:http://dx.doi.org/10.1175/JTECH2090.1.

- Serra, Yolande L., David K. Adams, Carlos Minjarez-Sosa, James M. Moker, Jr., Avelino Arellano, Christopher Castro, Arturo Quintanar, Luis Carlos Alatorre, Alfredo Granados-Olivas, Enrique Vazquez, Kirk L. Holub, Charles DeMets, 2016: The North American Monsoon GPS Transect Experiment 2013. *Bull. Amer. Meteor. Soc.*, 97, 2103–2115, doi: 10.1175/BAMS-D-14-00250.1. Sato, T and F. Kimura, 2005: Diurnal Cycle of Convective Instability around the Central
- Schmid, R., P. Steigenberger, G. Gendt, M. Ge, M. Rothacher, 2007: Generation of a consistent absolute phase-center correction model for GPS receiver and satellite antennas. Journal of Geodesy, 81: 781. 81, 12, 781–798, doi:10.1007/s00190-007-0148-y

Mountains in Japan during the Warm Season. J. Atmo. Sci., 62, 1626-1636.

Shangguan, M., S. Heise, M. Bender, G. Dick, M. Ramatschi, and J. Wickert, 2015: Validation of GPS atmospheric water vapor with WVR data in satellite tracking mode, Ann. Geophys., 33, 55-61, doi:10.5194/angeo-33-55-2015.

Shi, J., X. Chaoqian, G. Jiming and G. Yang, 2015: Real-Time GPS precise point positioning-based precipitable water vapor estimation for rainfall monitoring and forecasting, IEEE Trans. Geosci. Remote Sens, vol.53, pp. 3452-3459, doi: 10.1109/TGRS.2014.2377041.

- Solheim, F. S., J. Vivekanandan, R. H. Ware, and C. Rocken, 1999: Propagation delays induced in GPS signals by dry air, water vapor, hydrometeors, and other particulates. J. Geophys. Res., 104, 9663-9670, doi:10.1029/1999JD900095. Testud, J., E. L. Bouar, E. Obligis, and M. Ali-Mehenni, 2000: The Rain Profiling Algorithm Applied to Polarimetric Weather Radar. J. Atmos. Oceanic Technol., 17, 332-356, doi:http://dx.doi.org/10.1175/1520-5 0426(2000)017<0332:TRPAAT>2.0.CO;2. Torrence, C., and G. P. Compo, 1998: A Practical Guide to Wavelet Analysis. Bull. Amer. Meteor. Soc., 79, 61-78. Whitcher, P., D. B. Guttorp, and B. Percival, 2000: Wavelet analysis of covariance with application to atmospheric time series. J. Geophys. Res., 105, 14941-14962, doi::10.1029/2000JD900110. Wolfe, D. E., and S. I. Gutman, 2000: Developing an operational, surface-based, GPS, water vapor observing system for
- 10 NOAA: Network Design and Results. J. Atmos. Ocean. Technol., 17, 426-440. Zumberge, J. F., M. B. Heflin, D. C. Jefferson, M. M. Watkins, and F. H. Webb (1997), Precise point positioning for the efficient and robust analysis of GPS data from large networks, J. Geophys. Res., 102(B3), 5005–5017, doi:10.1029/96JB03860.

15

**Tables**

Table 1: Configuration items associated with ZTD estimates and respective values used GPS data processing using GOA-II software.

| Configuration item                       | Configuration item Parameter used                                                                                                         |                                 |                               |
|------------------------------------------|-------------------------------------------------------------------------------------------------------------------------------------------|---------------------------------|-------------------------------|
| GPS data file for GOA-II                 | Rinex format in sampling rate of 1 second                                                                                                 | Data collection from CHUVA      |                               |
| Mapping function                         | Global Mapping Function (Boehm et al. 2006)                                                                                               | Selected in the data processing |                               |
| Cut-off elevation angle:                 | $10^{\circ}$                                                                                                                              | Selected in the data processing |                               |
| Elevation weight in the Mapping function | Elevation dependent and apply 3 coefficients
depend on the latitude and height above sea
level of the observing site and on the DoY | (Boehm et al. 2006)             |                               |
| ZHD a priori                             | Constant using a Tropospheric model in function of high of the GPS station (2.126 m)                                                      | Default values                  |                               |
| ZWD a priori                             | Constant (0.10 m)                                                                                                                         | Default values                  |                               |
| Process noise for tropospheric delay     | Random walk                                                                                                                               | Default values                  |                               |
| Maximum drift for tropospheric delay     | 8.333E-8 (km per square-root second);                                                                                                     | Default values                  | Comentário [s41]: Reviser3E5  |
| Tropospheric Gradients
estimates      | yes                                                                                                                                       | Default values                  | Comentário [s42]: Reviser3E23 |
| Maximum drift for Gradients
estimates | 8.333E-9 (km per square-root second)                                                                                                      | Default values                  |                               |
| Sampling rate of the ZTD estimates       | 60 seconds                                                                                                                                | Selected in the data processing |                               |

|       | DoY | Maximal radar
precipitation
UTC Time (hh:mm) | Precipitation fraction
observed by XPol radar (%) |                       |                        | Terciles |
|-------|-----|----------------------------------------------------|------------------------------------------------------|-----------------------|------------------------|----------|
| Event |     |                                                    |                                                      |                       |                        |          |
|       |     |                                                    | Above                                                | Above                 | Above                  |          |
|       |     |                                                    | $50 \text{ mm h}^{-1}$                               | 35 mm h -1 | $20 \text{ mm h}^{-1}$ |          |
| 1     | 348 | 02:42                                              | 73                                                   | 85                    | 95                     |          |
| 2     | 354 | 21:12                                              | 28                                                   | 45                    | 63                     |          |
| 3     | 341 | 18:36                                              | 36                                                   | 41                    | 45                     | Upper    |
| 4     | 315 | 17:12                                              | 26                                                   | 41                    | 49                     | tercile  |
| 5     | 335 | 19:24                                              | 30                                                   | 38                    | 42                     |          |
| 6     | 352 | 20:00                                              | 2                                                    | 33                    | 84                     |          |
| 7     | 342 | 16:36                                              | 24                                                   | 27                    | 28                     |          |
| 8     | 326 | 21:18                                              | 8                                                    | 26                    | 45                     |          |
| 9     | 343 | 01:06                                              | 4                                                    | 9                     | 15                     | Middle   |
| 10    | 338 | 20:18                                              | 0                                                    | 8                     | 19                     | tercile  |
| 11    | 332 | 19:18                                              | 5                                                    | 5                     | 5                      |          |
| 12    | 333 | 19:42                                              | 0                                                    | 2                     | 8                      |          |
| 13    | 314 | 21:06                                              | 2                                                    | 2                     | 7                      |          |
| 14    | 327 | 00:36                                              | 0                                                    | 1                     | 25                     |          |
| 15    | 358 | 23:00                                              | 0                                                    | 1                     | 10                     | Lower    |
| 16    | 317 | 21:48                                              | 0                                                    | 1                     | 2                      | tercile  |
| 17    | 318 | 08:48                                              | 0                                                    | 0                     | 19                     |          |
| 18    | 331 | 17:12                                              | 0                                                    | 0                     | 2                      |          |

Table 2: Precipitation events observed by radar during CHUVA Vale experiment in different extension terciles as a function of the precipitation fractions above 35 mm  $h^{-1}$ .

\_\_\_\_

\_

| Statistical Measurements                 | Other cases | Terciles |        |        |  |
|------------------------------------------|-------------|----------|--------|--------|--|
|                                          |             | Lower    | Middle | Upper  |  |
| Average value (mm h -1 )      | +0.04       | -0.18    | -0.38  | 0.13   |  |
| Standard deviation (mm h -1 ) | ±2.52       | ±3.18    | ±4.76  | ±5.57  |  |
| Median (mm h -1 )             | 0.00        | 0.00     | +0.29  | +0.65  |  |
| Mode (mm $h^{-1}$ )                      | 0.00        | 0.00     | +1.00  | +1.00  |  |
| Maximal value (mm h -1 )      | +21.13      | +8.42    | +11.00 | +13.25 |  |
| Minimal value (mm h -1 )      | -19.07      | -6.99    | -17.30 | -14.15 |  |
| $\% > +9.5 \text{ mm h}^{-1}$            | 0.21%       | 0.00%    | 0.78%  | 7.81%  |  |
| $\%

---

## Author Comment (AC3) · 26 Mar 2017

* * *
**Response latter for Referee 2**
* * *
**General comments from Referee#2**

The manuscript presents the behavior of GPS-PWV time series during severe precipitation events recorded by an X-band Radar during the CHUVA Vale measurement campaign in 2011. GPS-PWV jumps have been detected between 32 and 64 minutes before the more intense rainfall events. The statistical characterization of this phenomenon has potential for nowcasting. The reasoning of the paper is based on a

wavelet analysis of the GPS-PWV times series and on GPS-PWV derivative analysis and distinguishes meteorological events into 3 classes of precipitation. The main value of the article is to focus on the behavior of PWV during intense weather events, expecting to improve their forecasting. However, the article contains a number of imprecision that are important to dissipate : too many repetitions, a presentation that deserves to be more rigorous, more structured, more synthetic on the basic methodology and more explicit on the work done:

- 1. The presentation of GPS data processing is only too partial and often confusing.

- 2. The use of GPS-PPP times series sampled at 1-min intervals is interesting but the manuscript did not present well which specific PPP products were used to get it.

- 3. The use of wavelet analysis is interesting only for part "3.2 Wavelet cross-correlation analysis" even if shown correlations seem to be weak (figures 5 and 6). Part 3.1 "Wavelet power spectrum analysis" is the effect of the GPS-PWV jumps, a well know wavelet power spectrum of a Dirac function. Part 3 and 4 should be merged : part 3.1 for presenting the GPS-PWV jumps and part 3.2 for presenting time lag correlation.

- 4. Part 4 is really interesting and could be more developed. However, the criterion on GPS-PWV derivative > +9.5 mm.hËẼ(-1) and < -9.5 mm.hËẼ(-1) to characterize extreme weather events is not enough analyzed.

About GPS data processing Part "2.2 High temporal resolution GPS-PWV time series" is too confusing and must be structured and clarified: - Orbit and clock data products : what kind of products did you use ? What is the sampling of these products ? Orbit and clock data products from JPL for PPP applications are sampled at 5 minutes

(https://gipsy-oasis.jpl.nasa.gov/index.php?page=data). This point determines the rest : I would like to be sure that the final sampling rate of the GPS-PWV values is really 1 minute, as usual. If you made a specific processing, you have to present it and made a comparison with a standard GPS data processing. - Elevation-Dependent Weighting used for GPS observations (constant ?, elevation dependent (aËȩ2/sin(elev)ËȨ2) ?) - Tropospheric models used : ZHD a priori (Is it GPT ?), ZWD time evolution constraint as a random walk ? Were tropospheric gradients estimated ?

**Response:** The presentation of the methodology and of the results analysis in the previous version of the work was significant improved taken into consideration the comments, suggestions and corrections pointed by reviser. Several imperfections and mistakes was corrected, particularly in the GPS data processing section, which was rewritten with more rigorous and structured way to present the work done. The main aspects treated in the manuscript in function of reviser's comment are:

- I. The bibliographic revision of previous work about the subject of this manuscript (in the introduction section and other parts of text) was improved and many phrase were rewritten. A total of 11 new papers was included, as suggested by reviser, and some important information from these mentioned references were included in the text.

- II. The manuscript was restructured and several section was rewritten. The GPS-PWV jump was moved to before wavelet analysis and parts of the section 3 and section 4 were merged, as well suggested. This restructure of the manuscript required very hard work and all sections had to be revised.

- III. The methodology used in the GPS Data processing to generate the PWV values in high temporal resolution was better described in the 2.2 section of the new version of manuscript, This section was totally rewritten and more details

about the GPS data processing was included. All items pointed by reviser were taken into consideration: a new table was included to organize these information.

- IV. GOA-II software was extensively explored to obtain the ZTD estimates in high temporal resolution and the configuration used here is the same suggested by JPL (default). The GOA-II, acronyms of GPS-Inferred Positioning SYstem (GIPSY) and Orbit Analysis Simulation Software (OASIS) software re-sampling the GPS satellite ephemeris using OASIS package which starts from specified initial orbits from JPL and uses several force models to integrate the dynamic equation to produce the orbit of the each GPS satellite.

- V. Incomplete information about the GPS antenna calibration was corrected following the observations and suggestions done by reviser. We used the phase center variation model for satellite and receivers antenna and the references were updated. Information about the kind of GPS antenna used in the CHUVA-VALE was included in the section 2.1.

- VI. We agree that the description of the methodology applied in the wavelet analysis was very poor in details. It was rewritten in new section 4 and information about the discrete and continuous wavelet are included.

- VII. The analysis of GPS-PWV derivative results were improved with the inclusion of a new figure and the derivative above +9.5 mm.hËE̩(-1) were better characterize during extreme weather events (upper tercile).

- VIII. Information about the stochastic model used in the data processing were presented, as well as information about azimuthal gradient estimation, and the constrains of the temporal evolution of these parameters. The used values were the default values suggested by JPL, which can impact the variability of the PWV estimates in high frequency and other values should be tested ahead in this research.

Other minor corrections were done and imperfection removed, based in this rigorous and detailed revision presented by referee 2, which was very important to better organizing the methodology employed in the data processing and emphasize the results obtained in this research. Specific comments point-by-point are presented below. The numbers (in the new version of the manuscript) of the changed lines and respective page in each point are listed and highlighted in green.The numbers of figure in the response refer to 13 figures of the new version of the manuscript.

**Specific comments**

**Page 1:**

1) **Referee#2:** line 23: "an unconventional solution" This is a usual method to get PWV and it has been validated for some time now.

**P1L24 Response:** Evaluating of the reviser's point of view, we are agree that the term "unconventional" is not appropriated and it was changed by "indirect", which express the organized idea in the next phrases in the manuscript.

2) **Referee#2:** line 24: "delay is associated with the atmospheric density (i.e., temperature, pressure and water vapor)" Explanation too confused: the magnitude of this delay is related to the integral of the refractivity index of the air as a function of temperature, pressure and water vapor (Bevis et al. 1992) on the optical path followed by the GNSS signal.

**P1L25-L26 Response:** The phrase suggested by reviser was included in the manuscript replacing the mentioned phrase.

3) **Referee#2:** Explanation line 25-27 is too expeditious. - lines 25-26: "with an error of approximately 5% under all weather conditions (Wolfe and Gutman 2000)" The given reference is outdated using old version of GPS data processing software, relative calibrations of antennas etc. The evaluation of the accuracy of GPS-PWV at around 5% should also be given in millimeter. The accuracy of GPS-PWV estimates remains a active topic of research, especially during severe weather conditions when all other meteorological instruments are down. - lines 26-27 "and in near real time (Rocken et al. 1994)." Outdated reference that could be used to put into perspective the improvements made since. It could be very interesting to emphasize on the methodological improvements made from 2000 and the first utilization of the GPS-PWV estimate until now. (Guerova et al. , 2016) ! Guerova G, Jones J, Dousa J, Dick G, De Haan S, Pottiaux E, Bock O, Pacione R, Elgered G, Vedel H, Bender M (2016) Review of the state-of-the-art and future prospects of the ground-based GNSS meteorology in Europe. Atmos Meas Tech Discuss. doi:10.5194/amt-2016-125

**P1L29-L30 P2L13-L15 Response:** The referee is right, the retrospective about the evolution of the GPS-PWV estimates was very simplified and many important work are not mentioned. We included some information about the evolution of the GPS data processing, which has been minimized the uncertainty and improvement in the accuracy of this estimates has been obtained. The more recent works from Moore et al. (2015) and Shangguan et al. (2015) were included. We agree that is interesting to emphasize the improvements of this technique and its potential for nowcasting applications and to do this we used the results presented by Guerova et al. (2016), as suggested.

4) **Referee#2:** line 31: "high temporal resolution (minutes)." I know it is not easy to present it but this formulation hides many methodological points about the methodology of GPS data processing: zero, single, double difference analysis or PPP, the different ways to model ZWD in analysis: : : The easiest way to get high temporal resolution (minutes) on GPS-PWV estimates is the PPP strategy (Zumberge et al, 1997) ! Zumberge, J. F., M. B. Heflin, D. C. Jefferson, M. M. Watkins, and F. H. Webb (1997), Precise point positioning for the efficient and robust analysis of GPS data from large networks, J. Geophys. Res., 102(B3), 5005–5017, doi:10.1029/96JB03860.

**P2L1 Response:** This reference was included in the manuscript after the phrase "high temporal resolution (minutes)" as well suggested by reviser. This reference is very appropriated to simplify the methodology involved GPS data processing in solution with high temporal resolution. The methodology used in the GPS data processing in this study is based on Precise point positioning (PPP) using the GOA-II software. More details about this process were included in the section 2, following the recommendation from reviser.

5) **Referee#2:** Page 1 Line 31 ! page 2 line 1: "Other promising applications become viable in dense networks and transects": using dense networks to do tomography is not a "promising application" even if it remains methodological issues (e. g. Champollion et al. (2005); Bastin et al. (2005); Brenot et al. (2014) ) ! Bastin, S., C. Champollion, O. Bock, P. Drobinski, and F. Masson (2005), On the use of GPS tomography to investigate water vapor variability during a Mistral/sea breeze event in southeastern France, Geophys. Res. Lett., 32, L05808, doi:10.1029/2004GL021907. ! Brenot, H., Walpersdorf, A., Reverdy, M., van Baelen, J., Ducrocq, V., Champollion, C., Masson, F., Doerflinger, E., Collard, P., and Giroux, P.: A GPS network for tropospheric tomography in the framework of the Mediterranean hydrometeorological observatory Cévennes-Vivarais (southeastern France), Atmos. Meas. Tech., 7, 553-578, doi:10.5194/amt-7-553-2014, 2014. ! C. Champollion, F. Masson, M.-N. Bouin, A. Walpersdorf, E. Doerflinger, O. Bock, J. Van Baelen, GPS water vapour tomography: preliminary results from the ESCOMPTE field experiment, Atmospheric

Research, Volume 74, Issues 1–4, March 2005, Pages 253-274, ISSN 0169-8095, http://dx.doi.org/10.1016/j.atmosres.2004.04.003.

**P2L5-L10 Response:** The term "promising" was removed of this phrase, and works about the tomography were included in the new version of the manuscript, mentioning the application and localization of the experiments, as well suggested.

**Page 2:**

6) **Referee#2:** line 3: "the diurnal cycle": more discussion about meteorological processes who have been detected according to areas? (For West African Monsoon, Bock et al. (2007)) ! Bock, O., F. Guichard, S. Janicot, J. P. Lafore, M.-N. Bouin, and B. Sultan (2007), Multiscale analysis of precipitable water vapor over Africa from GPS data and ECMWF analyses, Geophys. Res. Lett., 34, L09705, doi:10.1029/2006GL028039.
**P2L4-L13 Response:** This phrase was rewritten and all references were chronologically organized and associated with the meteorological process evaluated and the area of study were mentioned. Bock et al. (2007) was included because this paper presents an important study in a region with poor humidity data base.

7) **Referee#2:** About §2 (lines 5-12) and §3 (lines 13-22): Is it possible to merge §2 and §3 to emphasize on links between PWV, deep convective activity and the occurrence of intense rainfall?
**P2L17-L18 Response:** These paragraphs were merged to emphasize on links between PWV, deep convective activity and the occurrence of intense rainfall. The reference Muller et al. (2009) was explored, which presented a model for this relationship.

[Figure]

8) **Referee#2:** line 7: "PWV data from a microwave radiometer (MWR) with high temporal resolution" Provide an order of magnitude.
**P2L19 Response:** This phrase was rewritten following the recommendation pointed by reviser in the next specific comments. The resolution was included. Really Miller et al. 2009 did not use MWR data.

9) **Referee#2:** Line 8: Wrong reference with Muller et al. (2009) who do not used MWR-PWV data. However, it remains interesting to provide a reference in the article about relationship between the PWV and tropical precipitation.
**P2L17-L18 Response:** This phrase was rewritten, the mistake about the MRW-PWV data was removed, and the reference was maintained to provide a reference about relationship between the PWV and tropical precipitation, as suggested.

10) **Referee#2:** Line 8: Using "Chan 2009" can be useful to discuss about differences between MWR-PWV and GPS-PWV during severe weather events (see in particular the interesting §3.3 Comparison with GPS receivers of the article).
**P2L22-L23 Response:** We agree that is important shows a discuss about differences between MWR-PWV and GPS-PWV during severe weather events. A comment about the results reported by Chan (2009) was included.

11) **Referee#2:** Line 9: "useful indications of the accumulation of water vapor" : be more specific, in which these indications are useful ?
**P2L22 Response:** This phrase "useful indications of the accumulation of water vapor" was deleted because are not important in the context.

12) **Referee#2:** Line 14: "Mazany et al. (2002) developed a lightning prediction index for Florida based on the GPS-PWV magnitude and its temporal evolution": if this index is interesting, can you write more about it?
**P2L28-L30 Response:** More information about this GPS application was included in the manuscript.

13) **Referee#2:** Line 19: "This study showed that prior to deep convective events in the central Amazon, a 4-hour "ramp up" in the time derivative of GPS-PWV is observed, reaching a maximum approximately one hour before heavy precipitation. This sentence should be in the next paragraph to better distinguish results from precedent studies and results of the article. It is a repetition of sentence line 26-27 "The sharp increase in the GPSPWV values approximately one hour before the occurrence of more intense rainfall events, as found in this study and that of Adams et al. (2013)" and should be merged with it.
**P3L3 Response:** The phrase was moved for the next paragraph and adapted as suggested. The second mentioned phrase was rewritten to avoid inconsistency.

**Page 3**

14) **Referee#2:** line 8: "The CHUVA Vale campaign was carried out in São José dos Campos in São Paulo State in an elevated valley between the Mantiqueira and Serra do Mar mountain ranges." Add the reference of fig.1 given line 23: "Fig. 1 shows the geographic location of the CHUVA Vale campaign, emphasizing the sites at which the instruments were placed."
**P3L28-L29 Response:** A reference to Fig.1 was included in this sentence, as suggested. The geographic coordinates were added following the recommendation from reviser 1 (see specific comments 4-Referee1).

15) **Referee#2:** line 11: "During the CHUVA campaigns," Too many repetitions.
**P3L30 Response:** This phrase was completely removed following recommendation from reviser 1 to avoid this repetition (see specific comment 4-Referee1).

16) **Referee#2:** Lines 11-14: "GPS meteorology was used to monitor the horizontal and temporal variations in the PWV associated with the wide variety of convection-producing mechanisms for the 6 geographic regions. For example, Adams et al. (2015) described the temporal and spatial evolution of tropical sea-breeze convection with GPS meteorological transects during the CHUVA-Belem field campaign." Of topics in this part : be more concise. You can eventually add this point in your introduction part. You speak about "6 geographic regions" without using it after.
**P3L30 Response:** This first phrase about the different regions observed by CHUVA campaign was excluded of the manuscript, and the second was moved for introduction section, as suggested.

17) **Referee#2:** line 23: "Fig. 1 shows the geographic location of the CHUVA Vale campaign, emphasizing the sites at which the instruments were placed." See comments about line 11.
**P4L7-L8 Response:** This phrase was rewritten following the recommendation in this item and your comments about the line 11 in the item 14, excluding the repetitions.

**Page 4**

18) **Referee#2:** subsection "2.2 High temporal resolution GPS-PWV time series" is too confusing and must be structured and clarified.:

- GPS data of receiver sampled at one second frequencies ! under-sampling ?

- Orbit and clock data products : what kind of products did you use ? What is the sampling of these products ? Orbit and clock data products from JPL for PPP applications are sampled at 5 minutes (https://gipsyoasis. jpl.nasa.gov/index.php?page=data). This point determines the rest : I would like to be sure that the final sampling rate of the GPS-PWV values is really 1 minute and not 5 minutes, as usual. If you made a specific processing, you have to present it. In addition, you have to ensure that this kind of estimation is appropriated for your purposes and your estimation doesn't suffer the effect of an artifact.

- Elevation weight function used for GPS observations (constant ?, elevation dependent (a/sin(elev)ËĘ2) ?) - Cut-off (OK line 5)

- Tropospheric models used : mapping function (OK line 5), ZHD a priori (Is it GPT ?), ZWD time evolution constraint as a random walk ? Tropospheric gradients have been estimated ?

**P4L25-27 P4L30-LP5L3 Table 1 Response:** In this study the GOA-II software was extensively explored to obtain the ZTD estimates. The configuration used here are the same suggested by JPL (default in the software distribution) and some particular aspects used to generating the high temporal resolution are better described in the 2.2 section of the new version of manuscript. This section was totally rewritten and more details about the GPS data processing were included. All items pointed by reviser were taken into consideration:

- GPS datain for GOA-II: rinex format in sampling rate of 1 second;

- Origin of the Orbit and clock data products: precise from JPL website;

- Sampling rate of satellites ephemeris: 15 minutes for orbits and 5 minutes for satellite crock;

- Elevation weight function: elevation dependent and apply 3 coefficients depend on the latitude and height above sea level of the observing site and on the day of the year (Boehm et al. 2006);

- ZHD a priori: Constant using the a Tropospheric model in function of high of the GPS station: DZH=1.013*2.27*exp(-$h_0 * 0,116 * 10^{-3}) = 2.1260m$;

- ZWD a priori: constant 0.10 m;

- Data processing method: PPP (Precise Point Positioning)

- Process noise for tropospheric delay: Random walk;

- Maximum drift for tropospheric delay (Random walk parameter): 8.333E-8 (km per square-root second);

- Tropospheric Gradients estimates: yes;

- Sampling rate of the ZTD estimates: 60 seconds;

These information were organized in the new Table 1.

19) **Referee#2:** Line 6: "To ensure the quality of the PWV time series with high temporal resolution required in this study, a rigorous data-processing strategy was adopted with possible noise sources taken into consideration." What does it mean? Give a reference if you used a validated data-processing strategy else explicit it please.

**P4L23-L25 Response:** We would like to highlight in this phase that in the GPS data processing carried out to generate PWV time series with high temporal resolution is necessary to take into consideration the modeling of the known uncertainty sources. This phrase was rewritten in new structure of this section, which this rigorous data-processing was demonstrated with details through the items listed and suggested by reviser in his specific comment (Number 18)

[Figure]

20) **Referee#2:** Line 8: "recommended by the International Global Naviga-
tion Satellite System Service" : use IERS conventions that are authorita-
tive in the field. IERS Conventions (2010). Gérard Petit and Brian Luzum
(eds.). (IERS Technical Note ; 36) Frankfurt am Main: Verlag des Bunde-
samts für Kartographie und Geodäsie, 2010. 179 pp., ISBN 3-89888-989-6
https://www.iers.org/IERS/EN/DataProducts/Conventions/conventions.html.
**P4L28 Response:** This reference IERS Conventions (2010) was added as suggested
replacing the "International Global Navigation Satellite System Service".

21) **Referee#2:** lines 9-10: "absolute calibration was performed to ensure the correct
phase center variation, as reported by Görres (2006)" : It would be clearer to distin-
guish in absolute calibration (Schmidt et al., 2009) a and the specific absolute calibra-
tion of your antenna (Gorres, 2006) if you have done it. ! Schmid, R., P. Steigenberger,
G. Gendt, M. Ge and M. Rothacher (2007), Generation of a consistent absolute phase-
center correction model for GPS receiver and satellite antennas, Journal of Geodesy,
Volume 81, Number 12, 781-798, doi :10.1007/s00190-007-0148-y.
**P4L28-L30 Response:** The reviser is right. This phrase was incomplete and was
rewritten and the references were updated. We did not do specific calibration for GPS
antenna used in the study. We used the phase center variation model for satellite
and receivers antennas (the reference suggested was used: Schmid et al. 2007),
which is available in GOA-II with parameters provided by IGS web site and the refer-
ence cited is changed for Montenbruck et al. (2015). The current file is available in
https://igscb.jpl.nasa.gov/igscb/station/general/igs14_1935.atx. Information about the
kind of GPS antenna used in the CHUVA-VALE was included in the section 2.1.

22) **Referee#2:** Line 10: New paragraph to explain the conversion ZTD ! PWV? Line
13-14: Which TM and Pressure data have been used, and with which time resolution?

How are they computed at 1 minute sampling?
**P5L4-L8 Response:** The new paragraph was included to explain the conversion ZTD to PWV. A surface meteorological station measuring pressure, temperature and humidity every minute, was installed at a GPS antenna and coupled with the GPS receiver. These information are presented in the section 2.2 (GPS data collection) The TM values were calculated applying the model suggested by Sapucci (2014) in the same sampling rate that pressure and temperature were measured (1 minute). The sampling rate of Tm values was included in the manuscript.

23) **Referee#2:** Line 15: "The sampling rate of the GPS-PWV values was 1 minute." Again, if the sampling rate is 1 minute, that implies PPP products cannot be sampled at 5 minutes. Have you done a specific GPS data processing to compute your own PPP products to obtain a sampling rate of the GPS-PWV values at 1 minute?
**P4L25-L27 P5L8-L11 Response:** The data processing was done only applying the GOA-II software. The last version of this software is able to estimate parameters with high temporal resolution (one minute). The GOA-II software re-sampling the GPS satellite ephemeris using the Orbit Integrator package which starts from specified initial orbits from JPL and uses several force models to integrate the dynamic equation to produce the orbit of each GPS satellite. These force models are applied to treat the gravity force, solar pressure, atmospheric drag, earth tides and the relativity. The solution are obtained by numerical integration given the initial conditions. This information was added in the new version of the manuscript.

24) **Referee#2:** Line 16: "problems with the satellite ephemerides" Can you explicit these problems? I don't understand why you got a problem with it.
**P5L10 Response:** In some few periods the JPL ephemeris files are incomplete.

Ephemeris from IGS are not used to avoid discontinuity in the PWV time series.

25) **Referee#2:** Lines 16-17: "unavailable pressure measurements" Is it possible to complete it with a meteorological model the problem was not so important to solve it? **P5L10-L11 Response:** The most important in the data processing is avoid the inclusion of uncertainty, which contaminate the global analysis of wavelet power spectrum. The values from meteorological model have large uncertain and the temporal resolution is of 6 hours over Brazil.

26) **Referee#2:** Lines 17: "other unknown causes": If we consider GPS data from receiver, orbit/ clock products for PPP processing and pressure measurements, I don't see which other unknown causes can be possible. **P5L11 Response:** An example of unknown causes would be the traffic of the people and vehicles around the GPS antenna during process of instrument maintenance, which multi-path can be generate. This phrase was rewritten and the term "other unknown causes" was excluded.

**Subsection "2.3 Precipitation time series**

27) **Referee#2:** Line 23: "very small spatial scale": Provide an order of magnitude. **P5L18 Response:** The sampling area of the Joss-Waldvogel brand acoustic impact disdrometer (model RD 80) used in this work is of the 50 cm2. This information was included in the manuscript.

28) **Referee#2:** line 5-6 "The dimensions of the precipitation area that influences the GPS-PWV is a key factor in the development of this study": If it is the case, you should explicit what you have really done and not summarize quickly your tests to directly provide the area of 22x22: 1. "Different areas were tested": explicit 2. "found to be more representative of the observed area by GPS": what criteria have been used? 3. "better for exploring the correlation between the precipitation occurrence and GPS-PWV": Could you please provide quantified results?

**P5L28-P6L10 Response:** Three areas around GPS antenna were tested: 4.4km per 4.4km, 12km per 12 km and 22km per 22km. The Fig. 1 (final of this response latter) shown the first precipitation event observed, in which a study about the radar data in different area around GPS antenna were tested and compared with GPS-PWV time series. It is possible to observe that even during strong precipitation registered by radar taken in count the areas 12km per 12 km and 22km per 22km, the PWV time series not change the increase tendency, but before the precipitation occur in the area 4.4 km per 4.4km the PWV present the peak and decrease the values, after some moment the stronger precipitation begin. Note that during the PWV increase (called PWV-Jump) precipitation over GPS antenna not is observed. As we already mention in the text this area were used here are not should be used in other experiment without suitable investigation. The correlation between GPS-PWV time series and precipitation is very explored in this study by wavelet and lag analysis .

The rainfall area employed in this study is only a reference for the description of GPS-PWV jump. For instance, if a raingauge is employed only an area of 20 cm radius is recorded and it is considered represents the rainfall from a large region. From another side, if the whole radar area is employed a rainfall over 100 km radius is recorded and could be associated to the GPS-PWV. In both cases representativeness and the
lead time (for a nowcasting application) should be considered. In the raingauge case, rainfall will be underrepresented because only rainfall over the raingauge will be considered and for the radar case rainfall far from the GPS-PWV will be considered and cannot be associated to the local increase in PWV. The representativeness of the GPS measurement is still an open question because it depends from the vertical distribution of the water vapor and mainly from the combination of elevation angles of the GPS satellite and the elevation angle threshold employed in the PWV processing. Actually, the area employed presented as the best for this location, but it is expected to vary as function of the region and satellite configuration. These information and discussion, such as the tested areas, are included in the manuscript.

29) **Referee#2:** Line 14: "around" Can you specify?
**P6L10 Response:** The square area around the GPS antenna was better specify in the new version of the manuscript.

30) **Referee#2:** Line 22-24: "the statistical measurements calculated from the radar data were in the 95th percentile of the intensity of the precipitation observed in the area of 4.4 km per 4.4 km around the GPS antenna" OK it is a statistical way in order to examine the intensity of the precipitation.

- 1. Lack of data DOY 319 from Xpol radar?

- 2. b and c should be at the same scale.

- 3. On 3 rainfall events, precipitation intensities are above 125 mm/h according to Disdrometer (Fig. 2b) whereas the 95th percentile of the intensity of the precipitation observed by Xpol radar radar are below 100 mm/h (Fig. 2c): It seems strange that the disdrometer measures a statistical anomaly 3 times on 20 (>95th).

.

**P4L13-L14 Fig.2 P6L29-L32 Response:** (1) Due to a technical problem the radar was turned off during the 12:32 of the DoY 318 until 19:41 of the DoY 319, as indicated in Fig.2. This information was included in the manuscript. (2) The scale in plot (c) was changes para o range between 0 and 200, as suggested. (3) The total precipitation from disdrometer (Parsivel) is always larger than the one measured by raingauge and radar. The problem is the large droplet concentration, normally the integration cut off the largest raindrops in 4 mm (see Giangrande et al, 2016 JGR , doi: 10.1002/2015JD024537). In our case we didn't apply any filter and therefore the total value is larger than the radar and raingauge as you can see in Figure 1. We clarify this point on the text.

**Page 6 "3 Wavelet analysis"**

31) **Referee#2:** Lines 15-16: "In this study, both continuous and discrete wavelets are investigated to achieve intra- and interrelation analysis, respectively." Distinguish what wavelet decomposition should be used to answer to what scientific questions.
**P9L12-L25 Response:** We agree that the description of the methodology applied in the wavelet analysis was very poor in details. It was rewritten in new section 4 and information about the discrete and continuous wavelet are included.

**Page 7**

32) **Referee#2:** Lines 6-7: "The methodology employed to process the GPS data in one-minute intervals did not provide any additional information. Fig. 3 shows that the

GPS-PWV energy variability begins to be significant only for time scales longer than 16 minutes." The influence of stochastic constrains applied on temporal evolution of ZWD during the GPS data processing must be taken into account. If you have used a too small random walk parameter, that could explain what you have observed.

**P10L15-L16 P14L17-L20 Response:** This phrase was removed because the range of power spectrum associated with color scale used in the Fig. 6 did not permit showing the time scale lower than 16 minutes, which was selected to make possible see the results. Information about this range and the color scale used in Fig. 6 was included in the paper. The parameter used in the random walk process noise for zenithal tropospheric delay estimate was the default value suggested by JPL (8.333E-8 km per square-root second). We agree that these values can influence the variability of the PWV estimates in high frequency, and others values should be tested. Some comment about the impact of these values in the GPS-PWV variability was included in the manuscript in the final of the derivative analysis sections.

33) **Referee#2:** line 8: "Therefore, the one-minute time series representativeness is not a limitation, and if there is noise, it is white noise." Can you explicit and prove it?

**P10L15-L16 Response:** This phrase was removed (see specific comment 32-Referee2).

34) **Referee#2:** It is obvious that a jump in a time series will produce what is described lines 13-15: "there are expressive changes in the power between different time scales in those cases in which an increase in the power of the oscillation from low to high frequency is observed." It is the well known example of the Wavelet power spectrum of a Dirac signal.

**P10L21-L23 Response:** Although it is well known the example of the Wavelet power

spectrum of a Dirac signal, we believe it is important to reinforce it in this case. The phrase was rewritten and the GPS-PWV jump discussed the previous section of the wavelet analysis (as suggested by reviser) was mentioned here. Similar sentence in the conclusion section were also rewritten.

35) **Referee#2:** Figure 3: GPS-PWV presents a jump DOY 358 with a strong signal in the Wavelet Power Spectrum but Xpol radar did not detect any precipitation : have you any comment on it?
**P10L30-L32 Response:** At 23:00 DoY 358 a precipitation was observed, which was included in the Lower tercile. The problem pointed by reviser is pertinent if taken in count the GPS-PWV jump observed in the final of DoY 357, which the radar did not detect precipitation. If the nowcastig tool was based on GPS-PWV jump, this case would be a false alarm, which is a metric used by evaluate the its skill. A comment about this case was included in the text.

36) **Referee#2:** lines 31-32: "The results show that the wavelet correlation between the PWV and precipitation intensity is more evident and significant for the time scale between 32 and 64 minutes". Again it would be clearer to speak about PWV jump before speaking about the lag between GPS-PWV jump and rainfall and introduce wavelet to determine the lag precisely.
**P12L1-L4 Section 4 Response:** This phrase was rewritten in the new version of manuscript. The section 3 titled "Behavior of PWV time series before precipitation events: the GPS-PWV jumps" present the GPS-PWV Jump before wavelet analysis. The section 3 and 4 from previous version of the manuscript were merged, as suggested in your general comments, in the new section 4, denominated "High temporal resolution GPS-PWV time series analysis", which wavelet analysis and time lag correlation analysis are explored to evaluating which the timescale of the GPS PWV oscillations are associated with a more intense rainfall. This section was divided in the

following subsection: - 4.1 Wavelet analysis - 4.1.1 Wavelet power spectrum analysis - 4.1.2 Wavelet cross-correlation analysis - 4.2 Time lag correlation analysis - 4.3 GPS-PWV derivative analysis: potential for nowcasting application.

37) **Referee#2:** lines 31-32: Correlations shown figure 5 do not exceed 0.15 and do not look significant: it seems clear there is a lag between GPS-PWV and rainfall and the evaluation of this lag seems good but the correlations of figures 5 and 6 diminish the strength of the demonstration.

**P11L15-L16 Fig8 Fig.9 Response:** We agree the correlations are not very high, however, significance test shown that this results are statistically significant into 95% confidence interval, which taken into consideration a Gaussian Distribution after applying the Fisher's Z Transformation. This comment was included in the paper and information about the bars were included in the Captions (Fig. 8 and 9): "The 95% Confidence Interval for each WCC is estimated considering a Gaussian Distribution after applying the Fisher's Z Transformation (Whitcher et al 2000)."

**Page 8**

38) **Referee#2:** "4 Behavior of PWV time series before precipitation events: the GPS-PWV jumps": I appreciate the meteorological interpretation of GPS-PWV jump during severe weather events but this interpretation seems to be founded only on a single reference (Adams et al., 2013): Is there any other references on these meteorological processes during severe weather events?

**P7L1-P8L24 Response:** We explore the results reported by Adams et al. (2013), because they showed the behavior of PWV time series before precipitation events over

same region that we discussed here. However, we agree that would be interesting include another papers. So we included Kursinski et al (2008), Solheim et al. (1999) and Shi et al. (2015).

**Page 10**

39) **Referee#2:** line 3-5 "This result corroborates the pattern observed in Fig. 7, showing the GPSPWV maximum before the precipitation event and its minimum after the maximum precipitation" It would be clearer if the zoom of figure 7 has shown precipitation.
**Fig. 3 Response:** The precipitation values were included in zoom of Fig. 3, as suggested by reviser.

40) **Referee#2:** "4.2 GPS-PWV derivative analysis" Line 20-21: "Fig. 9 clearly shows an expressive change in the pattern of the derivative distribution as a function of the different precipitation intensity terciles." : I suppose you did it but did you check that for each severe weather event of upper tercile you got around 7.8% of GPS-PWV derivative >+9.5 mm.hẼE̜(-1) and around 5.47% of GPS-PWV derivative < -9.5 mm.hẼE̜(-1) because if I have well understand, you proposed section 4.3 to use this criterion to detect severe weather events for nowcasting application.
**P13L30-P14L5 Fig.13 Response:** In the fact, this analysis about the derivative >+9.5 mm.hẼE̜(-1) and <-9.5 mm.hẼE̜(-1) was not present in the section 4.2. Some comments about this results had been present in the 4.3 section, which the reviser 3 suggested to change it to section 4.2. (see specific comment 29-referee3). The section 4.2 was merged with 4.3 in the new version of manuscript. These derivatives are better explored in the discussion of the results shown by new Fig. 13.

41) **Referee#2:** line 28-30: "The wavelet analysis for the GPS-PWV time series was explored, and it clearly shows that during precipitation events, there are expressive changes in the power spectrum between different time scales, in which an increase of the power of the oscillation from low to high frequency is observed." It is a logical result due to the fact there is a jump in GPS-PWV time series during severe weather events (see comment 34).

**P15L8-L9 Response:** This phrase was rewritten to "The wavelet analysis for the GPS-PWV time series was explored to characterize the strong changes in the power spectrum between different time scales during precipitation events generated by the occurrence of the GPS-PWV jumps.".

Please also note the supplement to this comment:
http://www.atmos-meas-tech-discuss.net/amt-2016-378/amt-2016-378-AC3-supplement.pdf
* * *
[Figure]

**Fig. 1.**

**Supplement:**

**GPS-PWV jumps before intense rain events**

Luiz F. Sapucci[1], Luiz A. T. Machado[1], Eniuce Menezes de Souza[2], Thamiris B. Campos[3]

[1]Centro de Previsão de Tempo e Estudos Climáticos, Instituto Nacional de Pesquisas Espaciais, Cachoeira Paulista, Postal Code: 12630-000, Brazil.
[2]Departamento de Estatística - Universidade Estadual de Maringá, Maringá, Postal Code: 87020-900, Brazil
[3]Programa de Pós-Graduação em Meteorologia, Instituto Nacional de Pesquisas Espaciais, São José dos Campos, Postal Code: 12227-010, Brazil

*Correspondence to*: Luiz Sapucci (luiz.sapucci@cptec.inpe.br)

**Abstract.** A rapid increase in atmospheric water vapor is a fundamental ingredient for many intense rainfall events. High-frequency precipitable water vapor (PWV) estimates (one minute) from a Global Positioning System meteorological site (GPS) are evaluated here for intense rainfall events during the CHUVA Vale field campaign in Brazil (November-December 2011), in which precipitation events of differing intensities and spatial dimensions, as observed by an X-band radar, have been explored. A sharp increase in the GPS-PWV prior to the more intense events has been found and termed GPS-PWV "jumps". These jumps are associated with water vapor convergence and the continued formation of cloud condensate and precipitation particles. The correlation and lags between the high temporal resolution GPS-PWV time series and rainfall events are evaluated. A wavelet cross-correlation analysis shows that there are important spikes in the PWV that precede the more intensity/extension rainfall events on scales related to time periods from about 30 to 60 minutes. The GPS-PWV time-derivative histogram for the period of 60 minutes before the rainfall event reveals different distributions for higher intensity and extension events. This feature could indicate the occurrence of severe precipitation and consequently has the potential for application in nowcasting activities.

**1 Introduction**

The application of the Global Positioning System (GPS) tropospheric-induced signal delay to estimate the precipitable water vapor (hereafter, GPS-PWV) is a good example of an indirect solution for quantifying atmospheric humidity. The magnitude of this delay is related to the integral of the refractivity index of the air as a function of temperature, pressure and water vapor (Bevis et al. 1992) on the optical path followed by the GNSS signal. The wet component of this delay provides the precipitable water vapor (PWV) (Bevis et al. 1994), with an error of approximately 5% under all weather conditions (Wolfe and Gutman 2000) relative to other measurement techniques (Sapucci et al. 2007) and in near real time (Rocken et al. 1994 ). The methodology employed in GPS data processing has been improvement continually to minimize the uncertainty and the PWV estimate has been determined with an accuracy better than 2 mm (Moore et al. 2015; Shangguan et al. 2015) Although the vertical humidity structure is not captured in GPS-PWV estimates, the great advantage of GPS-PWV (in addition to its

Comentário [s1]: Reviser2E1

Comentário [s2]: Reviser2E2

Comentário [s3]: Reviser2E3

all-weather capacity) is its high temporal resolution (minutes) (Zumberge et al. 1997). An important application of the GPS-PWV estimate is its assimilation into the Numerical Weather Prediction process, which has a positive impact on short-range forecasts of humidity fields and, consequently, better precipitation forecasts for heavy rainfall events (Cucurull et al. 2004; Bennitt and Jupp 2012). Other applications become viable in dense networks and transects. GPS-PWV has been useful for: studying the diurnal cycle of convective instability in Japan (Sato and Kimura 2005); investigating water vapor variability during a mistral/sea breeze event in southeastern France exploring GPS-PWV tomography (Bastin et al. 2005); studying water vapor diurnal cycle over African continent (Bock et al. 2008); evaluation of the GPS-PWV values before precipitation (Kursinski et al. 2008); tracking water vapor advection over Amazonian (Adams et al. 2011); GPS-PWV tomography have been used to investigate the water vapor distribution related to convective rainfall events and better understanding and quantification of the hydrological cycle in southeastern France (Brenot et al. 2014); it has been used to propagate convective events and determining topographic effects on the evolution of convection (Adams et al. 2015); studies of convective mesoscale events during the North American Monsoon (Serra et al. 2016) and nowcasting thunderstorm activities, as foreseen by Jerrett and Nash (2001). Guerova et al. (2016) showed development and test of new multi GNSS (Global Navigation Satellite System) products to forecasting of severe weather and emphasized the fact that for short-term, high-resolution forecasting or nowcasting models require more detailed humidity observation, e. g. GPS-PWV estimates.

The relationship between the occurrence of intense rainfall and high concentrations of atmospheric water vapor is well known and has been explored in nowcasting applications. Muller et al. (2009) described a model for the relationship between the PWV and convective precipitation event in the tropics. PWV data from a microwave radiometer (MWR) with high temporal resolution (6 minutes) have been used to analyse and better understand the temporal relations of column water vapor and tropical precipitation (Holloway and Neelin, 2010). Chan (2009) evaluated the performance of a ground-based microwave radiometer in intense convective weather and reported an increasing degree of instability of the troposphere before the occurrence of heavy rain. In addition, he compared PWV from GPS and radiometer, the result showed that radiometer exhibited rather rapid fluctuations during intense precipitation, which are not observed in the GPS-PWV data. Madhulatha et al. (2013) reported a sharp increase in the PWV values approximately 2–4 hours prior to the occurrence of thunderstorms and developed a nowcasting technique using PWV values and 7 other thermodynamic indices from microwave radiometer observations. The relationship between deep convective activity and occurrence of intense rainfall and the temporal evolution of GPS-PWV has been studied for well over a decade. Mazany et al. (2002) developed a lightning prediction index for Florida based on the GPS-PWV magnitude and its temporal evolution. This index is a binary logistic regression model based PWV-GPS and other two variable predictors. The plot of the GPS lightning index time series showed a pattern several hours prior to a lightning, which was tested as forecasting tool of this events. Nowcasting employing GPS-PWV was reported by de Haan et al. (2004), who demonstrated its viability in improving thunderstorm and heavy precipitation forecasts during a cold front passage. De Haan (2006) developed a method for inferring the atmospheric stability from a nonisotropic GPS path-delay signal (slant delay). Book et al. (2008) carried out studies of the West African

[revised manuscript text omitted]

To ensure the quality of the PWV time series with high temporal resolution required in this study, in the data-processing strategy adopted the known uncertainty sources were taken into consideration applying the recommended models and adjustment of parameter exploring available stochastic models. The latest version of the GOA-II software (version 6.3) was used, which estimates parameter with high temporal resolution exploring the sophisticated orbit integrator package to estimate GPS satellite position in each epoch. The ocean tide model FES 2004 (Lyard et al. 2006), recommended by the International Earth Rotation and Reference Systems Service (IERS Conventions 2010) was applied in this processing. Method of antenna absolute calibration (Schmid et al. 2007) was applied by GOA-II to ensure the correct phase center variation of the satellites and receiver antennas using parameters provide by IGS web site (Montenbruck et al. 2015). The data processing with GOA-II software to obtain ZTD estimates was done selecting the Global Mapping Function (Boehm et al. 2006) and the sampling rate of the ZTD estimates of 60 seconds. The others possible parameters that can be selected were

Comentário [s20]: Reviser2E17

Comentário [s21]: Reviser2E30a

Comentário [s22]: Revisor2E4b

Comentário [s23]: Reviser2E19

Comentário [s24]: Reviser2E18

Comentário [s25]: Reviser2E23

Comentário [s26]: Reviser2E20

Comentário [s27]: Reviser2E21

used the configuration basic suggested by JPL. As the configuration items associated with ZTD estimates and respective values used can impact in the variability of the ZTD in high temporal resolution (basic information used in this study), they are listed in Table 1 in order to highlight them.

**Comentário [s28]:** Reviser2E18

[revised manuscript text omitted]

The high temporal resolution obtained with the GPS-PWV enables the evaluation of high frequency variations and their relationship with intense precipitation events. The GPS-PWV time series shows a well-defined sharp increase before the occurrence of precipitation, as reported by Kursinski et al. (2008) as a rapid rise in PWV preceding the rain events. Shi et al.

5  (2015) using GPS -PWV to monitoring the water vapor variation shown that ascending and descending patterns of GPS-PWV can be identified before and after each rainfall event. There are strong oscillations, generating a significant increase in the total water vapor content until a maximum is reached. Subsequently, a strong GPS-PWV reduction is observed, and after a short interval, the precipitation also reaches a maximum peak. Here, this sharp increase is called GPS-PWV jump. Fig. 3 shows a typical case exemplifying the PWV behavior before precipitation occurs on DoY 341; this was one of the strongest

10  events registered during the CHUVA Vale experiment. Before the severe precipitation begins, the GPS-PWV follows several pulses, increasing the value and forming the PWV jump, until it reaches a peak of maximum value. After the GPS-PWV crest, a decreasing period is observed some minutes before severe precipitation. Fig. 3 clearly shows this configuration of a crest in the GPS-PWV time series around precipitation (composed of several pulses) and its subsequent decrease immediately before the beginning of stronger precipitation.

15  This GPS-PWV behavior before precipitation occurs not only for more intense events but also for lower rainfall rates. Fig. 4 shows other GPS-PWV jumps observed before rain events with different intensity/extension occurred on: DoY 315 (41% of precipitation fraction above 35 mm h$^{-1}$), DoY 332 (5%) and DoY 358 (only 1% of precipitation fraction above 35 mm h$^{-1}$). This figure shows the intensity of these sharp increases in GPS-PWV are larger before more extensive precipitation. Table 2 presents all precipitation events in which the rainfall above 20 mm h$^{-1}$ was observed by XPol radar in the area of 4.4 km per

20  4.4km around GPS antenna. This table presents the DoY and time of maximal radar precipitation of the each event and the respective fraction rain above 20 mm h$^{-1}$, 35 mm h$^{-1}$ and 50 mm h$^{-1}$. In order to obtain an overview of the hardiness of the GPS-PWV jumps feature before precipitation, Fig. 5 show the composite mean of GPS-PWV time series from 60 minutes before to 60 minutes after maximum observed precipitation (fraction rain above 35 mm h$^{-1}$) for 18 events listed by Table 2. The composite presented in Fig. 5 is normalized by maximum GPS-PWV values before precipitation. The composite mean

25  shows that the GPS-PWV jump is strongly remarkable before the maximum precipitation, which the maximum of the composite mean is observed in 30 minutes and lower dispersion in 25 minutes before the maximum precipitation. The time lag between the maximum GPS-PWV and the time of the maximum rainfall is presented in a sub plot in Fig.5. It is important highlight that the there are many precipitation events of lower intensity and extension in which the GPS-PWV jumps are not observe. In these cases, the maximum GPS-PWV is observed in the maximum precipitation, consequently the time lag is

30  zero (37% of the cases evaluated). In this histogram, it is clear there is a range of lag time between the GPS-PWV and precipitation maximums, the precipitation area and the temporal resolution of GPS-PWV here employed have an impact on this lag time. For instance, Adams et al. (2013) using rain gauge and PWV with sampling rate of 30 minutes over Amazonian region found out lag zero, however, in this study we shows that the GPS-PWV signal can represent a precipitation area and

**Comentário [s39]:** Reviser2E38

not only a punctual measurement over the GPS antenna. An analysis of the time lag correlation is necessary to define this interval between the maximal GPS-PWV and precipitation, which is carried out in the next section.

The physical explanation for this behavior could be explained by different physical processes. First, the water vapor may increase through low-level moisture convergence. The variation of the moisture convergence generates a sequence of pulses of positive increases in the PWV value. The sequence of pulses of positive increases in the PWV could be a result of several physical processes. Some of the physical process that can explain these pulses could be low-level water vapor convergence forced by gravity wave (Raymond 1987) or simply unstable surface parcels accelerating upwards. After the increase of GPS-PWV at the crest of the jump, rainfall starts and PWV starts to decrease. The lag time between the crest and the maximum precipitation can vary from one region to another, from one rainfall cell to another because it depends on the cloud condensation nuclei and the precipitation efficiency, normally a function of the wind shear. In this study using precipitation measures based on area about the GPS antenna, we found a time lag between these maximum. Addams et al. (2013) considered that the conversion of water vapor to liquid water and precipitation are of second order during the process of PWV increasing. It is probably true during the phase of cloud formation, however, when the precipitation starts, water vapor decreases due to the formation of liquid water. The conversion of the water vapor to liquid water changes the dielectric medium, where the refractivity is induced by the displacement of charge (Solheim et al. 1999). While the refractivity from water vapor is due to the polar nature of the water molecule, the GPS phase delay induced by liquid water (hydrometeor) is proportional to the electric permittivity of the formed dielectric medium and, consequently, much lower than the delay generated by water vapor. Another important physical mechanism is the storm downdraft that is dryer and colder that the ascending moisture air and this downdraft can also contributes to the decrease in PWV. In addition, PWV decreases after precipitation starts can be simply associated to the final process of surface convergence as function of the rainfall and downdrafts on the surface, or by the advection process forced by the shear and storm movement. Given the limitations of the observations, our interpretation of the physical mechanisms responsible for the jumps remains speculative. To understand what the physical mechanisms responsible for the pulses and GPS-PWV jumps, a specific field campaigns design are needed.

**Comentário [s40]:** Reviser2E38

**4 High temporal resolution GPS-PWV time series analysis**

The high temporal resolution GPS-PWV time series and precipitation in different intensity and extension are evaluated and the PWV-GPS jumps are characterized. The wavelet analysis are explored to evaluating which the timescale of the GPS PWV oscillations are associated with a more intense rainfall occurrence and time lag correlation analysis are used to determine the lag between the rainfall intensity and the GPS-PWV time series. Additionally, PWV-GPS derivative analysis is explored and its potential for nowcasting application is discussed.

**4.1 Wavelet analysis**

Wavelet analysis was used to perform a detailed analysis of the GPS-PWV time series and to evaluate the variability within different time scales (denoted here as intra-relation), as well as to assess the relationship between the GPS-PWV time series and the precipitation time series (denoted here as interrelation wavelet analysis). This methodology enables simultaneous decomposition of the PWV time series as a function of time and frequency (Daubechies 1992). Consequently, accessing to the information regarding the signal amplitude/frequency and its variation as a function of time becomes possible.

To perform the intra-relation analysis evaluation of how spectral characteristics change over scales ($s$) and time ($t$), but with highly redundant information, the continuous wavelet analysis (Torrence and Compo 1998) was used to estimate the wavelet power spectrum. Thus, a decomposition of the GPS-PWV data into time-variability space allows an evaluation of the main frequencies composing the GPS-PWV time series during intense rainfall events. With continuous analysis, some hidden features of the time series can be identified, e.g., in which scale are the most representative behaviors of the time series. Continuous analysis is often easier to interpret because its redundancy tends to reinforce the traits and makes all information more visible. However, for some specific choices of values for time and frequency, it is possible to apply a discrete wavelet transform, which does not lose important information and has advantages of implementation and computational effort. This is the case of the non-decimated discrete wavelet transform (NDWT), also called Maximal Overlap Discrete Wavelet Transform, which can be seen as a compromise between the discrete wavelet transform (DWT) and CWT because of its redundancy, but not as redundant as CWT. The NDWT can be computed similarly to the ordinary DWT but without subsampling (decimation), ensuring the translational invariance, which is ideal for analysing time series, especially interrelations between different time series. A time-variant transform disrupts the lag-resolution in a cross-correlation analysis. Furthermore, estimators calculated using the NDWT are considered more preferable because they are asymptotically more efficient than the estimator based on the DWT (Percival and Walden 2000). As the bivariate relationship between two time series is essential for this research, a wavelet cross-correlation (WCC) constructed from NDWT (Whitcher et al. 2000) is ideal for analysing different scale structures and the interrelations of the dynamic behaviour of two time series, as well as the lead-lag relationships. Some lead-lag relations that could not be distinguished in the usual cross-correlation can be investigated in the WCC, which decomposes the cross-correlation on a scale-by-scale basis. Thus, a specific scale may be associated with water vapor convergence before precipitation occurrence, also providing a lead time for nowcasting, a specific time scale for calculations and additional information to understand the physical mechanisms associated with intense rainfall events in this region.

> **Comentário [s41]:** Reviser2E31

Considering the large quantity of available discrete mother wavelets, some of the most used in the literature were evaluated: Daubechies with 4, 6, 8, and 16 coefficients, denoted by D4, D6, D8, D16, and Daubechies Least Asymetric with 8, 16, and 20 coefficients, denoted by LA8, LA16, and LA20, respectively. To verify the statistically significance of the estimated wavelet correlations, the 95% confidence interval was estimated considering a Gaussian Distribution after applying the Fisher`s Z Transformation (Whitcher et al. 2000).

Because of implementation aspects considered in this study, the two time series were restricted to a power of two length, with 65,536 ($2^{16}$) observations, excluding one day at the beginning and another at the end of the GPS-PWV series. The time span corresponds to 45 days, 12 hours and 16 minutes, beginning on DoY 314 (00 UTC 10 November) and finishing on DoY

5    359 (1216 UTC 27 December). The same time series length was used for both intra- and interrelation analyses. Because each wavelet scale $j$ in the interrelation analysis corresponds to a frequency band from $2^j$ to $2^{j+1}$, its inversion allows the interpretation in terms of a period of time also in a dyadic interval. Thus, WCC enables the identification of the most important scale of the PWV oscillation during precipitation events. Furthermore, the WCC of GPS-PWV and precipitation time series also allows evaluating the lead-lag correlation that may exist between these time series for the different time

10   periods.

**4.1.1 Wavelet power spectrum analysis**

For an intra-relation analysis, the wavelet power spectrum of GPS-PWV and 95[th] percentile of the precipitation intensity time series are presented in Fig. 6. To emphasize the highest-frequency oscillations, the power spectrum in Fig. 6 shows the scales that represent the period below 512 minutes (~8.5 h). The PWV diurnal cycle presents strong power along all time

15   series, and consequently, it was not taken into consideration in this analysis. The range of power spectrum associated with color scale used in Fig. 6 is for scales related to time periods larger than 16 minutes.

The PWV series with a one-minute temporal resolution presents oscillations of high frequency; however, the frequency of occurrence of rainfall events is very low. For this reason, it is necessary to take into consideration a long time period (with several precipitation events) and a short time step, e.g., the one-minute interval used in this study. Consequently, in a general

20   analysis of the PWV wavelet power spectrum, it is difficult to clearly discern which power is associated with each time step. However, a more specific analysis of the wavelet power spectrum during precipitation events indicates that there are, as expected in function of described GPS-PWV jump, strong changes in the power between different scales; cases with an increase in the power of the oscillation from low to high frequency are observed. This result indicates that PWV oscillations on scales related to time periods smaller than 128 minutes occur more frequently during precipitation events than in periods

25   without rain. Fig. 7 presents the same wavelet power spectrum presented in Fig. 6 but with an enlargement applied for precipitation events observed during the period from DoY 340 to DoY 343. The analysis of the power spectrum in these cases makes the result discussed for Fig. 6 more understandable. For clarity, a vertical line was drawn for each precipitation maximum. Fig. 7 shows that the power of the PWV oscillations between 128 and 32 minutes is stronger during more intense precipitation events (for example, the events that occurred on DoY 341 at 1836 UTC and DoY 342 at 1636 UTC) than

30   during light rainfall events (events observed during DoY 343) and periods without rain (DoY 340). It is interesting to observe that GPS-PWV presents a jump at the end of the DoY 357 with a strong signal in the wavelet power spectrum, but XPol radar did not detect any precipitation. The wavelet power spectrum (Fig. 6) also shows the impact of the GPS failures

**Comentário [s42]:** Reviser2E32

**Comentário [s43]:** Reviser2E33

**Comentário [s44]:** Reviser2E34

**Comentário [s45]:** Reviser2E35

that occurred during DoY 331 and 348, which are unfortunately very close to the other intense precipitation events that occurred on DoY 332 and 348. For this reason, these cases are not spotlighted in the wavelet analysis.

**4.1.2 Wavelet cross-correlation analysis**

Before presenting the inter-relation results, one might question about the robustness of the method regarding the choice of mother wavelets. We estimated the wavelet correlations using Daubechies and Least Asymmetric wavelets with different filter lengths (coefficient number) as presented in Section 4.1. The results were quite similar independently of the mother wavelet, but the correlations were maximized when the mother wavelet has a larger filter or more coefficients (D16, LA16, and LA20). Thus, all estimated wavelet correlations were presented using the LA20 mother wavelet. Fig. 8 shows the wavelet correlation and its 95% confidence interval between the GPS-PWV and the 95[th] percentile of the precipitation intensity as a function of the wavelet scale, represented by the respective time periods, considering the lag zero. In this analysis, the precipitation data from XPol radar, originally with sampling rate of 6 minutes, were linearly interpolated to one-minute rate. The results show that the wavelet correlation between the PWV and precipitation intensity is stronger for the scale related to the period between 32 and 64 minutes, indicating the scale on which the most important GPS-PWV oscillations associated with precipitation events occur. After this scale, the correlation decreases, followed by another increase due to the influence of the diurnal cycle. Although the correlations are not very high, 95% confidence interval showed that these results are statistically significant.

Comentário [s46]: Reviser2E37

To evaluate the results presented in Fig. 8 for different intensity and extension of precipitation events, Fig. 9 shows the wavelet correlation for lag zero between GPS-PWV and precipitation fractions as a function of the period bands for different rain fraction intensities (>20 mm h$^{-1}$, Fig. 9a; >35 mm h$^{-1}$, Fig. 9b; and >50 mm h$^{-1}$, Fig. 9c). The 95% percentiles give an information about the maximum rain rate (which can be only one point) on the area of 4.4 km per 4.4 km around the GPS antenna. The rain fraction gives an information about the fraction of these studied area covered by rain rate above of these thresholds. There are some events where the rain intensity can be high and the fraction small, for instance, the cases of isolate clouds. Therefore, the area fraction presents a more close representation of rainfall events related to low level convergence because it gives information about the amount of liquid water in the area. The peak of the wavelet correlation observed in the time period from 32 to 64 minutes is significant for the three rain thresholds. The plots of Fig. 9 also show when only heavy to torrential precipitation events are taken into consideration (hereafter called intense rain events), stronger wavelet correlation is observed.

To perform the lead-lag analysis, the WCC is showed in Fig 10 between GPS-PWV and rain fraction for different intensities (>20 mm h$^{-1}$, Fig. 10a; >35 mm h$^{-1}$, Fig. 10b; and >50 mm h$^{-1}$, Fig. 10c). Although the precipitation and GPS-PWV time series present very distinct behaviors, the WCC permitted the identification of the correlation in lead-lag of about 30 minutes mainly for scales related to time periods from 32 to 64 minutes, indicating which GPS-PWV oscillations are important for predicting precipitation events. This results show also that considering the rain fraction for stronger intensities (>50 mm h$^{-1}$), some statistically significant correlations in lead-lag of about 30 minutes also appear stronger from period bands of 4- 16

and 16-32 minutes. This result indicates that the GPS-PWV carries some information, mainly, on the scale related to the period from 32 to 64 minutes. That signals the occurrence of precipitation events of large intensity and extension, which suggests that the GPS-PWV jumps are, for these storms evaluated, concentrated in this scale emphasizing its potential in a nowcasting application. It is also important to highlight that the correlation is larger on this scale than on adjacent scales.

**Comentário [s47]:** Reviser2E36

5 Furthermore, these results indicate that, although GPS-PWV jump before precipitation occurs not only for more intense events, the intensity of these oscillations associated to jumps is greater before the most intense and extensive ones. However, this feature should be considered as specific to each region because the time scale between the humidity convergence and rainfall may depend on many physical processes and environmental conditions (e.g., gravity wave-induced convergence, wind shear or thermodynamic instability).

10 **4.2 Time lag correlation analysis**

The relationship between the rainfall intensity (or rain fraction) and the GPS-PWV is different for each event; however, the GPS-PWV peak is a well-delineated pattern. The time interval between the moment of the PWV crest and the maximal precipitation can vary among cases. The WCC shows on which scale the correlation between GPS-PWV and precipitation time series is higher, as well as the lead-lag interrelation between them. However, evaluating the lag correlation for positive

[revised manuscript text omitted]

**Comentário [s49]:** Reviser2E40

and are significant before the most precipitation-extensive events. Fig. 13 shows the time series of GPS-PWV derivative before precipitation events of the upper tercile, and emphasizes the occurrence of this stronger derivative (above +9.5 mm h$^{-1}$) observed at least three of the evaluated events. These stronger derivatives, associated to PWV pulses, occur with larger frequency in the period from 55 to 25 minutes before the maximum precipitation, which is GPS-PWV jump is observed and the pick of PWV maximum also occur.

Although the GPS-PWV pattern before precipitation described in the previous section is well defined, its use as an index for the occurrence of severe storms is not simple. Several studies have taken into account the intensity of rainfall events. Therefore the use of only a maximum threshold from a GPS-PWV derivative, as suggested by Iwabuchi et al. (2006) and Shi et al. (2015), is not sufficient to predict intense rainfall, several atmosphere processes are very complex and highly nonlinear. The intensity of the precipitation is highly correlated with the intensity of the PWV value, which is formed by a succession of pulses of positive increases in the PWV value. Therefore, the occurrence of more intense rain is signaled not only by the maximum derivative but also by the increase in the frequency distribution of the positive and negative derivatives before the occurrence of precipitation.

The result found by GPS-PWV derivative analysis suggests that an algorithm for intense precipitation forecasting using the GPS-PWV should consider the following points: (a) increases in GPS-PWV positive variations compared with negative ones in which the median values of the variation in the last 60 minutes reach positive values and (b) a simultaneous increase in the population of the GPS-PWV derivatives above +9.5 mm h$^{-1}$. The value of the stochastic constrains applied on temporal evolution of ZWD during the GPS data processing can make influence the PWV variability in high temporal resolution. The value used in this study was the default, but some tests with different constraint can be carried out to identify the impact of this parameter in the derivative analysis before intense precipitation.

The GPS-PWV values evaluated in this study are post-processed, and an additional study is required to determine whether these estimates in real time are able to capture the jumps before the precipitation reported. True real-time processing for PWV estimates in dense and regional GPS networks has been explored in other studies related to nowcasting applications (Iwabuchi et al. 2006). De Haan and Holleman (2009) reported the construction and validation of a real-time PWV map from a GPS network combined with data from weather radar, a lightning detection network, and surface wind observations. They tested a nowcasting algorithm for three thunderstorm case studies and concluded that the GPS-PWV in real time can be helpful for the nowcasting of severe thunderstorms. Shi et al. (2015) studied the PWV estimates in real time for rainfall monitoring and forecasting and shown that this estimate has quality comparable with post-processed product. A significantly reduction in the latency was obtained with GPS data processing proposed by Shi et al. (2015), which demonstrated promising perspective of the PWV-GPS data for rainfall forecasting.

**Comentário [s50]:** Reviser2E32

**5 Conclusions**

This work evaluates the correlation between large and rapid increases in the GPS-PWV and the occurrence of rainfall events observed by radar during the CHUVA Vale experiment in Brazil. A detailed analysis of the GPS-PWV time series was carried out, and strong and sudden sharp increase composed predominantly by positive derivatives, before the precipitation events were identified and called as GPS-PWV jumps. In this process, a crest in the PWV series is remarkable before the precipitation events. Although this sharp increase can be observed for any precipitation event, it is preponderant before more intense and extensive precipitation events.

The wavelet analysis for the GPS-PWV time series was explored to characterize the strong changes in the power spectrum between different time scales during precipitation events generated by the occurrence of the GPS-PWV jumps. Additionally, the application of wavelet cross-correlation between the PWV and precipitation showed that important oscillations exist between these variables on the scale related to a time period from 32 to 64 minutes, which is stronger for events of large intensity and extension. These results corroborates with those reported by Adams et al. (2013), who showed that the strongest water vapor convergence is typically ~1 hour before heavy precipitation.

[revised manuscript text omitted]

Bock, O., M. N. Bouin,E. Doerflinger, P. Collard, F. Masson, R. Meynadier, S. Nahmani, M. Koité, K. Gaptia Lawan Balawan, F. Didé, D. Ouedraogo, S. Pokperlaar, J.-B. Ngamini, J. P. Lafore, S. Janicot, F. Guichard, M. Nuret. 2008: The West African Monsoon observed with ground-based GPS receivers during AMMA, J. Geophys. Res., 113 (D21105) doi: 10.1029/2008JD010327.

Boehm, J., B. Werl, and H. Schuh, 2006: Troposphere mapping functions for GPS and VLBI from ECMWF operational analysis data. J. Geophys. Res., 111, B02406, doi:10.1029/2005JB003629.

Brenot, H., A. Walpersdorf, M. Reverdy, J. van Baelen, V. Ducrocq, C. Champollion, F. Masson, E. Doerflinger, E., Collard, P., and Giroux, P. 2014: A GPS network for tropospheric tomography in the framework of the Mediterranean hydrometeorological observatory C´evennes-Vivarais (southeastern France), Atmos. Meas. Tech., 7, 553-578, doi:10.5194/amt-7-553-2014.

Calheiros, A. J. P., and L. A. T. Machado, 2014: Cloud and rain liquid water statistics in the CHUVA campaign. Atmosph. Research, 144, 126-140, doi:http://dx.doi.org /10.1016/j.atmosres.2014.03.006.

Chan, P. W., 2009: Performance and application of a multi-wavelength, ground-based microwave radiometer in intense convective weather. Meteorol. Z., 18, 3, 253-265, doi:10.1127/0941-2948/2009/0375.

Cucurull, L., F. Vandenberghe, D. Barker, E. Vilaclara, and A. Rius, 2004: Three-Dimensional Variational Data Assimilation of Ground-Based GPS ZTD and Meteorological Observations during the 14 December 2001 Storm Event over the Western Mediterranean Sea. Mon. Wea. Rev., 132, 749–763, doi:http://dx.doi.org/10.1175/1520-0493(2004)132<0749:TVDAOG>2.0.CO;2.

Daubechies, I., 1992: Ten Lectures on Wavelets. Society for Industrial and Applied Mathematics, 357 pp.

Davis, J. L., T. A. Herring, I. Shapiro, A. E. Rogers, and G. Elgened, 1985: Geodesy by radio Interferometry: Effects of Atmospheric Modeling Errors on Estimates of BaseLine Length. Radio Sci., 20, 1593-1607, doi: 10.1029/RS020i006p01593.

Giangrande, S. E., T. Toto, A. Bansemer, M. R. Kumjian, S. Mishra, and A. V. Ryzhkov, 2016: Insights into riming and aggregation processes as revealed by aircraft, radar, and disdrometer observations for a 27 April 2011 widespread precipitation event, J. Geophys. Res. Atmos., 121, 5846–5863, doi:10.1002/2015JD024537.

Gematronik, 2007: Dual-polarization weather radar handbook. In: Bringi, V.N., Thurai, M.,  Hannesen, R. (Eds.), Selex-SI gematronik, 2nd edition (163 pp.).

Gregorius, T., 1996: GIPSY-OASIS II How it works. Department of Geomatics, University of Newcastle upon Tyne, 167 pp. [Available online at: http://web.gps.caltech.edu/classes/ge167/file/gipsy-oasisIIHowItWorks.pdf.].

Guerova G, J. Jones, J. Dousa, G. Dick, S. De Haan, E. Pottiaux, O. Bock, R. Pacione, G. Elgered, H. Vedel, M. Bender, 2016: Review of the state-of-the-art and future prospects of the ground-based GNSS meteorology in Europe. Atmos Meas Tech Discuss. doi:10.5194/amt-2016-125

de Haan, S. D., S. Barlag, H. K. Baltink, F. Debie, and H. V. Marel, 2004: Synergetic Use of GPS Water Vapor and Meteosat Images for Synoptic Weather Forecasting. J. Appl. Meteoro., 43, 514-518, doi:http://dx.doi.org/10.1175/1520-0450(2004)043<0514: SUOGWV>2.0.CO;2.

de Haan, S. D., 2006: Measuring Atmospheric Stability with GPS. J. Appl. Meteoro. Climatol., 45, 467-475, doi: http://dx.doi.org/10.1175/JAM2338.1.

de Haan, S. D., and I. Holleman, 2009: Real-Time Water Vapor Maps from a GPS Surface Network: Construction, Validation, and Applications. J. Appl. Meteoro. Climatol., 48, 1302-1316, doi: http://dx.doi.org/10.1175/2008JAMC2024.1.

Holloway, C. and J. Neelin, 2010: Temporal Relations of Column Water Vapor and Tropical Precipitation. J. Atmos. Sci.,
5    67, 1091–1105, doi: http://dx.doi.org/10.1175/2009JAS3284.1.

Iwabuchi, T., C. Rocken, Z. Lukes, L. Mervat, J. Johnson, and M. Kanzaki, 2006: PPP and Network True Real-time 30 sec Estimation of ZTD in Dense and Giant Regional GPS Network and the Application of ZTD for Nowcasting of Heavy Rainfall. Proceedings of the 19th International Technical Meeting of the Satellite Division of The Institute of Navigation (ION GNS 2006), 1902-1909.

10   IERS Conventions, 2010: Gérard Petit and Brian Luzum (eds.). (IERS Technical Note ; 36) Frankfurt am Main: Verlag des Bundesamts für Kartographie und Geodäsie, 2010. 179 pp., ISBN 3-89888-989-6 https://www.iers.org/IERS/EN/Data Products/ Conventions/conventions.html.

Jerrett, D., and J. Nash, 2001: Potential Uses of Surface Based GPS Water Vapour Measurements for Meteorological Purposes. Phys. Chem. Earth (A), 26, 457-461, doi:10.1016/S1464-1895(01)00083-7.

15   Joss, J., and A. Waldvogel, 1967: Ein Spektrograph für Niederschlags-tropfen mit automatischer Auswertung (A spectrograph for rain drops with automatical analysis). Pure Appl. Geophys., 68, 240–246, http://dx.doi.org/10.1007/BF00874898.

Kinnell, P. I. A., 1976: Some Observations on the Joss-Waldvogel Rainfall Disdrometer, J. Appl. Meteor., 15, 499–502, doi:http://dx.doi.org/10.1175/1520-0450(1976)015<0499:SOOT JW>2.0.CO;2.

20   Kursinski, E. R., R. A. Bennett, D. Gochis, S. I. Gutman, K. L. Holub, R. Mastaler, C. Minjarez Sosa, I. Minjarez Sosa, and T. van Hove (2008), Water vapor and surface observations in northwestern Mexico during the 2004 NAME Enhanced Observing Period, Geophys. Res. Lett., 35, L03815, doi:10.1029/2007GL031404.

Lyard, F., F. Lefèvre, T. Letellier, and O. Francis, 2006: Modelling the global ocean tides: a modern insight from FES2004, Ocean Dynamics, 56, 394-415, doi:http://dx.doi.org/10.1007/s10236-006-0086-x.

25   Machado, L. A. T., M. A. F. Silva Dias, C. Morales, G. Fisch, D. Vila, R. Albrecht, S. J. Goodman, A. J. P. Calheiros, T. Biscaro, C. Kummerow, J. Cohen, D. Fitzjarrald, E. L. Nascimento, M. S. Sakamoto, C. Cunningham, J. P. Chaboureau, W. A. Petersen, D. K. Adams, L. Baldini, C. F. Angelis, L. F. Sapucci, P. Salio, H. M. J. Barbosa, E. Landulfo, R. A. F. Souza, R. J. Blakeslee, J. Bailey, S. Freitas, W. F. A. Lima, and A. Tokay, 2014: The Chuva Project: How Does Convection Vary across Brazil? Bull. Amer. Meteor. Soc., 95, 1365–1380, doi:http://dx.doi.org/10.1175/BAMS-D-13-
30   00084.1.

Madhulatha, A., M. Rajeevan, M. Venkat Ratnam, J. Bhate, and C. V. Naidu, 2013: Nowcasting severe convective activity over southeast India using ground-based microwave radiometer observations. J. Geophys. Res., 118, doi:10.1029/2012JD018174.

Mazany, R. A., S. Businger, S. I. Gutman, and W. Roeder, 2002: A lightning prediction index that utilizes GPS integrated precipitable water vapor. Wea. Forecasting, 17, 1034–1047, doi:10.1175/1520-0434(2002)017<1034:ALPITU>2.0.CO;2.

Montenbruck, O., R. Schmid, F. Mercier, P. Steigenberger, C. Noll, R. Fatkulin, S. Kogure, A.S. Ganeshan, 2015: GNSS satellite geometry and attitude models, Advances in Space Research, 56, 6, 1015-1029. Doi:http://dx.doi.org/10.1016/j.asr.2015.06.019.

Moore, A., I. Small, S. Gutman, Y. Bock, J. Dumas, P. Fang, J. Haase, M. Jackson and J. Laber, 2015: National Weather Service Forecasters Use GPS Precipitable Water Vapor for Enhanced Situational Awareness during the Southern California Summer Monsoon. Bull. Amer. Meteor. Soc., 96, 1867–1877, doi: 10.1175/BAMS-D-14-00095.1. Muller, C. J., L. E. Back, P. A. O'Gorman, and K. A. Emanuel, 2009: A model for the relationship between tropical precipitation and column water vapor. Geophys. Res. Lett., 36, L16804, doi:10.1029/2009GL039667.

Percival, D. B., and A. T. Walden, 2000: Wavelet Methods for Time Series Analysis. Cambridge University, 594 pp.

Raymond, D. J. 1987: A Forced Gravity Wave Model of Serf-Organizing Convection. J. Atmo. Sci., 44, 23, 3528-3543.

Rocken, C., T. VanHove, M. Rothacher, F. Solheim, R. Ware, M. Bevis, S. Businger, and R. Chadwick, 1994: Towards near-real-time estimation of atmospheric water vapor with GPS. Eos Trans. AGU, 7544, Fall Meet. Suppl., 173.

Sapucci, L. F., 2014: Evaluation of Modeling Water-Vapor-Weighted Mean Tropospheric Temperature for GNSS-Integrated Water Vapor Estimates in Brazil. J. Appl. Meteor. Climatol., 53, 715–730, doi:http://dx.doi.org/10.1175/JAMC-D-13-048.1.

Sapucci, L. F., L. A. T. Machado, J. F. G. Monico, and A. Plana-Fattori, 2007: Intercomparison of Integrated Water Vapor Estimative from multi-sensor in Amazonian Regions. J. Atmos. Oceanic Technol., 24, 1880–1894, doi:http://dx.doi.org/10.1175/JTECH2090.1.

Serra, Yolande L., David K. Adams, Carlos Minjarez-Sosa, James M. Moker, Jr., Avelino Arellano, Christopher Castro, Arturo Quintanar, Luis Carlos Alatorre, Alfredo Granados-Olivas, Enrique Vazquez, Kirk L. Holub, Charles DeMets, 2016: The North American Monsoon GPS Transect Experiment 2013. *Bull. Amer. Meteor. Soc.,* **97**, 2103–2115, doi: 10.1175/BAMS-D-14-00250.1. Sato, T and F. Kimura, 2005: Diurnal Cycle of Convective Instability around the Central Mountains in Japan during the Warm Season. J. Atmo. Sci., 62, 1626-1636.

Schmid, R., P. Steigenberger, G. Gendt, M. Ge, M. Rothacher, 2007: Generation of a consistent absolute phase-center correction model for GPS receiver and satellite antennas. Journal of Geodesy, 81: 781. 81, 12, 781–798, doi:10.1007/s00190-007-0148-y

Shangguan, M., S. Heise, M. Bender, G. Dick, M. Ramatschi, and J. Wickert, 2015: Validation of GPS atmospheric water vapor with WVR data in satellite tracking mode, Ann. Geophys., 33, 55-61, doi:10.5194/angeo-33-55-2015.

Shi, J., X. Chaoqian, G. Jiming and G. Yang, 2015: Real-Time GPS precise point positioning-based precipitable water vapor estimation for rainfall monitoring and forecasting, IEEE Trans. Geosci. Remote Sens, vol.53, pp. 3452-3459, doi: 10.1109/TGRS.2014.2377041.

Solheim, F. S., J. Vivekanandan, R. H. Ware, and C. Rocken, 1999: Propagation delays induced in GPS signals by dry air, water vapor, hydrometeors, and other particulates. J. Geophys. Res., 104, 9663–9670, doi:10.1029/1999JD900095.

Testud, J., E. L. Bouar, E. Obligis, and M. Ali-Mehenni, 2000: The Rain Profiling Algorithm Applied to Polarimetric Weather Radar. J. Atmos. Oceanic Technol., 17, 332–356, doi:http://dx.doi.org/10.1175/1520-0426(2000)017<0332:TRPAAT>2.0.CO;2.

Torrence, C., and G. P. Compo, 1998: A Practical Guide to Wavelet Analysis. Bull. Amer. Meteor. Soc., 79, 61-78.

Whitcher, P., D. B. Guttorp, and B. Percival, 2000: Wavelet analysis of covariance with application to atmospheric time series. J. Geophys. Res., 105, 14941–14962, doi::10.1029/2000JD900110.

Wolfe, D. E., and S. I. Gutman, 2000: Developing an operational, surface-based, GPS, water vapor observing system for NOAA: Network Design and Results. J. Atmos. Ocean. Technol., 17, 426-440.

Zumberge, J. F., M. B. Heflin, D. C. Jefferson, M. M. Watkins, and F. H. Webb (1997), Precise point positioning for the efficient and robust analysis of GPS data from large networks, J. Geophys. Res., 102(B3), 5005–5017, doi:10.1029/96JB03860.

**Tables**

**Table 1: Configuration items associated with ZTD estimates and respective values used GPS data processing using GOA-II software.**

| Configuration item | Parameter used | Comments |
|---|---|---|
| GPS data file for GOA-II | Rinex format in sampling rate of 1 second | Data collection from CHUVA |
| Mapping function | Global Mapping Function (Boehm et al. 2006) | Selected in the data processing |
| Cut-off elevation angle: | 10° | Selected in the data processing |
| Elevation weight in the Mapping function | Elevation dependent and apply 3 coefficients depend on the latitude and height above sea level of the observing site and on the DoY | (Boehm et al. 2006) |
| ZHD a priori | Constant using a Tropospheric model in function of high of the GPS station (2.126 m) | Default values |
| ZWD a priori | Constant (0.10 m) | Default values |
| Process noise for tropospheric delay | Random walk | Default values |
| Maximum drift for tropospheric delay | 8.333E-8 (km per square-root second); | Default values |
| Tropospheric Gradients estimates | yes | Default values |
| Maximum drift for Gradients estimates | 8.333E-9 (km per square-root second) | Default values |
| Sampling rate of the ZTD estimates | 60 seconds | Selected in the data processing |

Comentário [s52]: Reviser2E18d

**Table 2: Precipitation events observed by radar during CHUVA Vale experiment in different extension terciles as a function of the precipitation fractions above 35 mm h$^{-1}$.**

| Event | DoY | Maximal radar precipitation UTC Time (hh:mm) | Precipitation fraction observed by XPol radar (%) | | | Terciles |
|---|---|---|---|---|---|---|
| | | | Above 50 mm h$^{-1}$ | Above 35 mm h$^{-1}$ | Above 20 mm h$^{-1}$ | |
| 1 | 348 | 02:42 | 73 | 85 | 95 | |
| 2 | 354 | 21:12 | 28 | 45 | 63 | |
| 3 | 341 | 18:36 | 36 | 41 | 45 | Upper |
| 4 | 315 | 17:12 | 26 | 41 | 49 | tercile |
| 5 | 335 | 19:24 | 30 | 38 | 42 | |
| 6 | 352 | 20:00 | 2 | 33 | 84 | |
| 7 | 342 | 16:36 | 24 | 27 | 28 | |
| 8 | 326 | 21:18 | 8 | 26 | 45 | |
| 9 | 343 | 01:06 | 4 | 9 | 15 | Middle |
| 10 | 338 | 20:18 | 0 | 8 | 19 | tercile |
| 11 | 332 | 19:18 | 5 | 5 | 5 | |
| 12 | 333 | 19:42 | 0 | 2 | 8 | |
| 13 | 314 | 21:06 | 2 | 2 | 7 | |
| 14 | 327 | 00:36 | 0 | 1 | 25 | |
| 15 | 358 | 23:00 | 0 | 1 | 10 | Lower |
| 16 | 317 | 21:48 | 0 | 1 | 2 | tercile |
| 17 | 318 | 08:48 | 0 | 0 | 19 | |
| 18 | 331 | 17:12 | 0 | 0 | 2 | |

**Table 3. Statistical measurements of the GPS-PWV derivative for different extension terciles of precipitation events.**

| Statistical Measurements | Other cases | Terciles | | |
|---|---|---|---|---|
| | | Lower | Middle | Upper |
| Average value (mm h$^{-1}$) | +0.04 | -0.18 | -0.38 | 0.13 |
| Standard deviation (mm h$^{-1}$) | ±2.52 | ±3.18 | ±4.76 | ±5.57 |
| Median (mm h$^{-1}$) | 0.00 | 0.00 | +0.29 | +0.65 |
| Mode (mm h$^{-1}$) | 0.00 | 0.00 | +1.00 | +1.00 |
| Maximal value (mm h$^{-1}$) | +21.13 | +8.42 | +11.00 | +13.25 |
| Minimal value (mm h$^{-1}$) | -19.07 | -6.99 | -17.30 | -14.15 |
| % > +9.5 mm h$^{-1}$ | 0.21% | 0.00% | 0.78% | 7.81% |
| % < -9.5 mm h$^{-1}$ | 0.38% | 0.00% | 4.68% | 5.47% |

**Figures**

[Figure]

**Figure 1: Illustration of the CHUVA Vale experiment, in which the sites where the XPol radar, GPS receiver and disdrometer were installed are indicated. The area of 4.4 km per 4.4 km around the GPS station is highlighted in this figure over the precipitation field observed by XPol radar on December 14th (DoY 348) of 2011. Some details about the composition of this area using the points of the XPol gridded map are additionally presented.**

[Figure]

**Figure 2: Time series of the precipitation and GPS-PWV obtained during CHUVA Vale campaign: (a) GPS-PWV time series; (b) precipitation intensity observed by Joss disdrometer; (c) 95th percentile of the precipitation intensity observed by XPol radar in the area of 4.4 km per 4.4 km centered on the GPS antenna; and (d) precipitation fraction in the area of 4.4 km per 4.4 km centered on the GPS antenna, where the black bar is the fraction above 20 mm h⁻¹, the dark gray bar is the fraction above 35 mm h⁻¹ and the light gray bar is the fraction above 50 mm h⁻¹.**

**Comentário [s53]:** Reviser2E30b

[Figure]

**Figure 3: GPS-PWV jump observed in the 2-hour period before a heavy storm occurred during DoY 341 (1836 UTC 7 December 2011).**

**Comentário [s54]:** Reviser2E39

[Figure]

**Figure 4:** Other cases of GPS-PWV jumps observed before precipitation events with fraction above 35 mm h$^{-1}$ of different extension occurred during (a) DoY 315 (1712 UTC), (b) DoY 332 (1918 UTC) and (c) DoY 358 (2300 UTC).

**Comentário [s55]:** Reviser2E39

[Figure]

**Figure 5: Composite the GPS-PWV time series 60 minutes before and 60 minutes after of 18 precipitation events listed by Table 2 and the distribution of the time lag between the maximum GPS-PWV and the time of the maximum rainfall of these events (precipitation fraction above 35 mm h[-1] in the area of 4.4 km per 4.4 km centered on the GPS).**

[Figure]

**Figure 6: GPS-PWV time series at IEAV station (a) and wavelet power spectrum analysis (b). The cone of influence is plotted in green dashed line to avoid interpretations in border regions.** The precipitation intensity values observed by XPol radar (95[th] percentile) in the area of 4.4 km per 4.4 km around the GPS antenna are included in the bottom of this plot.

[Figure]

**Figure 7: Similar to Fig. 6 for a shorter period (from 0000 UTC on DoY 340 to 0000 UTC on DoY 344) to emphasize the details in the wavelet power spectrum during the occurrence of precipitation events. This period is inside of the cone of influence. A vertical line was drawn for each precipitation maximum to make easier the analysis.**

[Figure]

**Figure 8: Wavelet correlation values between GPS-PWV and 95[th] percentile of the precipitation intensity from XPol radar time series as a function of the different wavelet scales represented by their respective time periods. The correlation is estimated considering the lag zero.. The 95% Confidence Interval for each WCC is estimated considering a Gaussian Distribution after applying the Fisher`s Z Transformation (Whitcher et al. 2000).**

[Figure]

**Figure 9: Wavelet correlation values between GPS-PWV and precipitation time series in different wavelet scales (represented by their respective time periods) for (a) percentage of points above 20 mm h$^{-1}$ observed by radar around the GPS antenna; (b) the same above 35 mm h$^{-1}$; (c) and the same above 50 mm h-1. The correlation is estimated considering the lag zero. The 95% Confidence Interval for each WCC is estimated considering a Gaussian Distribution after applying the Fisher`s Z Transformation (Whitcher et al. 2000).**

[Figure]

**Figure 10: Wavelet cross-correlation values between GPS-PWV and (a) percentage of points above 20 mm h$^{-1}$ observed by radar around the GPS antenna; (b) the same above 35 mm h$^{-1}$; (c) and the same above 50 mm h-1; as function of lead-lag for different time periods. The dotted line represents the 95% confidence interval.**

[Figure]

5   **Figure 11: Spearman correlation histograms of maximum positive (a) and minimum negative (b) correlations as functions of the lag of occurrence for GPS-PWV values and precipitation events. All 18 precipitation events listed in Table 2 were taken into account in this analysis.**

[Figure]

**Figure 12: Frequency polygons of the GPS-PWV derivatives calculated over the period of 60 minutes before precipitation events for different terciles of the precipitation fraction above 35 mm h$^{-1}$ observed using XPol radar.**

Time series of PWV derivative before precipitation for upper tercile

[Figure]

**Figure 13: Time series of GPS-PWV derivative before precipitation for upper terciles. The mean values of GPS-PWV derivative, precipitation and GPS-PWV for this events are also shown in this figure.**